# Rethinking Self-Distillation: Label Averaging and Enhanced Soft Label Refinement with Partial Labels

**Hyeonsu Jeong & Hye Won Chung**
School of Electrical Engineering
Korea Advanced Institute of Science and Technology (KAIST)
Daejeon, South Korea
{hsjeong1121, hwchung}@kaist.ac.kr

## Abstract

We investigate the mechanisms of self-distillation in multi-class classification, particularly in the context of linear probing with fixed feature extractors where traditional feature learning explanations do not apply. Our theoretical analysis reveals that multi-round self-distillation effectively performs label averaging among instances with high feature correlations, governed by the eigenvectors of the Gram matrix derived from input features. This process leads to clustered predictions and improved generalization, mitigating the impact of label noise by reducing the model's reliance on potentially corrupted labels. We establish conditions under which multi-round self-distillation achieves 100% population accuracy despite label noise. Furthermore, we introduce a novel, efficient single-round self-distillation method using refined partial labels from the teacher's top two softmax outputs, referred to as the PLL student model. This approach replicates the benefits of multi-round distillation in a single round, achieving comparable or superior performance–especially in high-noise scenarios–while significantly reducing computational cost.

## 1    Introduction

Knowledge distillation, initially introduced by Hinton et al. (2015), has emerged as a powerful technique in deep learning, where a smaller student model is trained to replicate the predictions of a larger, more complex teacher model. In classification tasks, the teacher's output probability distribution, often obtained via softmax, serves as "soft targets" for training the student model. This approach has proven effective across various domains including computer vision, speech recognition, and natural language processing (Chebotar & Waters, 2016; Cui et al., 2017; Asif et al., 2019; Liu et al., 2019), yielding smaller, more efficient models that often generalize better than similarly sized models trained without distillation.

Building upon this foundation, self-distillation eliminates the need for a larger teacher model by using a model of the same architecture as its own teacher. In this process, models of the same size are trained over multiple rounds, with each iteration learning from the outputs of the previous one. Surprisingly, self-distillation has been shown to enhance performance, especially when repeated over multiple rounds (Furlanello et al., 2018; Zhang et al., 2019). This improvement is intriguing because it cannot be fully explained by traditional theories that focus on a student mimicking a more complex teacher, prompting a deeper exploration into the mechanisms underlying self-distillation.

Recent research has begun to provide diverse explanations for the benefits of self-distillation. For example, Allen-Zhu & Li (2022) describe self-distillation as a feature-learning process in neural networks, particularly under a multi-view data distribution where each class possesses multiple orthogonal features (e.g., "windows" and "wheels" for cars). The teacher's softmax outputs help the student model learn a richer set of features, improving generalization. Another line of research emphasizes that the benefits of self-distillation extend beyond neural networks to classical regression settings (Mobahi et al., 2020; Dong et al., 2019; Das & Sanghavi, 2023). In these contexts, self-distillation acts as a regularizer, preventing overfitting and promoting generalization.

Despite these insights, several critical questions remain unaddressed. First, **can self-distillation provide benefits in training a simple linear classifier atop a fixed pre-trained feature extractor–a process known as linear probing?** In this setting, traditional explanations involving feature learning do not apply, as the feature extractor is fixed. With the emergence of large-scale pre-trained networks, or foundation models (Bommasani et al., 2021), linear probing has become an increasingly popular method to leverage these networks' capabilities for downstream tasks. Yet, it remains unclear whether self-distillation can enhance this process without additional feature learning, and how repeated rounds of distillation impact the model outputs.

Furthermore, **can the benefits of multi-round self-distillation be achieved more efficiently–in just a single round–by developing methods that replicate these gains without repeated training cycles?** While multi-round self-distillation has shown performance improvements, it can be computationally intensive due to the need for multiple training iterations. Developing efficient methods to achieve similar gains without multiple rounds is therefore of significant practical interest.

**Our Contributions**    In this paper, we address these questions by:

- Analyzing Self-Distillation in Linear Probing: Our analysis reveals that self-distillation effectively performs label averaging among instances with high feature correlations when generating predictions on the training data. This averaging is governed by the eigenvectors of the Gram matrix derived from input features (defined in Sec. 2.2). As distillation progresses, the top-$K$ eigenvectors–where $K$ is the number of classes–dominate this process, leading to increasingly clustered predictions around instances from the same ground-truth class. This finding provides insights into how self-distillation benefits even in simple linear probing without feature updates.

- Quantifying Gains in Label-Noise Scenarios: We quantify the number of distillation rounds needed to achieve 100% population accuracy in presence of label noise–that is, correctly predicting the ground-truth label for any training instance as the number of instances goes to infinity. This number depends on the relative strength between signals from the given (potentially noisy) labels and the averaged labels of instances with high feature correlation. As the latter signal strengthens with more distillation rounds, the student model can correctly classify samples with noisy labels after a few iterations, demonstrating the robustness imparted by multi-round self-distillation.

- Introducing an Efficient Single-Round Method using Partial Label Refinement: We introduce a novel self-distillation approach that replicates the benefits of multi-round distillation in a single round. This method trains the student with refined partial labels, focusing on the top two softmax outputs from the teacher. We demonstrate that this approach outperforms traditional multi-round self-distillation in high-noise scenarios, while significantly reducing computational cost. Our findings are supported by experiments on synthetic and real datasets (Figure 3 and Section 6).

**Related Works**    Since its introduction by Hinton et al. (2015) and subsequent extensions (Romero et al., 2014; Tian et al., 2019; Lopez-Paz et al., 2015), several studies have sought to theoretically analyze the effects of knowledge distillation and self-distillation. Many of these works rely on specific assumptions about datasets, model architectures, or loss functions to facilitate tractable analysis. Mobahi et al. (2020); Dong et al. (2019) explored self-distillation in classical regression settings using mean squared error loss. They demonstrated that multi-round distillation reduces the number of basis functions in the model, which helps prevent overfitting initially but can lead to underfitting after excessive rounds. In another line of research focusing on binary classification with cross-entropy loss, Phuong & Lampert (2019); Ji & Zhu (2020); Das & Sanghavi (2023) analyzed the effects of distillation on linear networks or logistic regression under label noise. Das & Sanghavi (2023) specifically identified the range of label corruption where the student model outperforms the teacher.

In contrast to these studies, our paper examines multi-round self-distillation for multi-class classification with cross-entropy loss, particularly in the context of linear probing where the feature extractor is fixed and feature learning is absent. We demonstrate that the effect of self-distillation can be interpreted as label averaging among highly correlated instances, offering a new perspective on the mechanism of self-distillation. While Allen-Zhu & Li (2022) explain the benefits of self-distillation through feature learning under a multi-view data distribution, our results show that significant benefits can be achieved through label averaging alone, even without feature evolution. Specifically, our theoretical analyses (Theorems 3.1–4.1) illustrate how self-distillation leads to clustered outputs based on feature correlations, enhancing generalization by reducing dependence on given (potentially

noisy) labels. Additionally, our findings can be extended to include feature learning by considering the evolution of the feature map over multiple distillation rounds (as discussed in Corollary C.1).

Our work also introduces a novel method for efficiently distilling "dark knowledge" from the teacher model. Instead of utilizing the full softmax output as in traditional distillation, we refine the teacher's outputs into a set of feasible labels by focusing on the top two predictions. This approach is analogous to Partial Label Learning (PLL) (Cour et al., 2011), where each training sample is associated with a set of candidate labels rather than a single ground truth label. Various PLL methods aim to improve classification accuracy by weighting candidate labels using logits (Lv et al., 2020; Wang et al., 2021) or leveraging feature space topology (Wang et al., 2019; Xu et al., 2019; Lyu et al., 2019). Our method differs by directly employing the refined partial labels derived from the teacher's outputs, achieving the same benefits as multi-round distillation in just one round. Importantly, our approach outperforms traditional multi-round distillation in high-noise scenarios, as it consistently includes the ground-truth label among the top two predictions, as demonstrated by Theorem 5.1.

## 2 PRELIMINARIES AND OVERVIEW OF RESULTS

**Notation** We denote vectors (matrices) by lowercase (uppercase) bold letters. For a vector $\mathbf{v}$ and a matrix $\mathbf{A}$, let $[\mathbf{v}]_i$ or $v_i$ represent the $i$-th entry, and $[\mathbf{A}]_{i,j}$ the $(i,j)$-th entry. Let $\mathbf{e}(i)$ be a unit vector with 1 in the $i$-th entry. We use $\|\cdot\|_F$ for the Frobenius norm. Let $\mathbf{1}_{n\times m} \in \mathbb{R}^{n\times m}$ be all-one matrix.

### 2.1 SETTING: SELF-DISTILLATION FOR $K$-CLASS CLASSIFICATION

We consider the problem of $K$-class classification, where each sample $\mathbf{x} \in \mathcal{X}$ is associated with a ground-truth class label $y(\mathbf{x}) \in \mathcal{Y} := \{1, 2, \ldots, K\}$. Let $\mathcal{P}$ denote the underlying distribution over the sample space $\mathcal{X} \times \mathcal{Y}$. Assume that we have a feature map $\phi : \mathcal{X} \to \tilde{\mathcal{X}} \subset \mathbb{R}^d$. We are given a dataset $\{(\phi(\mathbf{x}_i), \hat{y}_i)\}_{i=1}^{Kn}$ consisting of $Kn$ feature-label pairs, where each $(\phi(\mathbf{x}_i), \hat{y}_i) \in \tilde{\mathcal{X}} \times \mathcal{Y}$. For simplicity, we assume that the feature vectors are normalized, i.e., $\|\phi(\mathbf{x}_i)\|_2 = 1, \forall i \in [Kn]$. The given label $\hat{y}_i$ may be noisy, while the ground-truth label of $\mathbf{x}_i$ is $y_i := y(\mathbf{x}_i)$. We further assume that the dataset is balanced with respect to both the ground-truth labels and the given labels, meaning that for each class $k \in [K]$, there are $n$ samples with $y_i = k$ and $n$ samples with $\hat{y}_i = k$.

We consider a classifier $\boldsymbol{\theta} \in \mathbb{R}^{d\times K}$ based on multinomial logistic regression (softmax regression). For an input $\mathbf{x}$, the classifier outputs $\mathbf{y} = \sigma(\boldsymbol{\theta}^\top \phi(\mathbf{x}))$, where $\sigma : \mathbb{R}^K \to (0, 1)^K$ is the softmax function. The predicted class label for input $\mathbf{x}$ is then $\arg\max_{k\in[K]}[\sigma(\boldsymbol{\theta}^\top \phi(\mathbf{x}))]_k$. The population accuracy is defined as $\mathbb{E}_{(\mathbf{x}, y(\mathbf{x}))\sim\mathcal{P}} \left[\mathbb{1}\left(\arg\max_{k\in[K]}[\sigma(\boldsymbol{\theta}^\top \phi(\mathbf{x}))]_k = y(\mathbf{x})\right)\right]$. This setting is equivalent to linear probing in the context of neural networks, where the network consists of a fixed feature extractor $\phi : \mathcal{X} \to \tilde{\mathcal{X}}$ followed by a tunable fully connected linear layer parameterized by $\boldsymbol{\theta}$ and a softmax layer. As the training objective, we consider the $\ell_2$-regularized cross-entropy (CE) loss between the target labels (e.g., $\mathbf{e}(\hat{y}_i)$ for the teacher model) and the model's predictions $\mathbf{y}_i := \sigma(\boldsymbol{\theta}^\top \phi(\mathbf{x}_i))$:

$$f_T(\boldsymbol{\theta}) = \frac{1}{Kn}\sum_{i=1}^{Kn} \mathsf{CE}(\mathbf{e}(\hat{y}_i), \mathbf{y}_i) + \frac{\lambda\|\boldsymbol{\theta}\|_F^2}{2}, \tag{1}$$

where $\lambda > 0$ is a regularization parameter, and $\mathsf{CE}(\mathbf{e}(\hat{y}_i), \mathbf{y}_i) := -\sum_{k=1}^K [\mathbf{e}(\hat{y}_i)]_k \log[\mathbf{y}_i]_k$.

In self-distillation, the student model, which shares the same architecture as the teacher, minimizes the same objective (1) over the same training set $\{\phi(\mathbf{x}_i)\}$, but with new targets obtained from the teacher's output. Let $\boldsymbol{\theta}_T := \arg\min_{\boldsymbol{\theta}} f_T(\boldsymbol{\theta})$ be the teacher's optimal parameters. The new target for an input $\mathbf{x}_i$ becomes the teacher's output prediction $\mathbf{y}_i^{(T)} := \sigma(\boldsymbol{\theta}_T^\top \phi(\mathbf{x}_i))$ rather than the given label $\mathbf{e}(\hat{y}_i)$. Self-distillation can be repeated iteratively, where the $t$-th distilled model uses the outputs of the $(t-1)$-th model as its targets. Denoting the output of the $(t-1)$-th model for the $i$-th instance as $\mathbf{y}_i^{(t-1)}$, the training objective for the $t$-th student model $\boldsymbol{\theta} \in \mathbb{R}^{d\times K}$, for $t \in \mathbb{Z}^+$, is given by

$$f^{(t)}(\boldsymbol{\theta}) = \frac{1}{Kn}\sum_{i=1}^{Kn} \mathsf{CE}(\mathbf{y}_i^{(t-1)}, \mathbf{y}_i^{(t)}) + \frac{\lambda\|\boldsymbol{\theta}\|_F^2}{2}, \tag{2}$$

for $\mathbf{y}_i^{(t)} := \sigma(\boldsymbol{\theta}^\top \phi(\mathbf{x}_i))$. $f^{(1)}(\boldsymbol{\theta})$ is the same objective as the teacher model (1) when $\mathbf{y}_i^{(0)} := \mathbf{e}(\hat{y}_i)$.

The optimal $\boldsymbol{\theta}$ that minimizes (2) for the $t$-th distilled model, denoted by $\boldsymbol{\theta}^{(t)}$, is expressed as

$$\boldsymbol{\theta}^{(t)\top} = \sum_{i=1}^{Kn} \frac{\left(\mathbf{y}_i^{(t-1)} - \mathbf{y}_i^{(t)}\right)}{Kn\lambda} \phi(\mathbf{x}_i)^\top = \sum_{i=1}^{Kn} \boldsymbol{\alpha}_i^{(t)} \phi(\mathbf{x}_i)^\top \text{ where } \boldsymbol{\alpha}_i^{(t)} := \frac{\left(\mathbf{y}_i^{(t-1)} - \mathbf{y}_i^{(t)}\right)}{Kn\lambda}, \quad (3)$$

since it satisfies the condition $\frac{df^{(t)}(\boldsymbol{\theta})}{d\boldsymbol{\theta}} = \frac{1}{Kn} \sum_{i=1}^{Kn} \phi(\mathbf{x}_i)(\mathbf{y}_i^{(t)} - \mathbf{y}_i^{(t-1)})^\top + \lambda\boldsymbol{\theta} = \mathbf{0}$. Note that $\boldsymbol{\theta}^{(t)}$ is a linear combination of the training instances $\{\phi(\mathbf{x}_i)\}$, with coefficients $\boldsymbol{\alpha}_i^{(t)}$ determined by the difference between the target $\mathbf{y}_i^{(t-1)}$ and the output $\mathbf{y}_i^{(t)}$. The key question is how the optimal classifier $\boldsymbol{\theta}^{(t)}$ and the corresponding output $\mathbf{y}^{(t)} = \sigma(\boldsymbol{\theta}^{(t)\top}\phi(\mathbf{x}))$ evolve as $t$ increases.

To answer this question, we need to analyze the coefficients $\{\boldsymbol{\alpha}_i^{(t)}\}_{i=1}^{Kn}$ in (3), which satisfy

$$Kn\lambda\boldsymbol{\alpha}_i^{(t)} = \mathbf{y}_i^{(t-1)} - \sigma(\boldsymbol{\theta}^{(t)\top}\phi(\mathbf{x}_i)) = \mathbf{y}_i^{(t-1)} - \sigma\left(\sum_{j=1}^{Kn} \langle\phi(\mathbf{x}_i), \phi(\mathbf{x}_j)\rangle\boldsymbol{\alpha}_j^{(t)}\right), \forall i \in [Kn], \quad (4)$$

derived from the definition. Solving (4) for $\{\boldsymbol{\alpha}_i^{(t)}\}$ is challenging due to the non-linearity of the softmax function $\sigma$. To address this, we consider the following linear approximation of the softmax:

**Assumption 1.** We approximate the softmax $\sigma(\mathbf{v})$ for the logits $\mathbf{v} \in \mathbb{R}^K$ by

$$\sigma(\mathbf{v}) = \left[\frac{\exp(v_1)}{\sum_{i=1}^K \exp(v_i)}, \cdots, \frac{\exp(v_K)}{\sum_{i=1}^K \exp(v_i)}\right] \approx \left[\frac{1 + v_1}{K + \sum_{i=1}^K v_i}, \cdots, \frac{1 + v_K}{K + \sum_{i=1}^K v_i}\right]. \quad (5)$$

This linear approximation has been considered in the analysis of knowledge distillation, as initially explored by Hinton et al. (2015), where the softmax output layer converts the logits $\mathbf{v}$ using $\sigma(\mathbf{v}/\tau)$ with a high temperature $\tau > 0$. When $\tau = 1$, the approximation remains valid if the logits are of sufficiently small magnitude. From (3), the logits of the $t$-th distilled model for an input $\phi(\mathbf{x}_i)$ are given by $\boldsymbol{\theta}^{(t)\top}\phi(\mathbf{x}_i) = \sum_{j=1}^{Kn} \langle\phi(\mathbf{x}_i), \phi(\mathbf{x}_j)\rangle\boldsymbol{\alpha}_j^{(t)}$. This is equivalent to $\sum_{j=1}^{Kn} \frac{\langle\phi(\mathbf{x}_i), \phi(\mathbf{x}_j)\rangle}{Kn\lambda}\left(\sigma(\boldsymbol{\theta}^{(t-1)}\phi(\mathbf{x}_j)) - \sigma(\boldsymbol{\theta}^{(t)}\phi(\mathbf{x}_j))\right)$. Thus, the magnitude of the logits becomes small if the average difference between the softmax outputs of the teacher (i.e., the $(t-1)$-th model) and the student (i.e., the $t$-th model) across the training instances is sufficiently small. In Appendix D.1, we empirically demonstrate that this approximation holds well for class-balanced datasets with a sufficient number of samples. Using the fact that the logits are zero-mean, i.e., $\sum_{i=1}^K v_i = 0$, which holds since $\mathbf{v} = \boldsymbol{\theta}^\top\phi(\mathbf{x}_i) = \sum_{j=1}^{Kn} \langle\phi(\mathbf{x}_i), \phi(\mathbf{x}_j)\rangle\boldsymbol{\alpha}_j$ and $\sum_{k=1}^K [\boldsymbol{\alpha}_j]_k = 0$ for all $j$, we have $\sigma(\mathbf{v}) \approx \sigma_L(\mathbf{v}) := \frac{1}{K}\mathbf{1}_K + \frac{1}{K}\mathbf{v}$. Thus, the softmax output can be approximated as a uniform vector $\mathbf{1}_K/K$ perturbed by a (scaled) logit $\mathbf{v}$. Under this approximation, we derive closed-form expressions for $\{\boldsymbol{\alpha}_i^{(t)}\}$ in (4) and analyze how the model's outputs change over multiple rounds of self-distillation.

## 2.2 OVERVIEW OF OUR RESULTS

Given a set of input features $\{\phi(\mathbf{x}_i)\}_{i=1}^{Kn}$, we define the Gram matrix $\boldsymbol{\Phi} \in \mathbb{R}^{Kn \times Kn}$ as $[\boldsymbol{\Phi}]_{ij} := \langle\phi(\mathbf{x}_i), \phi(\mathbf{x}_j)\rangle, \forall i, j \in [Kn]$, where $\langle\cdot, \cdot\rangle$ denotes the inner product. The eigen-decomposition of $\boldsymbol{\Phi}$ is written as $\boldsymbol{\Phi} = \sum_{i=1}^{Kn} \lambda_i \mathbf{v}_i \mathbf{v}_i^\top$, with orthonormal eigenvectors $\mathbf{v}_1, \ldots, \mathbf{v}_{Kn} \in \mathbb{R}^{Kn}$ and the corresponding eigenvalues $\lambda_1 \geq \cdots \geq \lambda_{Kn} \geq 0$.

**Theorem 2.1** (Informal version of Thm. 3.1). *As the number of self-distillation rounds $t \in \mathbb{Z}^+$ increases, the output $\mathbf{Y}^{(t)} = [\mathbf{y}_1^{(t)}, \ldots, \mathbf{y}_{Kn}^{(t)}] \in \mathbb{R}^{K \times Kn}$ of the $t$-th distilled model for the inputs of the training dataset $\{\phi(\mathbf{x}_i), \hat{y}_i\}_{i=1}^{Kn}$ evolves as*

$$\mathbf{Y}^{(t)} - \frac{1}{K}\boldsymbol{I}_{K \times Kn} = \left(\mathbf{Y}^{(0)} - \frac{1}{K}\boldsymbol{I}_{K \times Kn}\right) \sum_{i=1}^{Kn} \left(\frac{\lambda_i}{K^2 n\lambda + \lambda_i}\right)^t \mathbf{v}_i \mathbf{v}_i^\top, \quad (6)$$

*where $\mathbf{Y}^{(0)} = [\mathbf{e}(\hat{y}_1), \cdots, \mathbf{e}(\hat{y}_{Kn})]$ represents the one-hot encoded vectors of the given labels.*

This theorem reveals that after $t$ rounds of self-distillation, the model's predictions become a weighted average of the given labels, with weights determined by the transformed Gram matrix $\boldsymbol{\Phi}^{(t)} :=$

$\sum_{i=1}^{Kn} (\lambda_i/(K^2 n\lambda + \lambda_i))^t \mathbf{v}_i \mathbf{v}_i^\top$. Note that $\mathbf{\Phi}^{(t)}$ retains the same eigenvectors as the Gram matrix $\mathbf{\Phi}$, but its eigenvalues take the form $(\lambda_i/(K^2 n\lambda + \lambda_i))^t$, maintaining the same decreasing order as the eigenvalues of $\mathbf{\Phi}$ for $i \in [Kn]$. The eigenvalues of $\mathbf{\Phi}^{(t)}$ decay exponentially with $t$ since $\lambda_i/(K^2 n\lambda + \lambda_i) < 1$. Consequently, the label averaging process becomes increasingly dominated by the top eigenvectors of $\mathbf{\Phi}^{(t)}$, as shown in Fig. 2a. See Section 3 for details.

The label averaging effect of multi-round self-distillation is particularly advantageous in label-noise scenarios, as it allows the model's output prediction for each training instance to be influenced not only by the given (potentially noisy) label but also by the labels of other instances with high feature correlations. To quantify the benefits of multi-round self-distillation in the presence of label noise, we consider a low-rank structure for the Gram matrix $\mathbf{\Phi} \in \mathbb{R}^{Kn \times Kn}$:

$$\mathbf{\Phi} = \underbrace{\begin{pmatrix} \phantom{x} \end{pmatrix}}_{\text{class correlation matrix}} + \underbrace{\begin{pmatrix} \phantom{x} \end{pmatrix}}_{\text{superclass correlation matrix}} ; \quad [\mathbf{\Phi}]_{i,j} = \begin{cases} 1, & i = j, \\ c, & y_i = y_j, i \neq j, \\ d, & y_i \neq y_j, h(y_i) = h(y_j), \\ 0, & h(y_i) \neq h(y_j), \end{cases} \quad (7)$$

where $1 > c > d \geq 0$, and $h : [K] \to [R]$ maps each class to a superclass or a set of confusable classes ($R \leq K$). This block-wise structure of the Gram matrix is motivated by input feature correlations observed in real datasets, such as those obtained from neural networks pre-trained on ImageNet, as shown in Fig. 1. Under the structure (7), the top-$K$ eigenvectors of $\mathbf{\Phi}$ reveal clusters of instances sharing the same ground-truth class, inferred from their feature correlations. Consequently, the label averaging by $\mathbf{\Phi}^{(t)}$, as in (6), causes predictions for the instances from the same ground-truth class to become more similar as $t$ increase, as illustrated in Fig. 2b, even when labels are noisy. This clustering mitigates the impact of label noise by reducing the model's reliance on potentially corrupted given labels. Our main result quantifies the benefits of multi-round self-distillation by establishing conditions under which the $t$-th distilled model can achieve 100% population accuracy. These conditions are equivalent to those where the model

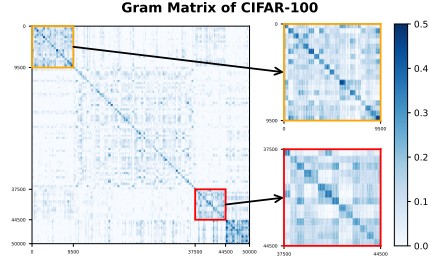

Figure 1: The Gram matrix $\mathbf{\Phi}$ of 50,000 instances of CIFAR-100, extracted from a ResNet34 network, pre-trained on ImageNet. Similar block-wise structures of the Gram matrix are observed for six different real datasets in Fig. 7.

can correctly classify all training data, including label-noisy instances, in the limit of $n \to \infty$.

**Theorem 2.2** (Informal version of Thm. 4.1). *Given a label corruption matrix $\mathbf{C} \in \mathbb{R}^{K \times K}$, where $[\mathbf{C}]_{k,k'} := |\{i : y_i = k, \hat{y}_i = k'\}|/n$, the $t$-th distilled model trained on this label-noisy dataset achieves 100% population accuracy if*

$$[\mathbf{C}]_{k,k} > [\mathbf{C}]_{k,k'} + 1/\left((\tilde{\lambda}_K/\tilde{\lambda}_{K+1})^t - 1\right), \quad \forall k, k' \in [K] \text{ with } k \neq k', \quad (8)$$

*where $\tilde{\lambda}_K = \lambda_K/(K^2 n\lambda + \lambda_K)$ and $\tilde{\lambda}_{K+1} = \lambda_{K+1}/(K^2 n\lambda + \lambda_{K+1})$, with $\lambda_K = 1 - c + n(c - d)$, $\lambda_{K+1} = 1 - c$, as per the assumed Gram matrix in (7).*

This theorem shows that as the number of distillation rounds $t$ increases, the condition for achieving 100% population accuracy becomes less stringent due to the eigenvalue ratio, $(\tilde{\lambda}_K/\tilde{\lambda}_{K+1})^t$, increasing exponentially with $t$. This allows the model to tolerate higher levels of label corruption while still guaranteeing correct classification, highlighting the robustness conferred by multi-round self-distillation in noisy environments. See Section 4 for a detailed analysis.

Lastly, to reduce the computational cost of multiple distillation rounds, we propose a novel method that achieves the benefits of multi-round self-distillation in a single round. The key idea is to refine the teacher's output into partial labels, focusing on the top two predicted classes. Specifically, we assign equal weights (1/2) to the top two label positions in the teacher's softmax output and use this refined target to train the student model.

**Theorem 2.3** (Informal ver. of Thm. 5.1). *The student model trained using partial labels, refined from the teacher's outputs, achieves 100% population accuracy, if $[\mathbf{C}]_{k,k} > [\mathbf{C}]_{k,k'}, \forall k, k'(\neq k) \in [K]$.*

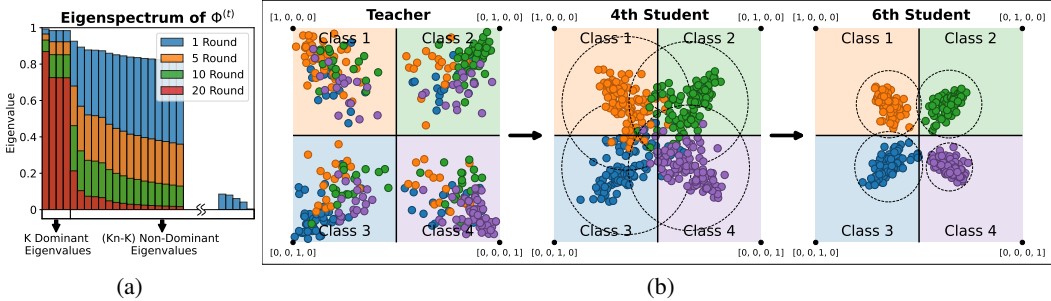

Figure 2: (a) The evolution of the eigenvalues of $\mathbf{\Phi}^{(t)}$ as the distillation rounds $t$ increases. Notably, only $K(=4)$ top-eigenvalues dominate the others as $t$ progresses. (b) A 2D plot showing the softmax outputs of the teacher model and the 4th and 6th student models for the training instances. Each corner represents the one-hot vectors of a class. As the distillation round $t$ increases, instances form distinct *clusters* based on their true labels (represented by colors in dots), driven by the label averaging among highly correlated input instances within the same class. See Appendix G for full evolution.

This condition requires that, for each class $k$, the proportion of samples correctly labeled as $k$ is greater than the proportion mislabeled as any other class $k'$. Under this mild condition, our method effectively attains the label averaging benefits of multi-round distillation without the need for repeated distillations. See Section 5 for details. Full proofs of our theoretical results are available in Appendix.

## 3 LABEL AVERAGING EFFECT OF MULTI-ROUND SELF-DISTILLATION

We present a closed-form solution for the softmax outputs $\mathbf{Y}^{(t)} = [\mathbf{y}_1^{(t)}, \ldots, \mathbf{y}_{Kn}^{(t)}] \in \mathbb{R}^{K \times Kn}$ of the $t$-th student model, in terms of the Gram matrix $\mathbf{\Phi} \in \mathbb{R}^{Kn \times Kn}$, defined as $[\mathbf{\Phi}]_{i,j} := \langle \phi(\mathbf{x}_i), \phi(\mathbf{x}_j) \rangle$.

**Theorem 3.1.** *As the number of self-distillation rounds $t \in \mathbb{Z}^+$ increases, the output predictions for the training instances evolve as*

$$\mathbf{Y}^{(t)} - \boldsymbol{I}_{K \times Kn}/K = \left(\mathbf{Y}^{(0)} - \boldsymbol{I}_{K \times Kn}/K\right)\left[\mathbf{I}_{Kn} - \left(\mathbf{I}_{Kn} + \mathbf{\Phi}/(K^2 n\lambda)\right)^{-1}\right]^t. \quad (9)$$

This result shows that each round of self-distillation effectively averages the given labels (columns of $\mathbf{Y}^{(0)}$) by repeatedly multiplying $(\mathbf{I}_{Kn} - (\mathbf{I}_{Kn} + \mathbf{\Phi}/(K^2 n\lambda))^{-1})$. By applying Woodbury's matrix identity and using the eigen-decomposition of $\mathbf{\Phi} = \sum_{i=1}^{Kn} \lambda_i \mathbf{v}_i \mathbf{v}_i^\top$, we can express the right-hand side of (9) as in (6), using $\mathbf{\Phi}^{(t)} := \left[\mathbf{I}_{Kn} - \left(\mathbf{I}_{Kn} + \mathbf{\Phi}/(K^2 n\lambda)\right)^{-1}\right]^t = \sum_{i=1}^{Kn} \left(\frac{\lambda_i}{K^2 n\lambda + \lambda_i}\right)^t \mathbf{v}_i \mathbf{v}_i^\top$. The relationship between the Gram matrix $\mathbf{\Phi}$ and the label averaging matrix $\mathbf{\Phi}^{(t)}$ reveals the effect of multi-round self-distillation, summarized as follows:

- Clustering Based on Feature Correlation: The Gram matrix $\mathbf{\Phi}$, defined as $[\mathbf{\Phi}]_{ij} := \langle \phi(\mathbf{x}_i), \phi(\mathbf{x}_j) \rangle$, captures the pairwise feature correlations among the training instances. If instances from the same ground-truth class exhibit higher feature correlations than those from different classes–as assumed in (7)–the top $K$ eigenvectors of $\mathbf{\Phi}$ indicate clusters representing each class (See Appendix J for details). Since $\mathbf{\Phi}^{(t)}$ shares the same eigenvectors as $\mathbf{\Phi}$, the label averaging process via $\mathbf{\Phi}^{(t)}$ causes the predictions for each instance to be increasingly influenced by the average labels of other instances within the same ground-truth class as the number of distillation rounds $t$ increases.

- Dominance of Top Eigenvectors: The effect of multi-round self-distillation is reflected in the eigenspectrum of $\mathbf{\Phi}^{(t)}$, where the eigenvalues are $\lambda_i^{(t)} := (\lambda_i/(K^2 n\lambda + \lambda_i))^t$, for $i \in [Kn]$. As $t$ increases, these eigenvalues decay at different rates depending on $\lambda_i$. Specifically, the top $K$ eigenvalues (associated with the largest $\lambda_i$) decay more slowly than the others. In Fig. 2a, we illustrate how the eigenvalues of $\mathbf{\Phi}^{(t)}$ evolve with increasing $t$. We use a training dataset with $K = 4$ classes and $n = 100$ samples per class, where the Gram matrix follows a block-wise structure specified in (7) with parameters $c = 0.4$ and $d = 0.1$ for $R = 1$, and is perturbed by a random matrix $\boldsymbol{E}$ with entries $[\boldsymbol{E}]_{i,j} \sim U(-0.05, 0.05)$. As the top-$K$ eigenvalues increasingly

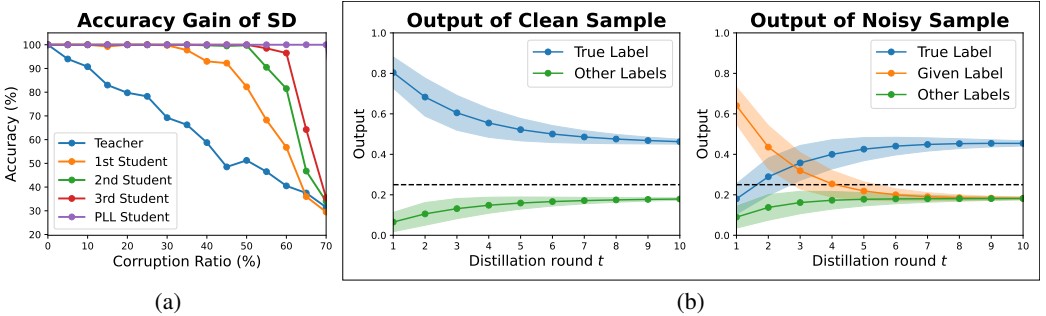

Figure 3: (a) Prediction accuracy of the teacher and student models for different distillation rounds $t$ as the label corruption rate increases. The 100% population accuracy regime expands for student models with more distillation rounds. (b) Softmax outputs for $K = 4$ classification as $t$ increases, using the same setup as Fig. 2 with a 50% label corruption rate. The BLUE line shows the average softmax value at the true label, the ORANGE line for noisy samples at the given label, and the GREEN line for other labels. Shaded regions show $\pm 1$ standard deviation. For clean samples, the true label maintains the highest value, while for noisy samples, the true label starts lower than the given label but surpasses it after a few rounds. The shaded region shrinks with $t$ due to label clustering.

dominate, the model's predictions become more clustered among instances of the same ground-truth class, as shown in Fig. 2b. This clustering effect persists as long as the top-$K$ eigenvectors of $\mathbf{\Phi}$ align with the ground-truth classes. The robustness of this alignment, even under perturbations from $\mathbf{E}$, is ensured provided that the perturbations are small relative to the eigen-gap $(\lambda_K - \lambda_{K+1})$, as established by the Davis-Kahan theorem (Davis & Kahan, 1970).

## 4 GAIN OF MULTI-ROUND SELF-DISTILLATION IN LABEL-NOISE SCENARIOS

In this section, we quantify the gains of multi-round self-distillation by identifying conditions under which the $t$-th student model can achieve 100% accuracy, even when trained on noisy dataset.

**Theorem 4.1.** *Suppose we have access to the population, i.e., $n \to \infty$. Let $\mathbf{C} \in \mathbb{R}^{K \times K}$ be the label corruption matrix of the training dataset, defined as $[\mathbf{C}]_{k,k'} := |\{i : y_i = k, \hat{y}_i = k'\}|/n$. We assume that $[\mathbf{C}]_{k,k'} = 0$ whenever $h(k) \neq h(k')$, where $h(k)$ denotes the superclass of class $k$. Then, the $t$-th distilled model, trained on this label-noisy dataset with Gram matrix $\mathbf{\Phi}$ as described in (7), achieves 100% population accuracy if*

$$[\mathbf{C}]_{k,k} > [\mathbf{C}]_{k,k'} + 1/\left((\tilde{\lambda}_K/\tilde{\lambda}_{K+1})^t - 1\right), \quad \forall k, k' \in [K] \text{ with } k \neq k' \tag{10}$$

*where $\tilde{\lambda}_K = \lambda_K/(K^2 n\lambda + \lambda_K)$ and $\tilde{\lambda}_{K+1} = \lambda_{K+1}/(K^2 n\lambda + \lambda_{K+1})$, with $\lambda_K = 1 - c + n(c - d)$ and $\lambda_{K+1} = 1 - c$. Conversely, if there exist $k, k' \in [K]$ with $k \neq k'$ such that*

$$[\mathbf{C}]_{k,k} < [\mathbf{C}]_{k,k'} + 1/\left((\tilde{\lambda}_K/\tilde{\lambda}_{K+1})^{t-1} - 1\right), \tag{11}$$

*then the $(t-1)$-th distilled model fails to correctly classify label-noisy samples from class $k$ that have been mislabelled as $\hat{y}_i = k'$, and thus fails to achieve 100% population accuracy.*

These conditions illustrate the advantages of multi-round self-distillation. Specifically, when,

$$[\mathbf{C}]_{k,k'} + 1/\left((\tilde{\lambda}_K/\tilde{\lambda}_{K+1})^t - 1\right) < [\mathbf{C}]_{k,k} < [\mathbf{C}]_{k,k'} + 1/\left((\tilde{\lambda}_K/\tilde{\lambda}_{K+1})^{t-1} - 1\right) \tag{12}$$

for some $k, k' \in [K]$ with $k \neq k'$, the $t$-th distilled model can correctly classify label-noisy samples from the class $k$ that have been mislabelled as $\hat{y}_i = k'$, while the $(t-1)$-th distilled model cannot. This demonstrates that multi-round self-distillation, through the label averaging effect described in Sec. 3, enhances the model's robustness to label noise, allowing it to tolerate higher rates of label corruption while still achieving 100% population accuracy. This phenomenon is illustrated in Fig. 3a, where we compare the classification accuracy of the teacher and student models across increasing rounds of distillation ($t$), as the label corruption rate increases, using the same experimental setup as

Fig. 2. As the distillation round $t$ increases, the range of label corruption for which the student model maintains 100% accuracy expands, as the condition (62) becomes easier to satisfy. In Fig. 3b, we illustrate how the softmax output components $[\mathbf{y}^{(t)}]_k$, for $k \in [K]$, evolves for clean samples (left) and label-noisy samples (right) as the distillation round $t$ increases, with a 50% label corruption rate. For label-noisy samples, while the teacher model ($t = 1$) tends to assign the highest softmax value to the incorrect noisy label due to memorization, after a few rounds of distillation ($t \geq 3$), the true label begins to attain the highest value in the softmax output, thus improving classification performance.

*Proof Sketch of Theorem 4.1.* The proof relies on analyzing the evolution of the model's outputs over multiple distillation rounds under the Gram matrix model (7). The key result is Lemma 4.1.

**Lemma 4.1.** *Under the Gram matrix model* (7)*, the output of the $t$-th distilled model for the training instance $(\phi(\mathbf{x}_i), \hat{y}_i)$ with true class $y_i$ is given by*

$$\mathbf{y}_i^{(t)} = p^t \mathbf{e}(\hat{y}_i) + (q^t - p^t)\bar{\mathbf{e}}_{y_i} + (r_s^t - q^t)\bar{\mathbf{e}}_{h(y_i)} + (1 - r_s^t)\mathbf{1}_K/K \tag{13}$$

*where $p, q, r_s$ are defined as*

$$p := (1-c)/(K^2 n\lambda + 1 - c); \quad q := (1 - c + n(c-d))/(K^2 n\lambda + 1 - c + n(c-d));$$
$$r_s := (1 - c + n(c-d) + K_s nd))/(K^2 n\lambda + 1 - c + n(c-d) + K_s nd), \tag{14}$$

*with $K_s$ being the number of classes in superclass $s = h(y_i)$, $\bar{\mathbf{e}}_{y_i} = \frac{1}{n}\sum_{\{j:y_j=y_i\}} \mathbf{e}(\hat{y}_j)$ is the average label vector over samples of the same true class $y_i$, $\bar{\mathbf{e}}_{h(y_i)} = \frac{1}{K_s n}\sum_{\{j:h(y_j)=s\}} \mathbf{e}(\hat{y}_j)$ is the average label vector over samples in the same superclass $h(y_i)$, and $\mathbf{1}_K$ is the all-ones vector in $\mathbb{R}^K$.*

This expression shows that the model's output $\mathbf{y}_i^{(t)}$ is a weighted combination of the given label $\mathbf{e}(\hat{y}_i)$, the average label vector $\bar{\mathbf{e}}_{y_i}$ over samples from the same true class $y_i$, the average label vector $\bar{\mathbf{e}}_{h(y_i)}$ over samples from the same superclass $h(y_i)$, and the uniform vector. Since $1 > r_s > q > p > 0$, as the distillation round $t$ increases, the weights assigned to these components shift away from the individual noisy label $\mathbf{e}(\hat{y}_i)$ towards the average labels of the same class and superclass, eventually making the output converge to the uniform vector as $t \to \infty$. Theorem 4.1 can be proved by analyzing (13) and finding the condition under which the correct label $y_i$ attains the highest value in the softmax output components $[\mathbf{y}_i^{(t)}]_k$, $k \in [K]$, for both clean ($\hat{y}_i = y_i$) and noisy ($\hat{y}_i \neq y_i$) samples. □

## 5 A NEW SELF-DISTILLATION METHOD WITH PARTIAL LABELS

In this section, we propose a novel self-distillation method that achieves the benefits of multi-round self-distillation, as outlined in Thm. 4.1, in just a single-round. The key insight stems from the observation that, under mild conditions on the label corruption rates, the teacher's softmax output at the true label (represented by the blue lines in Fig. 3b at $t = 1$) consistently ranks at least second-highest–even for label-noisy samples–even if it is lower than the given label. As self-distillation progresses through multiple rounds, the softmax output (or confidence) for the true label decreases in clean samples while increasing in noisy samples due to the label-averaging effect among instances from the same ground-truth class. This phenomenon improves classification accuracy for noisy samples but reduces the model's confidence in clean samples. This trend is illustrated in Fig. 3b, where the blue lines for clean (left) and noisy (right) samples converge to similar levels as $t$ increases.

Motivated by this observation, we aim to design a self-distillation method where, in a single round, the true label achieves the highest output value, even for noisy samples. To achieve this, we introduce the following label refinement scheme for the teacher's output: We generate a two-hot vector $\bar{\mathbf{y}}_i$ from the teacher's output $\mathbf{y}_i^{(T)} = \sigma(\boldsymbol{\theta}_T^\top \phi(\mathbf{x}_i))$ by selecting the top two labels with the highest values and assigning a weight of 1/2 to each, setting all the other entries to zero. These refined outputs become the new targets for training the student model $\boldsymbol{\theta}$, trained with the objective $f_P(\boldsymbol{\theta}) = \frac{1}{Kn}\sum_{i=1}^{Kn} \mathsf{CE}(\bar{\mathbf{y}}_i, \mathbf{y}_i^{(P)}) + \frac{\lambda\|\boldsymbol{\theta}\|_F^2}{2}$ for $\mathbf{y}_i^{(P)} = \sigma(\boldsymbol{\theta}^\top \phi(\mathbf{x}))$. We refer to this student model as the Partial Label Learning (PLL) student model, following the concept of PLL (Cour et al., 2011), which trains a classifier using a candidate set of labels for each input instance. Our label refinement emulates the effect of multi-round self-distillation by significantly boosting the confidence (assigning weight 1/2) to the true label for noisy samples while reducing it for clean samples. As a result, the PLL student model achieves 100% population accuracy after a single round of distillation.

**Theorem 5.1.** *The PLL student model, trained on the label-noisy dataset with the Gram matrix $\mathbf{\Phi}$ from* (7)*, achieves 100% population accuracy if*

$$[\mathbf{C}]_{k,k} > [\mathbf{C}]_{k,k'}, \quad \forall k, k' \in [K] \text{ with } k \neq k', \tag{15}$$

*where the label-corruption matrix $\mathbf{C}$ is defined by $[\mathbf{C}]_{k,k'} := |\{i : y_i = k, \hat{y}_i = k'\}|/n$, and we assume that $[\mathbf{C}]_{k,k'} = 0$ whenever $h(k) \neq h(k')$.*

*Proof Sketch.* The condition (15) ensures that, for each class $k$, the proportion of samples correctly labeled as $k$ is greater than the proportion mislabeled as any other class $k' \neq k$. Under this condition, the teacher's softmax output includes the true label among the top two highest values for both clean and label-noisy samples. By using the refined partial labels $\bar{\mathbf{y}}_i$, a single round of self-distillation generates an output for the $i$-th sample similar to (13):

$$\mathbf{y}_i^{(P)} = p\bar{\mathbf{y}}_i + (q - p)\left(\frac{1}{n}\sum_{\{j:y_j = y_i\}}\bar{\mathbf{y}}_j\right) + (r_s - q)\left(\frac{1}{K_s n}\sum_{\{j:h(y_j) = s\}}\bar{\mathbf{y}}_j\right) + (1 - r_s)\frac{\mathbf{1}_K}{K}. \tag{16}$$

This expression ensures that the true label $y_i$ has the highest value in $\mathbf{y}_i^{(P)}$, i.e., $[\mathbf{y}_i^{(P)}]_{y_i} > [\mathbf{y}_i^{(P)}]_k$ for all $k \neq y_i$, as each partial label $\bar{\mathbf{y}}_i$ assigns a weight of 1/2 to the true label. $\square$

**Remark 1** (Comparison with Multi-Round Self-Distillation). The condition (62) for achieving 100% accuracy with the $t$-th self-distilled model is more stringent than the condition (15) for the PLL student model. Therefore, the PLL student model not only replicates the effect of multi-round distillation in a single round but also achieves better performance in scenarios with high label-noise rates, where (15) is satisfied but (62) is not. This advantage is illustrated in Fig. 3a, where the PLL student model achieves 100% population accuracy across a wider range of label corruption rates.

## 6 EXPERIMENT

In this section, we validate our theoretical findings through experiments on real datasets. We employ a two-layer neural network composed of a linear layer followed by a softmax layer on top of a fixed feature extractor. We conduct experiments on six multi-class image classification benchmarks: CIFAR-100 (Krizhevsky et al., 2009), Caltech-101/256 (Griffin et al., 2007), Flowers-102 (Nilsback & Zisserman, 2008), Food-101 (Bossard et al., 2014), and StanfordCars (Krause et al., 2013), utilizing the PyTorch torchvision library. For our fixed feature extractor, we use ResNet34 (He et al., 2016) networks pre-trained on ImageNet. More experimental details can be found in Appendix E. We explore three different label corruption scenarios: *symmetric*, *asymmetric*, and *superclass* corruption, as described in Table 1, for a label corruption rate $\eta \in [0, 1]$ and $\Omega_k := \{l \in [K] : h(l) = h(k)\}$.

Table 1: Definition of label corruption matrix $\mathbf{C}$ for different corruption scenarios

| Symmetric corruption | Asymmetric corruption | Superclass corruption |
|---|---|---|
| $[\mathbf{C}]_{k,k'} := \begin{cases} 1 - \eta, & k' = k, \\ \frac{\eta}{K-1}, & k' \neq k. \end{cases}$ | $[\mathbf{C}]_{k,k'} := \begin{cases} 1 - \eta, & k' = k, \\ \frac{2\eta}{K}, & k' = \tilde{k}(k), \\ \frac{\eta}{K}, & k' \neq k, \tilde{k}(k). \end{cases}$ | $[\mathbf{C}]_{k,k'} := \begin{cases} 1 - \eta, & k' = k, \\ \frac{\eta}{|\Omega_k|-1}, & k' \neq k, k' \in \Omega_k, \\ 0, & k' \neq \Omega_k, \end{cases}$ |

Fig. 4 illustrates the accuracy improvement of student models compared to the base (teacher) model in the superclass corruption scenario. Results for other datasets and corruption scenarios are available in Appendix F. Across all datasets, we observe that test accuracy progressively increases with the number of distillation rounds. Notably, the PLL student model outperforms other multi-round distillation models at higher corruption rates ($\eta \geq 0.6$), achieving this gain efficiently in just a single round of distillation. These results support our theoretical findings, showing the effectiveness of multi-round self-distillation–and especially the PLL student model–in mitigating the effects of label noise.

**Verification of the Label Averaging Effect** To verify the occurrence of label averaging effect by self-distillation in real datasets, we plot the evolution of model outputs across distillation steps. We arbitrarily select four classes from the CIFAR-100 dataset and conduct experiments only on samples from these classes. Under 50% symmetric label corruption, we visualize the output evolution of

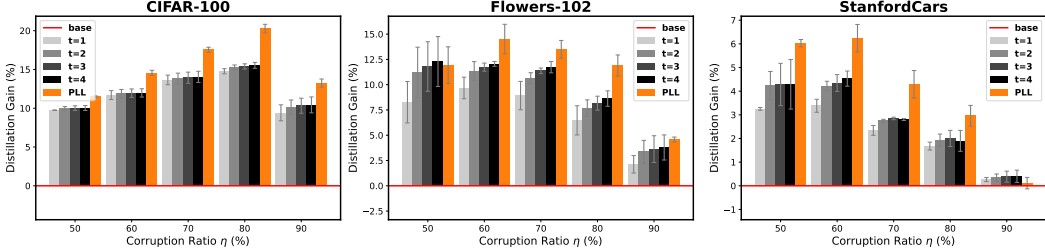

Figure 4: Distillation gain in accuracy (%) by each student model compared to the teacher in the superclass label corruption scenario. PLAIN represents the improvement of the multi-round students over the teacher model, while ORANGE represents the improvement by the PLL student model.

both the teacher and student models in a 2D plot in Fig. 5a. As the number of distillation rounds $t$ increases, we observe that the outputs tend to form clusters based on their true labels. Additionally, some instances initially misclassified by the teacher are correctly classified by the student model due to the label averaging effect. This suggests that the distillation process helps the student model recover from some of the teacher's errors, especially in cases where the labels are corrupted.

**Ablation Study of Top-$k$ Targets**   In the PLL student model, using a larger set of partial labels increases the likelihood of including the true label but introduces a trade-off by reducing the confidence in instances that are already well-classified. Our theoretical analysis in Sec. 5 indicates that the true label's prediction is at least the second highest in the teacher's output, supporting the use of a top-2 candidate set. To validate this choice, we conduct experiments under the same conditions as Fig. 4, using the top-$k$ partial labels as targets for various $k$. In Fig. 5b, we plot the accuracy gain of each PLL student model with top-$k$ partial labels, chosen from the teacher's output, on the CIFAR-100 dataset. Results for other datasets are provided in Appendix H.1. As shown, using more candidates generally leads to lower test accuracy across most datasets, supporting our choice of top two labels.

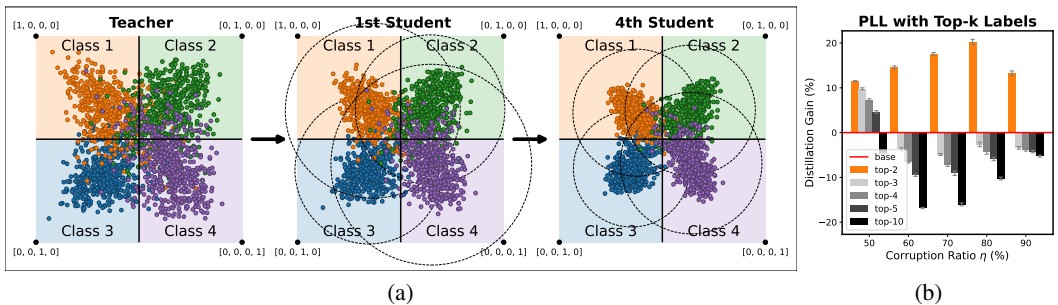

Figure 5: (a) Visualization of the softmax outputs of the teacher model, and the first and fourth student models on the CIFAR-100 dataset, reduced to four-class classification outputs for clarity. (b) Accuracy improvement (%) for each PLL student model using top-$k$ partial labels compared to the teacher model in the superclass label corruption scenario on the CIFAR-100 dataset.

## 7  CONCLUSION

We investigated self-distillation for multi-class classification in the context of linear probing with fixed feature extractors. Our theoretical analysis showed that multi-round self-distillation effectively performs label averaging among instances with high feature correlations, which helps correct label noise by reducing the model's reliance on potentially corrupted labels. Additionally, we introduced the PLL student model as an efficient approach for label refinement, offering a practical method to distill the teacher's "dark knowledge" into a candidate set of labels. This method achieves the benefits of multi-round self-distillation in just a single round, providing a practical and efficient way to improve model performance. Limitations and future works are available in Appendix B.

ACKNOWLEDGEMENT

This work was supported by the National Research Foundation of Korea (NRF) grant funded by the Korea government (MSIT) (No. RS-2024-00408003 and RS-2025-00516153), and by the Institute for Information & communications Technology Planning & Evaluation (IITP) grants funded by the Korean government (MSIT) (RS-2024-00444862).

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

## A   BROADER IMPACT

Our paper theoretically analyzes self-distillation, particularly highlighting its advantages in multi-class classification problems with label noise. We also explore the integration of self-distillation with partial label learning, demonstrating its superior performance gains over traditional multi-round self-distillation, especially in high noise rate regimes. This research directly addresses the prevalent challenges in handling real-world data, where acquiring clean and accurately labeled datasets is often difficult and costly.

With the recent breakthroughs in large-scale pre-trained neural networks or foundation models, various studies have been conducted to effectively utilize well-trained large neural networks as feature extractor backbones. We believe that our result can be applied to such scenarios and theoretically demonstrate the performance gains achievable by self-distillation through a simple linear probing within a pre-trained neural network.

By enhancing our understanding of knowledge distillation and self-distillation, we also provide the way for the development of more efficient models, capable of matching the performance of their larger and more complex counterparts, which is especially beneficial in scenarios with limited resources.

## B   LIMITATIONS AND FUTURE WORKS

In this work, we theoretically analyzed the effect of multi-round self-distillation and self-distillation with refined outputs by deriving closed-form solutions for the outputs of the student model as the number of distillation rounds increases (Sec. 3). The main technical challenge in this analysis lies in the non-linearity of softmax functions. We addressed this by using a linear approximation of the softmax function, similar to the original work by Hinton et al. (2015), as detailed in Assumption 1. We justified this assumption within proper parameter regimes through numerical experiments in Sec. D.1. Generalizing our results to fully incorporate the non-linear nature of the softmax function remains a future work.

After deriving closed-form solutions for the outputs of the distilled models in Theorem 3.1 in terms of the general Gram matrix $\mathbf{\Phi}$, representing instance-wise feature correlations, we considered a particular low-rank feature correlation map (7) to conduct further analysis on the student models' outputs. The practicality of this feature correlation model is validated using six different real datasets, as detailed in Sec. D.2, demonstrating its applicability in real-world model training. We also provided additional analysis for four additional feature correlation maps in Sec. J, including a more generalized version of the superclass feature correlation model. Relaxing the assumption on a class-wise feature correlation, for example by generalizing it to

$$\mathbb{E}[\langle \phi(\mathbf{x}_i), \phi(\mathbf{x}_j) \rangle | (y_i, y_j) = (a, b)] = \omega(a, b),$$

remains a future work.

Lastly, our main analysis and experimental results show the gain of self-distillation in training a linear classifier on top of a fixed feature extractor. Our results can also be combined with the feature learning perspective of self-distillation by considering the feature map evolving over the distillation rounds, as illustrated in Corollary C.1. Understanding the role of self-distillation in more general setups, including distilling foundational models with more diverse applications, such as natural language processing, is an important direction for future work.

## C  SELF-DISTILLATION WITH FEATURE LEARNING

We next provide an extended version of our analysis for the full-to-full self-distillation scenarios, where not only the last linear layer but also the feature extractor can be updated through multi-round self-distillations. To illustrate this point, consider the quantified gains of self-distillation in label-noise scenarios presented in Theorem 4.1. This theorem provides the sufficient number of distillation rounds required for the student's softmax output to assign the highest value to the ground-truth label for both clean and noisy samples. The condition depends not only on the class-wise label corruption rates but also on the relative gap in feature correlations between samples of the same ground-truth class, parameterized by $c$, and those of different ground-truth classes, parameterized by $d$, in the Gram matrix model $\mathbf{\Phi}$ (7).

We can generalize our theory by allowing these correlations to evolve over distillation rounds, denoted by $c^{(i)}$ and $d^{(i)}$, due to feature updates during self-distillation. Recent work by Allen-Zhu & Li (2022) demonstrates that self-distillation's effectiveness arises from an implicit ensemble of teacher and student models, enabling the student to learn more diverse features when using the teacher's softmax outputs as targets. This suggests that the intra-class feature correlation $c^{(i)}$ may increase with each distillation round $i$, enhancing the separation between classes.

Assuming the class-wise feature map defined in (7), and allowing $c$ and $d$ to change over distillation steps, our parameters $p$ and $q$ in (14), which govern the label averaging effect, also become functions of $i$:

$$p^{(i)} := \frac{1 - c^{(i)}}{K^2 n\lambda + 1 - c^{(i)}}$$
$$q^{(i)} := \frac{1 - c^{(i)} + n(c^{(i)} - d^{(i)})}{K^2 n\lambda + 1 - c^{(i)} + n(c^{(i)} - d^{(i)})}. \tag{17}$$

Under the extended class-wise feature correlation assumption, our Theorem 4.1 can be generalized to:

**Corollary C.1** (Extended version). *Under the evolving feature correlation model, the $t$-th distilled model achieves 100% population accuracy if*

$$[\mathbf{C}]_{k,k} > [\mathbf{C}]_{k,k'} + \frac{1}{\prod_{i=1}^{t}(q^{(i)}/p^{(i)}) - 1}, \quad \forall k, k'(\neq k) \in [K].$$

If the student model learns more diverse features than the teacher model, leading to an increased intra-class feature correlation $c^{(i)}$ as $i$ increases, then $q^{(i)}/p^{(i)}$ also increases. This means that the regime for a student model to achieve 100% population accuracy expands, ultimately leading to performance gains from self-distillation. Thus, our result can be integrated with the aspect of Allen-Zhu & Li (2022), where the gain from self-distillation is explained by diverse features learned by the student model. Specifically, Corollary C.1 quantifies the gains both from the feature learning and label averaging effects of self-distillation under label noise scenarios. The proof of Corollary C.1 can be found in Appendix K.3.

## D  JUSTIFICATION OF ASSUMPTIONS

In this section, we provide empirical justifications for the assumptions we have imposed in the theoretical analysis.

### D.1  JUSTIFICATION OF SOFTMAX APPROXIMATION (ASSUMPTION 1)

In Assumption 1, we approximate the softmax function $\sigma(\mathbf{v})$ to the linear function $1/K + \mathbf{v}/K$, assuming a small enough magnitude of $\mathbf{v} \in \mathbb{R}^K$. To justify our approximation, let us recall the equation for $\mathbf{y}_i^{(t)}$:

$$\mathbf{y}_i^{(t)} = \sigma\left(\sum_{j=1}^{Kn}\langle\phi(\mathbf{x}_i), \phi(\mathbf{x}_j)\rangle\boldsymbol{\alpha}_j^{(t)}\right) = \sigma\left(\sum_{j=1}^{Kn}\frac{\langle\phi(\mathbf{x}_i), \phi(\mathbf{x}_j)\rangle}{Kn\lambda}\left(\mathbf{y}_j^{(t-1)} - \mathbf{y}_j^{(t)}\right)\right). \tag{18}$$

We numerically validate Assumption 1, by analyzing the softmax approximation error. The softmax approximation error is defined as the maximum difference ($\ell_\infty$-norm) between the softmax outputs derived in Lemma 4.1, based on the linear approximation of the softmax, and the outputs obtained by numerically solving (18) for $\{\mathbf{y}_i^{(t)}\}_{i=1}^{Kn}$. (see Appendix E.4 for details.)

We set $K = 4$, $\lambda = 10^{-4}$, and the Gram matrix $\mathbf{\Phi}$ follows a block-wise structure as in (7) with parameters $c = 0.4$ and $d = 0.1$. Under the symmetric label corruption with rate $\eta = 0.5$, we analyze the softmax error as the number of samples per class $n$ increases. As shown in Fig. 6, the softmax error becomes negligible, especially as the number of samples exceeds $10^4$.

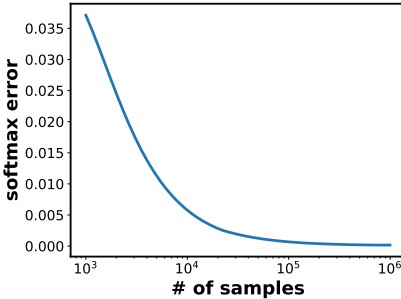

Figure 6: We plot the softmax approximation error as the number of samples increases. We use $K = 4$, $\lambda = 10^{-4}$, $c = 0.4$, $d = 0.1$, and impose the symmetric label corruption with rate $\eta = 0.5$.

## D.2 JUSTIFICATION OF CLASS-WISE FEATURE CORRELATION MAP

In (7), we argue that the feature correlation matrix exhibits a block diagonal structure, i.e.,

$$\langle \phi(\mathbf{x}_i), \phi(\mathbf{x}_j) \rangle = \begin{cases} 1 & i = j; \\ c, & y_i = y_j, i \neq j; \\ d, & y_i \neq y_j, h(y_i) = h(y_j); \\ 0, & h(y_i) \neq h(y_j). \end{cases} \tag{19}$$

We empirically demonstrate the validity of this claim on diverse real-world datasets (with dataset details in Appendix E.1). We first conduct hierarchical clustering to rearrange the order of classes so that the highly-correlated classes become adjacent. The results of hierarchical clustering are presented in Appendix L.2. Under the rearranged class orders, the Gram matrices (input feature correlation matrices) have a block-diagonal matrix structure for the real datasets, as shown in Fig. 7.

Additionally, we assume that feature correlation between any two instances is determined by their corresponding true labels. To verify this assumption, we investigate the mean and standard deviation values of the feature correlation for all pairs of instances, for three cases as follows: 1) $y_i = y_j$, 2) $y_i \neq y_j, h(y_i) = h(y_j)$, and 3) $h(y_i) \neq h(y_j)$. We calculate the mean and standard deviation for all sample pairs corresponding to each case. As shown in Table 2, we find that there exists a gap in the mean feature correlations between the three cases. This empirical evidence supports the validity of our class-wise feature map, modeled as (7).

Table 2: Statistics for three different cases of instance-wise feature correlation.

| Dataset | # of superclass | $y_i = y_j$ | $y_i \neq y_j, h(y_i) = h(y_j)$ | $h(y_i) \neq h(y_j)$ |
|---------|-----------------|-------------|---------------------------------|----------------------|
| CIFAR-100 | 3 | 0.25±0.16 | 0.02±0.12 | -0.03±0.10 |
| Caltech-101 | 4 | 0.42±0.15 | 0.03±0.08 | -0.02±0.07 |
| Caltech-256 | 3 | 0.35±0.16 | 0.02±0.08 | -0.03±0.07 |
| Flowers-102 | 2 | 0.36±0.15 | 0.03±0.12 | -0.04±0.13 |
| Food-101 | 2 | 0.17±0.15 | 0.01±0.12 | -0.02±0.12 |
| StanfordCars | 2 | 0.21±0.16 | 0.06±0.14 | -0.07±0.13 |

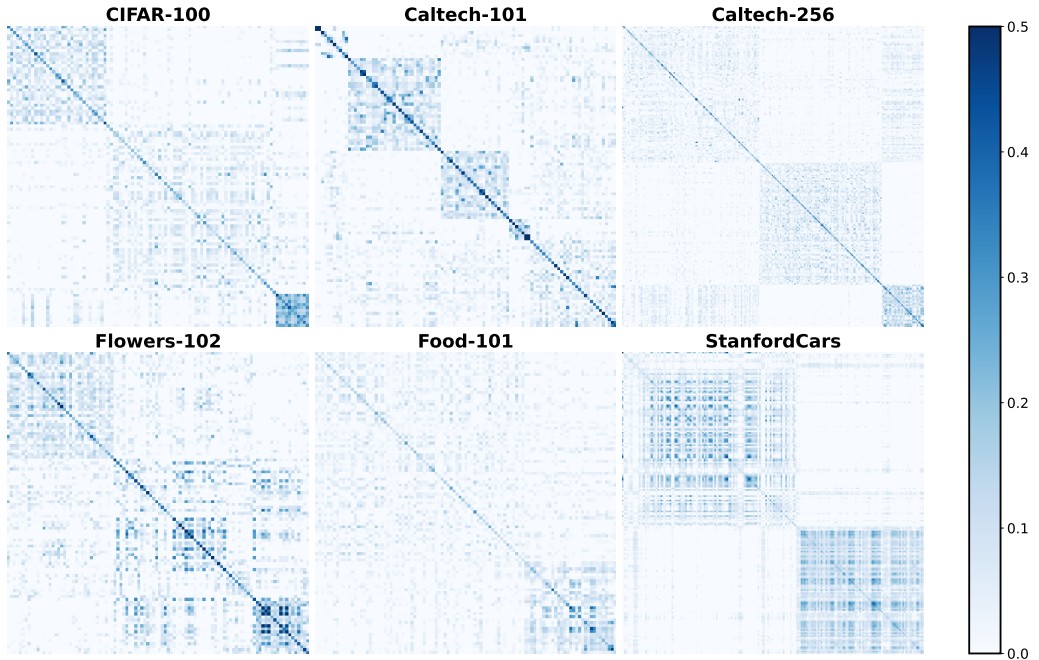

Figure 7: Class-wise feature correlation map of six different real-world datasets. The features are extracted by a pre-trained ResNet34. Block diagonal patterns are exhibited in all the six datasets, as assumed in our feature correlation map (7).

## D.3 EFFECT OF TEMPERATURE SCALING

Let us define the softmax $\sigma^\tau : \mathbb{R}^K \to (0, 1)^K$ with a temperature scaling parameter $\tau$ as $\sigma^\tau(\mathbf{v}) = \left[ \frac{\exp(v_1/\tau)}{\sum_{i=1}^K \exp(v_i/\tau)}, \ldots, \frac{\exp(v_K/\tau)}{\sum_{i=1}^K \exp(v_i/\tau)} \right]$ for the logits $\mathbf{v} = [v_1, \ldots, v_K] \in \mathbb{R}^K$. We set $\tau = 1$ and denote $\sigma^1(\cdot) = \sigma(\cdot)$ in the main document. In this section, we show that the temperature scaling of the softmax, combined with the proper scaling of the cross-entropy (CE) loss function in the regularized CE loss (1), results in the same optimal solution for $\boldsymbol{\theta}$ as in $\tau = 1$. Let $\mathbf{y}_i^{(t)}$ represent the output of the $t$-th student model for the $i$-th sample. The objective function for the $t$-th student model $f^{(t)}(\boldsymbol{\theta})$, scaled by temperature $\tau$ is given by

$$f^{(t)}(\boldsymbol{\theta}) = \frac{\tau^2}{Kn} \sum_{i=1}^{Kn} \mathsf{CE}(\mathbf{y}_i^{(t-1)}, \sigma^\tau(\boldsymbol{\theta}^\top \phi(\mathbf{x}_i))) + \frac{\lambda \|\boldsymbol{\theta}\|_F^2}{2}, \tag{20}$$

for $t \in \mathbb{N}$. Then, the optimal $\boldsymbol{\theta}$, denoted as $\boldsymbol{\theta}^*$ satisfies

$$\boldsymbol{\theta}^{*\top} = \tau \sum_{i=1}^{Kn} \underbrace{\frac{1}{Kn\lambda} \left( \mathbf{y}_i^{(t-1)} - \mathbf{y}_i^{(t)} \right)}_{:=\boldsymbol{\alpha}_i^{(t)}} \phi(\mathbf{x}_i)^\top = \tau \sum_{i=1}^{Kn} \boldsymbol{\alpha}_i^{(t)} \phi(\mathbf{x}_i)^\top. \tag{21}$$

Substituting the optimal $\boldsymbol{\theta}^*$ yields

$$\mathbf{y}_i^{(t)} = \sigma^\tau \left( \boldsymbol{\theta}^{*\top} \phi(\mathbf{x}_i) \right)$$

$$= \sigma^\tau \left( \tau \sum_{j=1}^{Kn} \langle \phi(\mathbf{x}_i), \phi(\mathbf{x}_j) \rangle \boldsymbol{\alpha}_j^{(t)} \right)$$

$$= \sigma^1 \left( \sum_{j=1}^{Kn} \frac{\langle \phi(\mathbf{x}_i), \phi(\mathbf{x}_j) \rangle}{Kn\lambda} \left( \mathbf{y}_j^{(t-1)} - \mathbf{y}_j^{(t)} \right) \right).$$

This implies that even with temperature scaling with $\tau$, the equation of the optimal prediction $\mathbf{y}_i^{(t)}$ remains unchanged for $f^{(t)}(\boldsymbol{\theta})$. In other words, temperature scaling does not affect the optimal prediction.

# E  EXPERIMENTAL DETAILS

In this section, we provide the details of the experiments presented in Sec. 6. Our code is publicly available at https://github.com/Hyeonsu-Jeong/SelfPLL.

## E.1  IMAGE CLASSIFICATION DATASETS

To validate our theoretical findings, we conduct experiments on several real-world datasets using the PyTorch torchvision library. Below are the details of the real-world datasets we used.

- **CIFAR-100** (Krizhevsky et al., 2009): CIFAR-100 consists of 60,000 32x32 images in 100 classes. This dataset covers a wide range of object categories such as animals, vehicles, and household items, with each class having 600 images.

- **Caltech-101** (Griffin et al., 2007): Caltech-101 contains images from 101 object categories with a background category named "BACKGROUND_Google". For each category, there are about 40 to 800 samples, with an average of 50 samples per category. We removed the background category and divided the dataset into training and validation sets using an 8:2 ratio.

- **Caltech-256** (Griffin et al., 2007): Caltech-256 is a large-scale dataset consisting of 30,607 images across 256 object categories. Similar to the Caltech-101 dataset, we removed the "clutter" category and divided the dataset into training and validation sets using an 8:2 ratio.

- **Flowers-102** (Nilsback & Zisserman, 2008): The Flowers-102 dataset contains 102 different categories of flowers, with a total of 8,189 images. Each class consists of varying numbers of images, ranging from 40 to 258 images per class. Since the test set of the Flowers-102 dataset is larger than the training set, we swapped the training and test sets for use.

- **Food-101** (Bossard et al., 2014): Food-101 is a large-scale dataset specifically designed for food recognition tasks. It contains 101,000 images of food items categorized into 101 classes, with each class consisting of 1,000 images.

- **StanfordCars** (Krause et al., 2013): The StanfordCars dataset is focused on car recognition and consists of 16,185 images of 196 classes of cars. The dataset includes images of cars taken from various viewpoints and under different lighting conditions.

## E.2  CHOOSING AN APPROPRIATE HYPERPARAMETER $\lambda$

In real experiments, we extract the features of input instances of the diverse datasets using the pre-trained ResNet34 model. Recall the definition of $p$ and $q$ from (14):

$$p := \frac{1-c}{K^2 n\lambda + 1 - c}, \tag{22}$$

$$q := \frac{1-c+n(c-d)}{K^2 n\lambda + 1 - c + n(c-d)}. \tag{23}$$

In this setting, we visualize the variation of the $q/p$ ratio as $\lambda$ changes across the six real datasets to find an appropriate $\lambda$ considering, the intra-class feature correlation $c$ and the inter-class feature correlation $d$ from Table 2. When the $q/p$ ratio is too small or too large, it becomes hard to demonstrate the effect of distillation from Thm. 4.1.

In Fig. 8, we observe that to achieve a $q/p$ ratio close to 2, $\lambda$ should be selected around $10^{-3}$ for Caltech-101, Flowers-102, and StanfordCars, while around $10^{-4}$ for CIFAR-100, Caltech-256, and Food-101. Therefore, in the main text, we set $\lambda$ to $5 \times 10^{-4}$ for Caltech-101, Flowers-102, and StanfordCars, and $5 \times 10^{-5}$ for CIFAR-100, Caltech-256, and Food-101 to illustrate the impact of distillation. Additional experimental results for various values of $\lambda$ are presented in Appendix L.1.

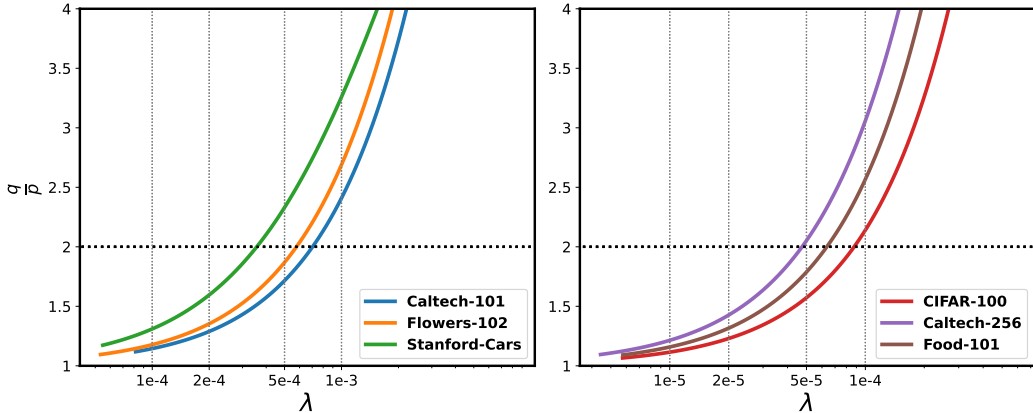

Figure 8: Variation of $q/p$ with changing weight decay $\lambda$.

### E.3   Training

We train a two-layer network consisting of a linear layer and a softmax layer. Our loss function is defined as the cross-entropy between the softmax output of the teacher model and the softmax output of the student model. To mitigate uncertainty, we conduct three repeated runs for all experiments with seeds 0, 1 and 2. We perform a grid search over learning rates, in the set $\{0.1, 0.05, 0.01, 0.005, 0.001\}$. Each model is trained for 200 epochs, employing the SGD optimizer with a momentum value of 0.9. In our experiments, we observe that using CE loss with the PLL student model often leads to instability during training. The PLL student model trains with a set of candidate labels for each sample with equal weights–in our case, the top two labels with weights of $1/2$ each. Using CE loss with equally weighted candidate labels can cause instability since the model may converge incorrectly when the candidate set includes incorrect labels. Hence, for the stable convergence of the PLL student model, we use Generalized Cross Entropy(GCE) (Zhang & Sabuncu, 2018) loss with the hyperparameter $q = 0.7$. Our neural networks are trained using multiple NVIDIA RTX A6000 GPUs.

### E.4   Numerical Method for Calculating the Outputs of Student Models

For synthetic experiments in Fig. 2–3, we compute the outputs of multi-round self-distillation models using numerical methods, instead of applying the approximation in Assumption 1. Recall from (3), the equation for $\boldsymbol{\theta}^{(t)}$:

$$\boldsymbol{\theta}^{(t)\top} = \sum_{i=1}^{Kn} \frac{\left(\mathbf{y}_i^{(t-1)} - \mathbf{y}_i^{(t)}\right)}{Kn\lambda} \phi(\mathbf{x}_i)^\top$$

The optimal output of the $t$-th disitlled model $\{\mathbf{y}_i^{(t)}\}_{i=1}^{Kn}$ satisfies the following equation:

$$\mathbf{y}_i^{(t)} = \sigma\left(\sum_{j=1}^{Kn} \frac{\langle\phi(\mathbf{x}_i), \phi(\mathbf{x}_j)\rangle}{Kn\lambda}\left(\mathbf{y}_j^{(t-1)} - \mathbf{y}_j^{(t)}\right)\right) \tag{24}$$

Therefore, given the feature correlations and the output of the $(t-1)$-th distilled student model $\{\mathbf{y}_i^{(t-1)}\}_{i=1}^{Kn}$, the optimal output for the $t$-th distilled student model $\{\mathbf{y}_i^{(t)}\}_{i=1}^{Kn}$ can be found by minimizing the difference between the left-hand side and right-hand side of (24). The corresponding pseudo-code for this process is outlined below:

---

**Algorithm 1:** Optimal Output Calculation by Numerical Method

---

**Input:** $K$: number of classes, $n$: number of samples per class, $\lambda$: weight-decay, $T$: total distillation steps, $\eta$: learning rate, $\epsilon$: convergence tolerance

**Input: Given label (target)** $\mathbf{Y}^{(0)} = [\mathbf{y}_1^{(0)}, \dots, \mathbf{y}_{Kn}^{(0)}] \in \mathbb{R}^{K \times Kn}$

**Output: Output of $t$-th distilled model** $\mathbf{Y}^{(t)} = [\mathbf{y}_1^{(t)}, \dots, \mathbf{y}_{Kn}^{(t)}] \in \mathbb{R}^{K \times Kn}$ for $t = 1, \dots, T$

1 **for** $t = 1, 2, \dots, T$ **do**
2      **Step 1: Initialize** $\mathbf{Y}^{(t)}$;
3      **for** *each instance* $i = 1, 2, \dots, Kn$ **do**
4          $\mathbf{y}_i^{(t)} = \frac{\mathbf{u}}{\sum_{k=1}^{Kn} u_k}, \quad u_k \sim U(0,1)$;

5      **while** *iter < max_iter* **do**
6          **Step 2: Compute** $\hat{\mathbf{Y}}^{(t)}$;
7          **for** *each instance* $i = 1, 2, \dots, Kn$ **do**
8              $\hat{\mathbf{y}}_i^{(t)} = \sigma\left(\sum_{j=1}^{Kn} \frac{\langle \phi(\mathbf{x}_i), \phi(\mathbf{x}_j)\rangle}{Kn\lambda} \left(\mathbf{y}_j^{(t-1)} - \mathbf{y}_j^{(t)}\right)\right)$;

9          **Step 3: Compute Loss** $\mathcal{L}$;
10          $\mathcal{L}(\mathbf{Y}^{(t)}) = \|\mathbf{Y}^{(t)} - \hat{\mathbf{Y}}^{(t)}\|$;

11          **Step 4: Update** $\mathbf{Y}^{(t)}$;
12          $\mathbf{Y}^{(t)} \leftarrow \mathbf{Y}^{(t)} - \eta \nabla_{\mathbf{Y}^{(t)}} \mathcal{L}(\mathbf{Y}^{(t)})$;

13          **Step 5: Check for Convergence**;
14          **if** $\mathcal{L}(\mathbf{Y}^{(t)}) < \epsilon$ **then**
15              **Break**;

---

# F FULL EXPERIMENTAL RESULTS ON THREE DIFFERENT LABEL CORRUPTION SCENARIOS

For our Theorems 4.1 and 5.1 in the main text, we formulate the gain of self-distillation using the label corruption matrix $\mathbf{C}$ without making any specific assumptions about the label corruption. To validate our theoretical results, we conduct experiments with various types of label corruption. Fig. 9 shows the accuracy gain of the student models for six different datasets under the three corruption scenarios, described in Table 1. The precise accuracies of each distillation models can be found in Appendix M. Overall, we observe that there is a gain as the distillation step $t$ increases in multi-round distillation, and we confirm that the PLL student model outperforms other multi-round distillation models. This results validate our theoretical finding in Sec. 4 and 5, where we show that self-distillation plays a role as label averaging, facilitating the correction of label noise.

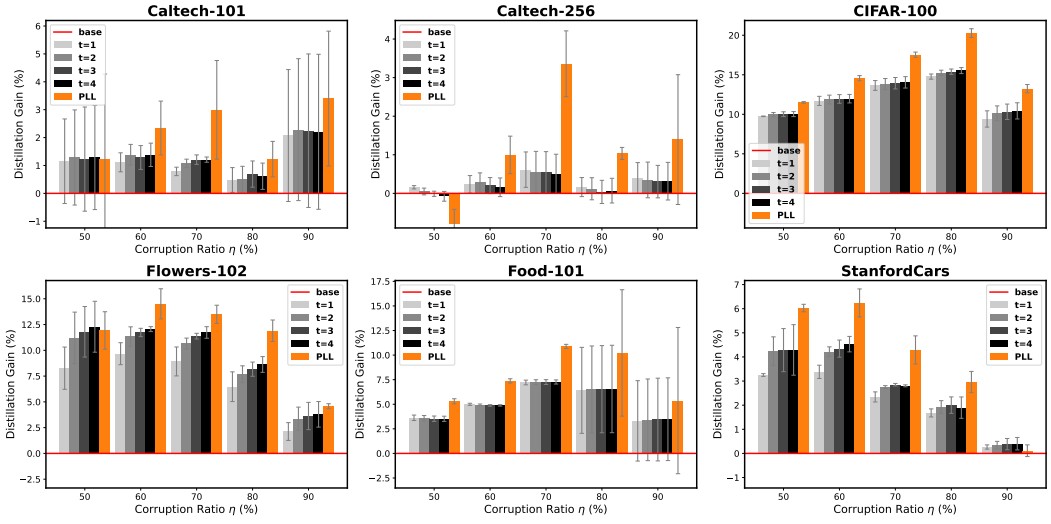

(a) Superclass label corruption (Full version of Fig. 4)

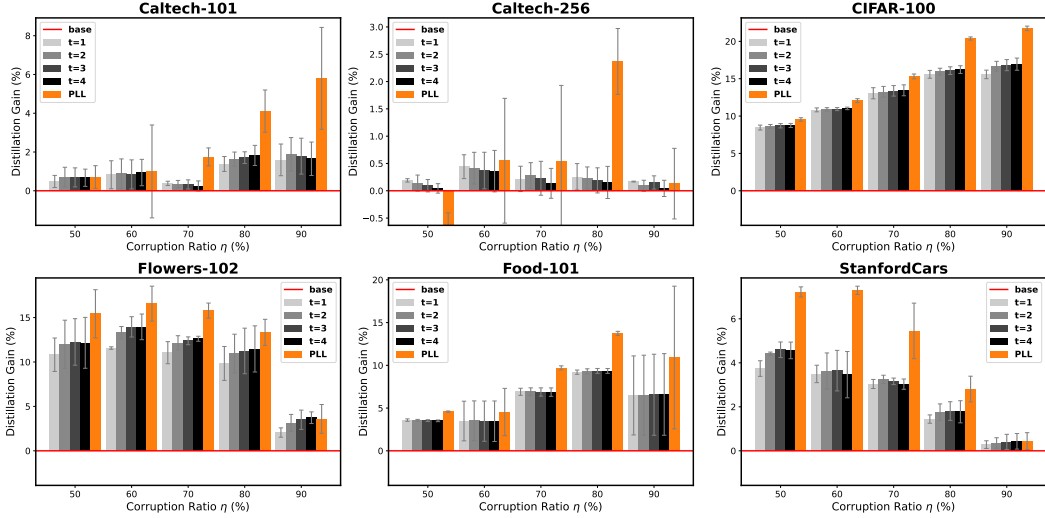

(b) Symmetric label corruption

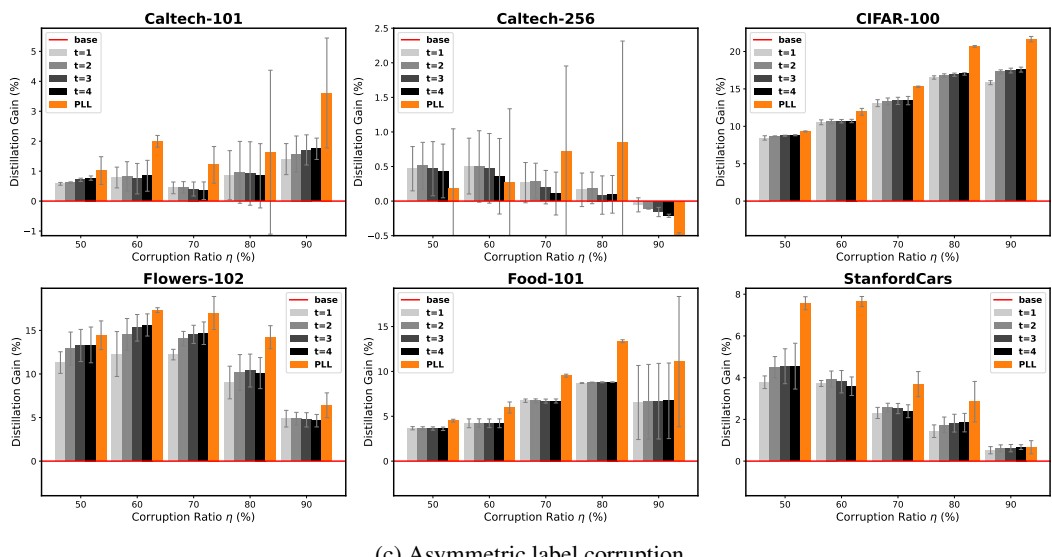

(c) Asymmetric label corruption

Figure 9: Distillation gain in accuracy (%) by each student model compared to the teacher model in three different label corruption scenarios. PLAIN represents the gap between the multi-round SD students and the teacher model, while ORANGE represents the gap between the PLL students and the teacher model. We use weight decay value $\lambda = 5 \times 10^{-4}$ for Caltech-101, Flowers-102, and StanfordCars datasets, and $\lambda = 5 \times 10^{-5}$ for CIFAR-100, Caltech-256, and Food-101 datasets.

## G FULL OUTPUT EVOLUTION OF FIG. 2B AND FIG. 5A

In Sec. 3, our analysis reveals that multi-round self-distillation effectively performs label averaging among instances with high feature correlations. This process leads to clustered predictions and improved generalization. To verify our claim, we conduct experiments on both synthetic and real data and illustrate the clustering effect of softmax outputs of the student models as the number of distillation rounds $t$ increases. Fig. 10 presents the full version of Fig. 2b, depicting the output evolution for $K = 4$ class classification task for synthetic dataset. Each dotted circle represents a cluster containing all instances of a true label. As the number of distillation round $t$ progresses, the student model's outputs converge to unique clustering points, depending on its true label.

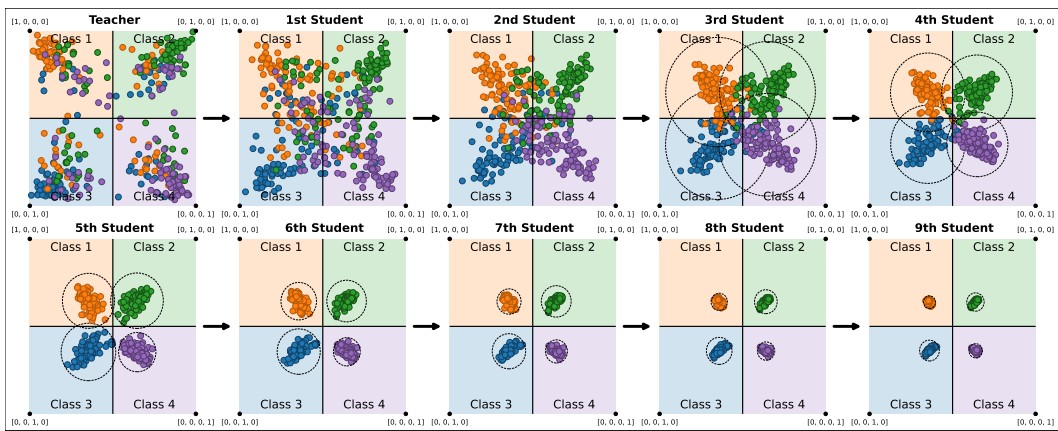

Figure 10: (Full version of Fig. 2b) The output evolution of the teacher model and the multi-round self-distilled student model. Each corner represents the one-hot vectors of the respective class. We use $n = 100$ and $\lambda = 3.125 \times 10^{-4}$, with 50% symmetric label corruption. The Gram matrix is assumed to follow a block-wise structure as described in (7), with parameters $c = 0.4$ and $d = 0.1$, added by a random perturbation matrix $\mathbf{E}$, where each entry $[\mathbf{E}]_{i,j} \sim U(-0.05, 0.05)$.

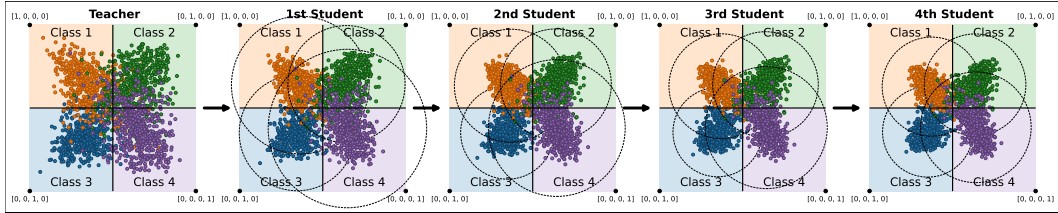

Figure 11: (Full version of Fig. 5a) The output evolution of the teacher model and the multi-round self-distilled student models for the 4-class CIFAR-100 dataset. Each corner represents the one-hot vectors of the respective class. We use $\lambda = 2.5 \times 10^{-4}$, with 50% symmetric label corruption.

Fig. 11 is the full version of Fig. 5a. For the CIFAR-100 dataset, we arbitrarily select 4 classes and use only the instances belonging to these 4 classes, with 500 samples per class. As the distillation step $t$ progresses, the outputs form clusters due to the label averaging effect.

# H  ABLATION STUDY

## H.1  TARGETING TOP-k PARTIAL LABELS IN PLL STUDENT MODEL

In Sec. 5, we theoretically demonstrate the effectiveness of our novel student model, the PLL student model. By selecting the top two label positions from the teacher model's output, we ensure the inclusion of the ground-truth label in the partial label list. While increasing the number of candidate labels may improve the likelihood of including the true label, it can also reduce confidence in the correct label for both noisy and clean-label samples, potentially leading to performance degradation. We conduct additional ablation studies on the sizes of candidate sets to validate the appropriateness of using only the top-two labels, as shown in Fig. 12. We observe that the model targeting the top-two vectors achieves the best trade-offs between accurately including the correct label in the partial label list while maintaining a high enough confidence for the correct label within the list, resulting in the best test accuracies across six benchmark datasets.

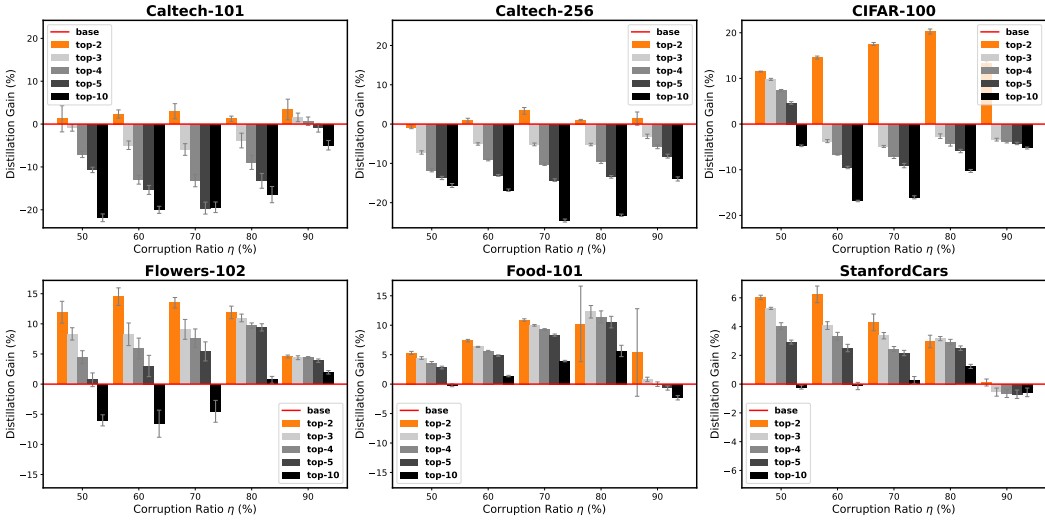

Figure 12: Full version of 5b. Distillation gain in accuracy (%) by each top-$k$ PLL student model compared to the teacher model in the superclass label corruption scenario. ORANGE represents the improvement by our proposed top-2 PLL student model compared to the teacher model, while PLAIN represents the gap between the top-$k$ PLL students and the teacher model for $k > 2$.

## H.2 Employing a Larger Feature Extractor Backbone

We also conduct additional experiments using a larger backbone–a pretrained Vision Transformer (Vit-B) (Dosovitskiy et al., 2021) model–as a fixed feature extractor. The distillation gains in test accuracy (%) over six benchmark datasets across different label corruption rates are presented in Fig. 13.

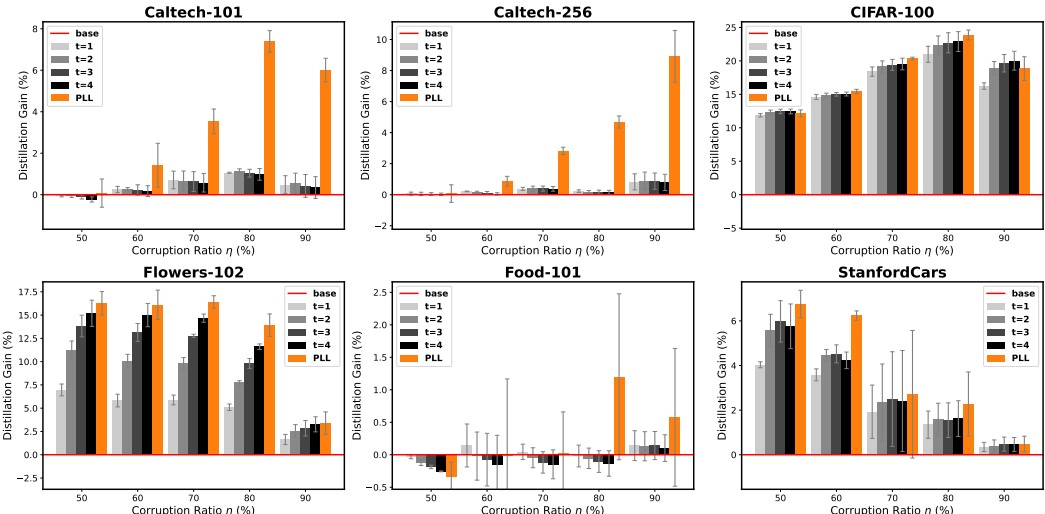

Figure 13: Distillation gain in accuracy (%) by each student model compared to the teacher model in three different label corruption scenarios, using ViT-B as a fixed feature extractor. PLAIN represents the gap between the multi-round SD students and the teacher model, while ORANGE represents the gap between the PLL students and the teacher model. We use weight decay value $\lambda = 5 \times 10^{-4}$ for Caltech-101, Flowers-102, and StanfordCars datasets, and $\lambda = 5 \times 10^{-5}$ for CIFAR-100, Caltech-256, and Food-101 datasets.

Using a larger backbone like ViT-B enhances feature extraction capabilities, resulting in a greater disparity between intra-class and inter-class feature correlations. For example, when calculating feature correlations on the CIFAR-100 dataset using the pretrained ViT-B model, we observe that the average intra-class feature correlation increases to 0.35, compared to 0.25 with ResNet34. This higher intra-class correlation amplifies the clustering effect of self-distillation on model predictions, allowing significant performance improvements to be achieved with fewer distillation steps, as implied by our Theorem 4.1.

In experiments with the larger backbone (ViT-B), we find that most of the distillation gains occur within the first few rounds. As shown in Fig. 13, nearly all performance improvements are observed in the first and second distillation steps when using ViT-B, although additional distillation steps still bring slight gains in high noise rate regimes. The PLL student model also effectively achieves the gains of multi-round self-distillation in a single round in high label-noise regimes for the ViT-B backbone. These additional experiments confirm that our approach is effective with larger models, further validating the versatility and robustness of our method.

# I  PROOF OF THEOREM 3.1

Recall (4), the equation for the coefficients $\{\boldsymbol{\alpha}_i^{(t)}\}_{i=1}^{Kn}$:

$$Kn\lambda\boldsymbol{\alpha}_i^{(t)} = \mathbf{y}_i^{(t-1)} - \sigma\left(\sum_{j=1}^{Kn}\langle\phi(\mathbf{x}_i),\phi(\mathbf{x}_j)\rangle\boldsymbol{\alpha}_j^{(t)}\right). \tag{25}$$

By applying the softmax approximation (Assumption 1), we have

$$K^2 n\lambda\boldsymbol{\alpha}_i^{(t)} = K\mathbf{y}_i^{(t-1)} - \sum_{j=1}^{Kn}\langle\phi(\mathbf{x}_i),\phi(\mathbf{x}_j)\rangle\boldsymbol{\alpha}_j^{(t)} - \mathbf{1}_K. \tag{26}$$

Define the output matrix $\mathbf{Y}^{(t)} \in \mathbb{R}^{K\times Kn}$ and the coefficient matrix $\mathbf{A}^{(t)} \in \mathbb{R}^{K\times Kn}$ as:

$$\mathbf{Y}^{(t)} := [\mathbf{y}_1^{(t)},\ldots,\mathbf{y}_{Kn}^{(t)}], \quad \mathbf{A}^{(t)} := [\boldsymbol{\alpha}_1^{(t)},\ldots,\boldsymbol{\alpha}_{Kn}^{(t)}]. \tag{27}$$

We also define the Gram matrix $\boldsymbol{\Phi} \in \mathbb{R}^{Kn\times Kn}$, where each entry is given by

$$[\boldsymbol{\Phi}]_{i,j} = \langle\phi(\mathbf{x}_i),\phi(\mathbf{x}_j)\rangle, \quad \forall i,j \in [Kn]. \tag{28}$$

Denoting the $i$-th column of a matrix $\mathbf{A}$ by $\mathbf{A}[:,i]$, (26) can be written as

$$K^2 n\lambda\mathbf{A}^{(t)}[:,i] = K\mathbf{Y}^{(t-1)}[:,i] - \sum_{j=1}^{Kn}[\boldsymbol{\Phi}]_{i,j}\mathbf{A}^{(t)}[:,j] - \mathbf{1}_K$$

$$= K\mathbf{Y}^{(t-1)}[:,i] - (\mathbf{A}^{(t)}\boldsymbol{\Phi}^\top)[:,i] - \mathbf{1}_{K\times Kn}[:,i].$$

Since $\boldsymbol{\Phi}$ is symmetric, we have

$$\mathbf{A}^{(t)}(K^2 n\lambda\mathbf{I}_{Kn} + \boldsymbol{\Phi}) = K\mathbf{Y}^{(t-1)} - \mathbf{1}_{K\times Kn}.$$

Also, from the definition of $\mathbf{A}^{(t)}$, we obtain the relation between $\mathbf{Y}^{(t)}$ and $\mathbf{Y}^{(t-1)}$ as follows:

$$\frac{1}{Kn\lambda}\left(\mathbf{Y}^{(t-1)} - \mathbf{Y}^{(t)}\right) = K\left(\mathbf{Y}^{(t-1)} - \frac{1}{K}\mathbf{1}_{K\times Kn}\right)\left(K^2 n\lambda\mathbf{I}_{Kn} + \boldsymbol{\Phi}\right)^{-1};$$

$$\left(\mathbf{Y}^{(t-1)} - \mathbf{Y}^{(t)}\right) = \left(\mathbf{Y}^{(t-1)} - \frac{1}{K}\mathbf{1}_{K\times Kn}\right)\left(\mathbf{I}_{Kn} + \frac{1}{K^2 n\lambda}\boldsymbol{\Phi}\right)^{-1};$$

$$\left(\mathbf{Y}^{(t-1)} - \frac{1}{K}\mathbf{1}_{K\times Kn}\right) - \left(\mathbf{Y}^{(t)} - \frac{1}{K}\mathbf{1}_{K\times Kn}\right) = \left(\mathbf{Y}^{(t-1)} - \frac{1}{K}\mathbf{1}_{K\times Kn}\right)\left(\mathbf{I}_{Kn} + \frac{1}{K^2 n\lambda}\boldsymbol{\Phi}\right)^{-1};$$

$$\left(\mathbf{Y}^{(t)} - \frac{1}{K}\mathbf{1}_{K\times Kn}\right) = \left(\mathbf{Y}^{(t-1)} - \frac{1}{K}\mathbf{1}_{K\times Kn}\right)\left[\mathbf{I}_{Kn} - \left(\mathbf{I}_{Kn} + \frac{1}{K^2 n\lambda}\boldsymbol{\Phi}\right)^{-1}\right].$$

From this recursive relation, we can express $\mathbf{Y}^{(t)}$ in terms of $\mathbf{Y}^{(0)}$ and $\boldsymbol{\Phi}$ as:

$$\mathbf{Y}^{(t)} - \frac{1}{K}\mathbf{1}_{K\times Kn} = \left(\mathbf{Y}^{(0)} - \frac{1}{K}\mathbf{1}_{K\times Kn}\right)\left[\mathbf{I}_{Kn} - \left(\mathbf{I}_{Kn} + \frac{1}{K^2 n\lambda}\boldsymbol{\Phi}\right)^{-1}\right]^t. \tag{29}$$

Similarly, for the outputs of the student model with partial labels, the only difference is that $\mathbf{Y}^{(0)}$ is replaced by $\bar{\mathbf{Y}}$:

$$\mathbf{Y}^{(P)} - \frac{1}{K}\mathbf{1}_{K\times Kn} = \left(\bar{\mathbf{Y}} - \frac{1}{K}\mathbf{1}_{K\times Kn}\right)\left[\mathbf{I}_{Kn} - \left(\mathbf{I}_{Kn} + \frac{1}{K^2 n\lambda}\boldsymbol{\Phi}\right)^{-1}\right], \tag{30}$$

where $\mathbf{Y}^{(P)}$ is the output of the student model with partial labels using the input $\bar{\mathbf{Y}} = [\bar{\mathbf{y}}_1,\cdots,\bar{\mathbf{y}}_{Kn}]$.

## J  Proof of Lemma 4.1 and the Closed-Form Output Solutions for Five Different Feature Correlations

Recall the closed-form solution for $\mathbf{Y}^{(t)}$ from (6):

$$\mathbf{Y}^{(t)} - \frac{1}{K}\mathbf{1}_{K\times Kn} = \left(\mathbf{Y}^{(0)} - \frac{1}{K}\mathbf{1}_{K\times Kn}\right)\sum_{i=1}^{Kn}\left(\frac{\lambda_i}{K^2 n\lambda + \lambda_i}\right)^t \mathbf{v}_i\mathbf{v}_i^\top, \tag{31}$$

where $\{(\lambda_i, \mathbf{v}_i)\}_{i=1}^{Kn}$ represents the eigenvalue-eigenvector pairs of the Gram matrix $\mathbf{\Phi}$, satisfying $\mathbf{\Phi} = \sum_{i=1}^{Kn}\lambda_i\mathbf{v}_i\mathbf{v}_i^\top$. In this section, we compute $\{(\lambda_i, \mathbf{v}_i)\}_{i=1}^{Kn}$ for diverse class-wise feature correlation scenarios and simplify the closed-form solution for $\mathbf{Y}^{(t)}$.

Assume the feature correlation between two instances $\langle\phi(\mathbf{x}_i), \phi(\mathbf{x}_j)\rangle$ is determined by their true labels:

$$\langle\phi(\mathbf{x}_i), \phi(\mathbf{x}_j)\rangle = \begin{cases} 1, & i = j; \\ \omega(y_i, y_j), & i \neq j. \end{cases} \tag{32}$$

Define the class-wise feature correlation matrix $\mathbf{W} \in \mathbb{R}^{K\times K}$, with entries

$$[\mathbf{W}]_{i,j} := \omega(i, j), \quad \forall i, j \in [K]. \tag{33}$$

Without loss of generality, we assume that the dataset is sorted by true labels $y_i$. Then, the Gram matrix $\mathbf{\Phi}$ can be written as

$$\mathbf{\Phi} = \underbrace{\begin{pmatrix} & & & \\ & & & \\ & & & \\ & & & \end{pmatrix}}_{\text{block matrix with } n\times n \text{ blocks}} + \underbrace{\begin{pmatrix} & & & \\ & & & \\ & & & \\ & & & \end{pmatrix}}_{\text{diagonal matrix}} = \mathbf{P}^\top\mathbf{W}\mathbf{P} + \mathbf{D}([1 - [\mathbf{W}]_{i,i}]_{i=1}^K\mathbf{P}), \tag{34}$$

where $\mathbf{D}(\mathbf{z})$ is a diagonal matrix with elements from the vector $\mathbf{z}$, and $\mathbf{P} \in \mathbb{R}^{K\times Kn}$ is defined as:

$$\mathbf{P} = [\mathbf{e}(y_1), \dots, \mathbf{e}(y_{Kn})] = \begin{pmatrix} \underbrace{1\cdots 1}_{n} & \underbrace{0\cdots 0}_{n} & \cdots & 0\cdots 0 \\ 0\dots 0 & 1\cdots 1 & \cdots & 0\cdots 0 \\ \vdots & \vdots & \ddots & \\ 0\cdots 0 & 0\cdots 0 & \cdots & \underbrace{1\cdots 1}_{n} \end{pmatrix}. \tag{35}$$

In the following sub-sections, we derive closed-form solutions for the model outputs (6) for five different class-wise feature correlations $\mathbf{W}$.

### J.1  Case I: Positive Correlations Only Among Same-Class Instances

Assume that the class-wise feature correlation $\omega(y_i, y_j)$ is defined as follows:

$$\omega(y_i, y_j) = \begin{cases} c, & y_i = y_j; \\ 0, & y_i \neq y_j, \end{cases} \tag{36}$$

for $c > 0$. From (34), the Gram matrix $\mathbf{\Phi}$ can be computed as:

$$\begin{aligned} \mathbf{\Phi} &= \mathbf{P}^\top\mathbf{W}\mathbf{P} + \mathbf{D}([1 - [\mathbf{W}]_{i,i}]_{i=1}^K\mathbf{P}) \\ &= \mathbf{P}^\top(c\mathbf{I}_K)\mathbf{P} + \mathbf{D}([1 - c]_{i=1}^K\mathbf{P}) \\ &= nc\left(\frac{1}{\sqrt{n}}\mathbf{P}\right)^\top\left(\frac{1}{\sqrt{n}}\mathbf{P}\right) + (1 - c)\mathbf{I}_{Kn}. \end{aligned}$$

Let us represent $\mathbf{P}$ in (35) by $\mathbf{P} = [\mathbf{p}_1, \dots, \mathbf{p}_K]^\top$ for $\mathbf{p}_i \in \mathbb{R}^{Kn}$. It can be shown that $\mathbf{\Phi}$ has two types of eigenvalue-eigenvector pairs $\{\lambda_i, \mathbf{v}_i\}_{i=1}^{Kn}$:

- $K$ pairs with eigenvalues
$$\lambda_i = nc + 1 - c, \quad \text{for } i = 1, \dots, K,$$
and the corresponding eigenvectors $\mathbf{v}_i = \frac{1}{\sqrt{n}}\mathbf{p}_i$;
- $(Kn - K)$ pairs with the eigenvalues $(1 - c)$ and the corresponding eigenvectors, denoted by $\mathbf{v}_i$ for $i = K + 1, \dots, Kn$.

Note that $\sum_{i=K+1}^{Kn} \mathbf{v}_i \mathbf{v}_i^\top$, the sum of outer-product of $(Kn - K)$ eigenvectors satisfies

$$\sum_{i=K+1}^{Kn} \mathbf{v}_i \mathbf{v}_i^\top = \mathbf{I}_{Kn} - \sum_{i=1}^{K} \mathbf{v}_i \mathbf{v}_i^\top = \mathbf{I}_{Kn} - \frac{1}{n}\mathbf{P}^\top\mathbf{P}. \tag{37}$$

Hence, the label averaging matrix $\mathbf{\Phi}^{(t)}$ is given as

$$\mathbf{\Phi}^{(t)} = \sum_{i=1}^{Kn} \left( \frac{\lambda_i}{K^2 n\lambda + \lambda_i} \right)^t \mathbf{v}_i \mathbf{v}_i^\top$$

$$= \frac{1}{n} \left( \frac{1 - c + nc}{K^2 n\lambda + 1 - c + nc} \right)^t \mathbf{P}^\top\mathbf{P} + \left( \frac{1 - c}{K^2 n\lambda + 1 - c} \right)^t \left( \mathbf{I}_{Kn} - \frac{1}{n}\mathbf{P}^\top\mathbf{P} \right).$$

For simplicity, we define two constants $p$ and $q$:

$$p = \frac{1 - c}{K^2 n\lambda + 1 - c}, \quad q = \frac{1 - c + nc}{K^2 n\lambda + 1 - c + nc}. \tag{38}$$

The label averaging matrix $\mathbf{\Phi}^{(t)}$ can be expressed as

$$\mathbf{\Phi}^{(t)} = \underbrace{\frac{1}{n}(q^t - p^t)\mathbf{P}^\top\mathbf{P}}_{\text{class-wise label averaging matrix}} + p^t\mathbf{I}_{Kn}. \tag{39}$$

Then, the closed-form of $\mathbf{Y}^{(t)}$ is

$$\mathbf{Y}^{(t)} = \left( \mathbf{Y}^{(0)} - \frac{1}{K}\mathbf{1}_{K\times Kn} \right) \mathbf{\Phi}^{(t)} + \frac{1}{K}\mathbf{1}_{K\times Kn}$$

$$= \left( \mathbf{Y}^{(0)} - \frac{1}{K}\mathbf{1}_{K\times Kn} \right) \left( \frac{1}{n}(q^t - p^t)\mathbf{P}^\top\mathbf{P} + p^t\mathbf{I}_{Kn} \right) + \frac{1}{K}\mathbf{1}_{K\times Kn}$$

$$= p^t\mathbf{Y}^{(0)} + \frac{1}{n}\left( q^t - p^t \right)\mathbf{Y}^{(0)}\mathbf{P}^\top\mathbf{P} + \frac{1}{K}(1 - q^t)\mathbf{1}_{K\times Kn},$$

where the last equality holds due to $\mathbf{1}_{K\times Kn}\mathbf{P}^\top\mathbf{P} = n\mathbf{1}_{K\times Kn}$.

Finally, we arrive at the closed form of $\mathbf{y}_i^{(t)}$:

$$\mathbf{y}_i^{(t)} = p^t\mathbf{e}(\hat{y}_i) + \frac{1}{n}(q^t - p^t) \sum_{\{j:y_j=y_i\}} \mathbf{e}(\hat{y}_j) + \frac{1}{K}(1 - q^t)\mathbf{1}_K. \tag{40}$$

## J.2 CASE II: CLASS-DEPENDENT INTRA-CLASS CORRELATIONS

Let the feature correlation between two samples with the same true label change depending on the class. We model the feature correlation between instances as a function of their true labels:

$$\omega(y_i, y_j) = \begin{cases} \omega(y_i), & y_i = y_j; \\ 0, & y_i \neq y_j. \end{cases} \tag{41}$$

Here, we assume $\omega(y_i) > 0$ for all $y_i \in [K]$. The Gram matrix $\mathbf{\Phi}$ can be computed as

$$\mathbf{\Phi} = \mathbf{P}^\top\mathbf{W}\mathbf{P} + \mathbf{D}([1 - [\mathbf{W}]_{i,i}]_{i=1}^K\mathbf{P})$$

$$= \mathbf{P}^\top\mathbf{D}([\omega(1), \dots, \omega(K)])\mathbf{P} + \mathbf{D}([(1 - \omega(i))]_{i=1}^K\mathbf{P})$$

$$= \sum_{i=1}^{K} n\omega(i) \left( \frac{1}{\sqrt{n}}\mathbf{p}_i \right) \left( \frac{1}{\sqrt{n}}\mathbf{p}_i \right)^\top + \mathbf{D}([(1 - \omega(i))]_{i=1}^K\mathbf{P}).$$

It can be checked that $\mathbf{\Phi}$ has two types of eigenvalue-eigenvector pairs:

- $K$ pairs with eigenvalues
$$\lambda_i = n\omega(i) + 1 - \omega(i), \quad i = 1, \dots, K,$$
and the corresponding eigenvectors $\mathbf{v}_i = \frac{1}{\sqrt{n}}\mathbf{p}_i$;

- $(Kn - K)$ pairs with eigenvalues
$$[\lambda_{K+1}, \dots, \lambda_{Kn}] = [\underbrace{1 - \omega(1), \dots, 1 - \omega(1)}_{(n-1) \text{ eigenvalues}}, \dots, \underbrace{1 - \omega(K), \dots, 1 - \omega(K)}_{(n-1) \text{ eigenvalues}}], \quad (42)$$
and the corresponding eigenvectors $\mathbf{v}_i$ for $i = K + 1, \dots, Kn$.

Hence, the label averaging matrix $\mathbf{\Phi}^{(t)}$ is given as

$$\mathbf{\Phi}^{(t)} = \sum_{i=1}^{Kn} \left(\frac{\lambda_i}{K^2 n\lambda + \lambda_i}\right)^t \mathbf{v}_i \mathbf{v}_i^\top$$

$$= \frac{1}{n}\sum_{i=1}^{K}\left(\frac{1 - \omega(i) + n\omega(i)}{K^2 n\lambda + 1 - \omega(i) + n\omega(i)}\right)^t \mathbf{p}_i^\top \mathbf{p}_i + \sum_{i=1}^{K}\left(\frac{1 - \omega(i)}{K^2 n\lambda + 1 - \omega(i)}\right)^t \sum_{\{j:\lambda_j=1-\omega(i)\}} \mathbf{v}_j \mathbf{v}_j^\top$$

Note that

$$\sum_{\{j:\lambda_j=1-\omega(i)\}} \mathbf{v}_j \mathbf{v}_j^\top = \mathbf{D}([\underbrace{0, \dots, 0}_{(i-1)n}, \underbrace{1, \dots, 1}_{n}, \underbrace{0, \dots, 0}_{(K-i)n}]) - \frac{1}{n}\mathbf{p}_i^\top \mathbf{p}_i. \quad (43)$$

For simplicity, let us define two vectors $\mathbf{p} = [p_1, \dots, p_K]^\top, \mathbf{q} = [q_1, \dots, q_K]^\top \in \mathbb{R}^K$:

$$p_i = \frac{1 - \omega(i)}{K^2 n\lambda + 1 - \omega(i)}, \quad q_i = \frac{1 - \omega(i) + n\omega(i)}{K^2 n\lambda + 1 - \omega(i) + n\omega(i)}. \quad (44)$$

Then, $\mathbf{\Phi}^{(t)}$ can be simplified as

$$\mathbf{\Phi}^{(t)} = \sum_{i=1}^{K}\frac{1}{n}(q_i^t - p_i^t)\mathbf{p}_i^\top \mathbf{p}_i + \sum_{i=1}^{K} p_i^t \mathbf{D}([\underbrace{0, \dots, 0}_{(i-1)n}, \underbrace{1, \dots, 1}_{n}, \underbrace{0, \dots, 0}_{(K-i)n}])$$

$$= \underbrace{\frac{1}{n}\mathbf{P}^\top \mathbf{D}(\mathbf{q}^t - \mathbf{p}^t)\mathbf{P}}_{\text{class-wise label averaging matrix}} + \mathbf{D}(\mathbf{p}^t\mathbf{P}).$$

Therefore, we arrive at the closed form for $\mathbf{y}_i^{(t)}$:

$$\mathbf{y}_i^{(t)} = \left(\mathbf{Y}^{(0)} - \frac{1}{K}\mathbf{1}_{K\times Kn}\right)\mathbf{\Phi}^{(t)}[:, i] + \frac{1}{K}\mathbf{1}_K$$

$$= p_{y_i}^t \mathbf{e}(\hat{y}_i) + \frac{1}{n}\left(q_{y_i}^t - p_{y_i}^t\right)\sum_{\{j:y_i=y_j\}}\mathbf{e}(\hat{y}_j) + \frac{1}{K}\left(1 - q_{y_i}^t\right)\mathbf{1}_K. \quad (45)$$

### J.3 CASE III: CORRELATION GAP BETWEEN INTRA- AND INTER-CLASS INSTANCES

Let $c$ be the feature correlation between two samples with the same true label, and let $d$ be the correlation between two samples with different true labels. The class-wise feature correlation $\omega(y_i, y_j)$ is defined as follows:

$$\omega(y_i, y_j) = \begin{cases} c, & y_i = y_j; \\ d, & y_i \neq y_j, \end{cases} \quad (46)$$

for $c > d \geq 0$. The Gram matrix $\mathbf{\Phi}$ can be computed as

$$\mathbf{\Phi} = \mathbf{P}^\top \mathbf{W}\mathbf{P} + \mathbf{D}([1 - [\mathbf{W}]_{i,i}]_{i=1}^{K}\mathbf{P})$$

$$= \mathbf{P}^\top(d\mathbf{1}_{K\times K} + (c - d)\mathbf{I}_K)\mathbf{P} + (1 - c)\mathbf{I}_{Kn}.$$

Note that $\mathbf{W} = d\mathbf{1}_{K\times K} + (c - d)\mathbf{I}_K$ is a type of diagonal-shift to a low-rank matrix, which has two types of eigenvalue-eigenvector pairs $\{\sigma_i, \mathbf{u}_i\}_{i=1}^{K}$:

- one pair with an eigenvalue
$$\sigma_1 = Kd + (c - d),$$
  and the corresponding eigenvector $\mathbf{u}_1 = \frac{1}{\sqrt{K}}\mathbf{1}_K$;
- $(K - 1)$ pairs with eigenvalues
$$\sigma_i = (c - d), \quad i = 2, \ldots, K,$$
  and the corresponding eigenvectors $\mathbf{u}_i$.

Then, the Gram matrix $\boldsymbol{\Phi}$ can be expressed as
$$\boldsymbol{\Phi} = \mathbf{P}^\top \mathbf{W} \mathbf{P} + (1 - c)\mathbf{I}_{Kn} = \sum_{i=1}^{K} n\sigma_i \left(\frac{1}{\sqrt{n}}\mathbf{P}^\top \mathbf{u}_i\right)\left(\frac{1}{\sqrt{n}}\mathbf{P}^\top \mathbf{u}_i\right)^\top + (1 - c)\mathbf{I}_{Kn}.$$

It can be checked that $\boldsymbol{\Phi}$ has three types of eigenvalue-eigenvector pairs:

- one pair with an eigenvalue
$$\lambda_1 = n\sigma_1 + (1 - c) = Knd + n(c - d) + (1 - c),$$
  and the corresponding eigenvector
$$\mathbf{v}_1 = \frac{1}{\sqrt{n}}\mathbf{P}^\top \mathbf{u}_1 = \frac{1}{\sqrt{Kn}}\mathbf{P}^\top \mathbf{1}_K;$$
- $(K - 1)$ pairs with eigenvalues
$$\lambda_i = n\sigma_i + (1 - c) = n(c - d) + (1 - c), \quad i = 2, \ldots, K,$$
  and the corresponding eigenvectors
$$\mathbf{v}_i = \frac{1}{\sqrt{n}}\mathbf{P}^\top \mathbf{u}_i, \quad i = 2, \ldots, K;$$
- $(Kn - K)$ pairs with eigenvalues
$$\lambda_i = (1 - c), \quad i = K + 1, \ldots, Kn,$$
  and the corresponding eigenvectors $\mathbf{v}_i$'s.

Define $p$, $q$, and $r$ according to the following equations:
$$p = \frac{1 - c}{K^2 n\lambda + 1 - c},$$
$$q = \frac{1 - c + n(c - d)}{K^2 n\lambda + 1 - c + n(c - d)},$$
$$r = \frac{1 - c + n(c - d) + Knd}{K^2 n\lambda + 1 - c + n(c - d) + Knd}.$$

Then, the label averaging matrix $\boldsymbol{\Phi}^{(t)}$ is given as
$$\boldsymbol{\Phi}^{(t)} = \sum_{i=1}^{Kn} \left(\frac{\lambda_i}{K^2 n\lambda + \lambda_i}\right)^t \mathbf{v}_i \mathbf{v}_i^\top$$
$$= \frac{r^t}{n}\mathbf{P}^\top \mathbf{u}_1 \mathbf{u}_1^\top \mathbf{P} + \frac{q^t}{n}\sum_{i=2}^{K}\mathbf{P}^\top \mathbf{u}_i \mathbf{u}_i^\top \mathbf{P} + p^t \sum_{i=K+1}^{Kn} \mathbf{v}_i \mathbf{v}_i^\top$$
$$= \frac{r^t}{Kn}\mathbf{P}^\top \mathbf{1}_{K \times K}\mathbf{P} + \frac{q^t}{n}\mathbf{P}^\top \left[\mathbf{I}_K - \frac{1}{K}\mathbf{1}_{K \times K}\right]\mathbf{P} + p^t \left(\mathbf{I}_{Kn} - \frac{1}{n}\sum_{i=1}^{K}\mathbf{P}^\top \mathbf{u}_i \mathbf{u}_i^\top \mathbf{P}\right)$$
$$= \frac{r^t - q^t}{Kn}\mathbf{1}_{Kn \times Kn} + \underbrace{\frac{q^t - p^t}{n}\mathbf{P}^\top \mathbf{P}}_{\text{class-wise label averaging matrix}} + p^t \mathbf{I}_{Kn}.$$

Finally, we obtain the closed form for $\mathbf{y}_i^{(t)}$:

$$
\begin{aligned}
\mathbf{y}_i^{(t)} &= \left( \mathbf{Y}^{(0)} - \frac{1}{K} \mathbf{1}_{K \times Kn} \right) \mathbf{\Phi}^{(t)}[:, i] + \frac{1}{K} \mathbf{1}_K \\
&= p^t \mathbf{e}(\hat{y}_i) + (q^t - p^t) \left( \frac{1}{n} \sum_{\{j:y_i=y_j\}} \mathbf{e}(\hat{y}_j) \right) + (r^t - q^t) \left( \frac{1}{Kn} \sum_{j=1}^{Kn} \mathbf{e}(\hat{y}_j) \right) + (1 - r^t) \frac{1}{K} \mathbf{1}_K \\
&= p^t \mathbf{e}(\hat{y}_i) + (q^t - p^t) \left( \frac{1}{n} \sum_{\{j:y_i=y_j\}} \mathbf{e}(\hat{y}_j) \right) + (1 - q^t) \frac{1}{K} \mathbf{1}_K.
\end{aligned}
\tag{47}
$$

The last equality holds due to the balanced given labels, $\{\hat{y}_i\}$, across the $K$ classes. Note that the closed form for $\mathbf{y}_i^{(t)}$ consists of the following three components, across all the three cases considered so far:

- $\mathbf{e}(\hat{y}_i)$: the given label,
- $\frac{1}{n} \sum_{\{j:y_j=y_i\}} \mathbf{e}(\hat{y}_j)$: the average label vector among samples from the same true class $y_i$,
- $\frac{1}{K} \mathbf{1}_K$: the uniform vector.

### J.4 Case IV: Superclass Feature Correlation (Proof of Lemma 4.1)

Let us assume that there exists a set of superclasses, where the classes within the same superclass have higher feature correlation, while the classes from different superclasses have zero feature correlation. Let $h : [K] \to [R]$ be a function that map the class index to a superclass index. The model for the superclass feature correlation is defined as

$$
\omega(y_i, y_j) = \begin{cases} c, & y_i = y_j; \\ d, & y_i \neq y_j, h(y_i) = h(y_j); \\ 0, & h(y_i) \neq h(y_j) \end{cases}
\tag{48}
$$

for $c > d \geq 0$. Without loss of generality, we assume that the labels are sorted according to the superclass index, which implies that $h$ is a non-decreasing function. Then, the class-wise feature correlation matrix $\mathbf{W} \in \mathbb{R}^{K \times K}$ can be expressed as the sum of rank-$R$ block diagonal matrix and a diagonal matrix:

$$
\mathbf{W} = \underbrace{\phantom{\Bigg(\Bigg)}}_{\text{block diagonal matrix}} + \underbrace{\phantom{\Bigg(\Bigg)}}_{\text{diagonal matrix}} = d\mathbf{Q}^\top \mathbf{Q} + (c - d)\mathbf{I}_K,
\tag{49}
$$

where $\mathbf{Q} \in \mathbb{R}^{R \times K}$ are defined as

$$
\mathbf{Q} = [\mathbf{e}(h(1)), \dots, \mathbf{e}(h(R))] = \begin{pmatrix} \underbrace{1 \cdots 1}_{K_1} & \underbrace{0 \cdots 0}_{K_2} & \cdots & 0 \cdots 0 \\ 0 \dots 0 & 1 \cdots 1 & \cdots & 0 \cdots 0 \\ \vdots & \vdots & \ddots & \\ 0 \cdots 0 & 0 \cdots 0 & \cdots & \underbrace{1 \cdots 1}_{K_R} \end{pmatrix}.
\tag{50}
$$

with $K_i := |\{k \in [K] : h(k) = i\}|$ for $i \in [R]$, where $\sum_{i=1}^R K_i = K$.

Let us represent $\mathbf{Q}$ in (50) as $\mathbf{Q} = [\mathbf{q}_1, \dots, \mathbf{q}_R]^\top$ for $\mathbf{q}_i \in \mathbb{R}^K$. Then, $\mathbf{W}$ can be written as

$$
\mathbf{W} = \sum_{i=1}^R K_i d \left( \frac{1}{\sqrt{K_i}} \mathbf{q}_i \right) \left( \frac{1}{\sqrt{K_i}} \mathbf{q}_i \right)^\top + (c - d)\mathbf{I}_K.
$$

It can be shown that $\mathbf{W}$ has two types of eigenvalue-eigenvector pairs $\{\sigma_i, \mathbf{u}_i\}_{i=1}^K$:

- $R$ pairs with eigenvalues
$$\sigma_i = K_i d + (c - d), \quad i = 1, \ldots, R,$$
and the corresponding eigenvectors $\mathbf{u}_i = \frac{1}{\sqrt{K_i}} \mathbf{q}_i$.

- $(K - R)$ pairs with the eigenvalues
$$\sigma_i = (c - d), \quad i = R + 1, \ldots, K,$$
and the corresponding eigenvectors $\mathbf{u}_i$.

Then, $\boldsymbol{\Phi}$ can be expressed as follows:

$$\boldsymbol{\Phi} = \mathbf{P}^\top \mathbf{W} \mathbf{P} + (1 - c)\mathbf{I}_{Kn} = \sum_{i=1}^{K} n\sigma_i \left( \frac{1}{\sqrt{n}} \mathbf{P}^\top \mathbf{u}_i \right) \left( \frac{1}{\sqrt{n}} \mathbf{P}^\top \mathbf{u}_i \right)^\top + (1 - c)\mathbf{I}_{Kn}$$

Thus, it can be shown that $\boldsymbol{\Phi}$ has three types of eigenvalue-eigenvector pairs:

- $R$ pairs with eigenvalues
$$\lambda_i = n\sigma_i + (1 - c) = K_i n d + n(c - d) + (1 - c), \quad i = 1, \ldots, R,$$
and the corresponding eigenvectors
$$\mathbf{v}_i = \frac{1}{\sqrt{n}} \mathbf{P}^\top \mathbf{u}_i = \frac{1}{\sqrt{K_i n}} \mathbf{q}_i;$$

- $(K - R)$ pairs with eigenvalues
$$\lambda_i = n\sigma_i + (1 - c) = n(c - d) + (1 - c), \quad i = R + 1, \ldots, K,$$
and the corresponding eigenvectors
$$\mathbf{v}_i = \frac{1}{\sqrt{n}} \mathbf{P}^\top \mathbf{u}_i;$$

- $(Kn - K)$ pairs of eigenvalues
$$\lambda_i = (1 - c), \quad i = K + 1, \ldots, Kn,$$
with eigenvectors $\mathbf{v}_i$.

Define $p$, $q$ and $\mathbf{r} = [r_1, \ldots, r_R] \in \mathbb{R}^R$ as follows:
$$p = \frac{1 - c}{K^2 n\lambda + 1 - c}$$
$$q = \frac{1 - c + n(c - d)}{K^2 n\lambda + 1 - c + n(c - d)}$$
$$r_i = \frac{1 - c + n(c - d) + K_i n d}{K^2 n\lambda + 1 - c + n(c - d) + K_i n d}, \quad i \in [R].$$

Then, the label averaging matrix $\boldsymbol{\Phi}^{(t)}$ is given as

$$\boldsymbol{\Phi}^{(t)} = \sum_{i=1}^{Kn} \left( \frac{\lambda_i}{K^2 n\lambda + \lambda_i} \right)^t \mathbf{v}_i \mathbf{v}_i^\top$$

$$= \sum_{i=1}^{R} \frac{r_i^t}{n} \mathbf{P}^\top \mathbf{u}_i \mathbf{u}_i^\top \mathbf{P} + \frac{q^t}{n} \sum_{i=R+1}^{K} \mathbf{P}^\top \mathbf{u}_i \mathbf{u}_i^\top \mathbf{P} + p^t \sum_{i=K+1}^{Kn} \mathbf{v}_i \mathbf{v}_i^\top$$

$$= \sum_{i=1}^{R} \frac{r_i^t}{n} \mathbf{P}^\top \mathbf{u}_i \mathbf{u}_i^\top \mathbf{P} + \frac{q^t}{n} \mathbf{P}^\top \left[ \mathbf{I}_K - \sum_{i=1}^{R} \mathbf{u}_i \mathbf{u}_i^\top \right] \mathbf{P} + p^t \left[ \mathbf{I}_{Kn} - \frac{1}{n} \sum_{i=1}^{K} \mathbf{P}^\top \mathbf{u}_i \mathbf{u}_i^\top \mathbf{P} \right]$$

$$= \sum_{i=1}^{R} \frac{r_i^t - q^t}{K_i n} \mathbf{P}^\top \mathbf{q}_i \mathbf{q}_i^\top \mathbf{P} + \frac{q^t - p^t}{n} \mathbf{P}^\top \mathbf{P} + p^t \mathbf{I}_{Kn}$$

$$= \underbrace{\mathbf{P}^\top \mathbf{D} \left( \left[ \frac{r_i^t - q^t}{K_i n} \mathbf{1}_{K_i \times K_i} \right]_{i=1}^{R} \right) \mathbf{P}}_{\text{superclass-wise label averaging matrix}} + \underbrace{\frac{q^t - p^t}{n} \mathbf{P}^\top \mathbf{P}}_{\text{class-wise label averaging matrix}} + p^t \mathbf{I}_{Kn}.$$

Finally, we arrive at the closed form of $\mathbf{y}_i^{(t)}$,

$$\mathbf{y}_i^{(t)} = p^t\mathbf{e}(\hat{y}_i) + (q^t - p^t)\left(\frac{1}{n}\sum_{\{j:y_i=y_j\}}\mathbf{e}(\hat{y}_j)\right) + (r_s^t - q^t)\left(\frac{1}{K_s n}\sum_{\{j:h(y_j)=s\}}\mathbf{e}(\hat{y}_j)\right) + (1-r_s^t)\frac{1}{K}\mathbf{1}_K \tag{51}$$

where $s = h(y_i)$ is the superclass of $y_i$. The closed form for $\mathbf{y}_i^{(t)}$ consists of the following four components:

- $\mathbf{e}(\hat{y}_i)$: the given label,
- $\frac{1}{n}\sum_{\{j:y_j=y_i\}}\mathbf{e}(\hat{y}_j)$: the average label vector among samples from the same true class $y_i$,
- $\frac{1}{K_s n}\sum_{\{j:h(y_j)=s\}}\mathbf{e}(\hat{y}_j)$: the average label vector among samples from the same superclass class $s = h(y_i)$,
- $\frac{1}{K}\mathbf{1}_K$: the uniform vector.

### J.5 CASE V: EXTENDED SUPERCLASS FEATURE CORRELATION

As the last case, we allow a non-zero inter-superclass correlation, fixed as a constant $e$. The model for the class-wise feature correlation is defined as

$$\omega(y_i, y_j) = \begin{cases} c, & y_i = y_j; \\ d, & y_i \neq y_j, h(y_i) = h(y_j); \\ e, & h(y_i) \neq h(y_j) \end{cases} \tag{52}$$

for $c > d \geq e \geq 0$. Under this model, the class-wise feature correlation matrix $\mathbf{W}$ is expressed as:

$$\mathbf{W} = e\mathbf{1}_{K\times K} + (d-e)\mathbf{Q}^\top\mathbf{Q} + (c-d)\mathbf{I}_K, \tag{53}$$

and the Gram matrix $\mathbf{\Phi}$ is given by:

$$\mathbf{\Phi} = \mathbf{P}^\top\mathbf{W}\mathbf{P} + \mathbf{D}([1 - [\mathbf{W}]_{i,i}]_{i=1}^K\mathbf{P}) = \mathbf{P}^\top\mathbf{W}\mathbf{P} + (1-c)\mathbf{I}_{Kn}$$

Then, the label averaging matrix of the $\mathbf{\Phi}^{(t)}$ can be formulated as:

$$\mathbf{\Phi}^{(t)} = \left(\mathbf{I}_{Kn} - \left(\mathbf{I}_{Kn} + \frac{1}{K^2 n\lambda}\mathbf{\Phi}\right)^{-1}\right)^t$$

$$= \left(\mathbf{I}_{Kn} - \left(\frac{K^2 n\lambda + 1 - c}{K^2 n\lambda}\mathbf{I}_{Kn} + \frac{1}{K^2 n\lambda}\mathbf{P}^\top\mathbf{W}\mathbf{P}\right)^{-1}\right)^t$$

Since $\mathbf{P}\mathbf{P}^\top = n\mathbf{I}_{Kn}$, the following matrix inversion holds for any $\mathbf{U} \in \mathbb{R}^{K\times K}$ and $v \in R$:

$$\left(v\mathbf{I}_{Kn} + \frac{1}{n}\mathbf{P}^\top\mathbf{U}\mathbf{P}\right)^{-1} = \frac{1}{v}\mathbf{I}_{Kn} + \frac{1}{n}\mathbf{P}^\top\left[(v\mathbf{I}+\mathbf{U})^{-1} - \frac{1}{v}\mathbf{I}\right]\mathbf{P}.$$

Thus, we have

$$\left(\frac{K^2 n\lambda + 1 - c}{K^2 n\lambda}\mathbf{I}_{Kn} + \frac{1}{K^2 n\lambda}\mathbf{P}^\top\mathbf{W}\mathbf{P}\right)^{-1} = \frac{K^2 n\lambda}{K^2 n\lambda + 1 - c}\mathbf{I}_{Kn} + \frac{1}{n}\mathbf{P}^\top\left[\mathbf{V}^{-1} - \frac{K^2 n\lambda}{K^2 n\lambda + 1 - c}\mathbf{I}\right]\mathbf{P}$$

where $\mathbf{V} \in \mathbb{R}^{K\times K}$ is given by:

$$\mathbf{V} = \frac{K^2 n\lambda + 1 - c}{K^2 n\lambda}\mathbf{I}_K + \frac{\mathbf{W}}{K^2\lambda}.$$

Substituting back into $\mathbf{\Phi}^{(t)}$, we get:

$$\mathbf{\Phi}^{(t)} = \left(\left(1 - \frac{K^2 n\lambda}{K^2 n\lambda + 1 - c}\right)\mathbf{I}_{Kn} + \frac{1}{n}\mathbf{P}^\top\left[\frac{K^2 n\lambda}{K^2 n\lambda + 1 - c}\mathbf{I} - \mathbf{V}^{-1}\right]\mathbf{P}\right)^t,$$

*CLAIM:* For any $t \geq 1$, $\mathbf{U} \in \mathbb{R}^{K\times K}$, and $v \in \mathbb{R}$,

$$\left(\frac{1}{n}\mathbf{P}^\top\mathbf{U}\mathbf{P} + v\mathbf{I}_{Kn}\right)^t = \frac{1}{n}\mathbf{P}^\top[(\mathbf{U}+v\mathbf{I}_K)^t - v^t\mathbf{I}_K]\mathbf{P} + v^t\mathbf{I}_{Kn}. \tag{54}$$

*Proof.* Since $\mathbf{P}\mathbf{P}^\top = n\mathbf{I}_{Kn}$, we have:

$$\left(\frac{1}{n}\mathbf{P}^\top\mathbf{U}\mathbf{P} + v\mathbf{I}_{Kn}\right)^t = v^t\mathbf{I}_{Kn} + \sum_{i=1}^t \binom{t}{i}v^i\left(\frac{1}{n}\mathbf{P}^\top\mathbf{U}\mathbf{P}\right)^{t-i}$$

$$= v^t\mathbf{I}_{Kn} + \frac{1}{n}\sum_{i=1}^t\binom{t}{i}v^i\mathbf{P}^\top\mathbf{U}^{t-i}\mathbf{P}$$

$$= v^t\mathbf{I}_{Kn} + \frac{1}{n}\mathbf{P}^\top\left[\sum_{i=1}^t\binom{t}{i}v^i\mathbf{U}^{t-i}\right]\mathbf{P}$$

$$= v^t\mathbf{I}_{Kn} + \frac{1}{n}\mathbf{P}^\top[(\mathbf{U}+v\mathbf{I}_K)^t - v^t\mathbf{I}_K]\mathbf{P}.$$

$\square$

From (54), we can express $\boldsymbol{\Phi}^{(t)}$ as

$$\boldsymbol{\Phi}^{(t)} = \left(1 - \frac{K^2n\lambda}{K^2n\lambda + 1 - c}\right)^t\mathbf{I}_{Kn} + \frac{1}{n}\mathbf{P}^\top\left[(\mathbf{I} - \mathbf{V}^{-1})^t - \left(1 - \frac{K^2n\lambda}{K^2n\lambda + 1 - c}\right)^t\mathbf{I}_K\right]\mathbf{P}. \tag{55}$$

The main problem is to calculate the inverse of $\mathbf{V}$, which can be written as

$$\mathbf{V}^{-1} = \left[\frac{K^2n\lambda + 1 - c + n(c - d)}{K^2n\lambda}\mathbf{I}_K + \frac{e}{K^2\lambda}\mathbf{1}_{K\times K} + \frac{d - e}{K^2\lambda}\mathbf{Q}^\top\mathbf{Q}\right]^{-1} \tag{56}$$

By simple matrix calculations, we have

$$\mathbf{1}_{K\times K}\mathbf{1}_{K\times K} = K\mathbf{1}_{K\times K};$$
$$\mathbf{1}_{K\times K}\mathbf{Q}^\top\mathbf{Q} = \mathbf{Q}^\top\mathbf{D}([K_i]_{i=1}^R)\mathbf{Q};$$
$$\mathbf{Q}^\top\mathbf{Q}\mathbf{Q}^\top\mathbf{Q} = \mathbf{Q}^\top\mathbf{D}([K_i]_{i=1}^R)\mathbf{Q}.$$

Hence, the following matrix inversion holds true for the corresponding $\alpha > 0$, $\beta$ and $\gamma$ of (56):

$$\left(\alpha\mathbf{I}_K + \beta\mathbf{1}_{K\times K} + \gamma\mathbf{Q}^\top\mathbf{Q}\right)^{-1} = \left(\frac{1}{\alpha}\mathbf{I}_K - \frac{\beta}{\alpha(\alpha + K\beta)}\mathbf{1}_{K\times K} - \mathbf{Q}^\top\mathbf{D}([\delta_i]_{i=1}^R)\mathbf{Q}\right),$$

where $\delta_i$ is given as

$$\delta_i = \frac{1}{\alpha + K_i\beta + K_i\gamma}\left(\frac{\gamma}{\alpha} - \frac{K_i\beta\gamma}{\alpha(\alpha + K\beta)}\right).$$

Define $p$, $q$, $s$ and $\mathbf{r} = [r_1, \ldots, r_R] \in \mathbb{R}^R$ according to the following equations:

$$p = \frac{1 - c}{K^2n\lambda + 1 - c};$$

$$q = \frac{1 - c + n(c - d)}{K^2n\lambda + 1 - c + n(c - d)};$$

$$r_i = \frac{1 - c + n(c - d) + K_ind}{K^2n\lambda + 1 - c + n(c - d) + K_ind}, \quad i \in [R];$$

$$s = \frac{1 - c + n(c - d) + Kne}{K^2n\lambda + 1 - c + n(c - d) + Kne}.$$

Then, the coefficients for $\mathbf{I}_K$, $\mathbf{1}_{K\times K}$, $\mathbf{1}_{K_i\times K_i}$ of $\mathbf{V}^{-1}$ in (56) can be written, respectively, as follows:

$$\frac{1}{\alpha} = 1 - q,$$

$$-\frac{\beta}{\alpha(\alpha + K\beta)} = \frac{1}{K}\left(\frac{1}{\alpha + K\beta} - \frac{1}{\alpha}\right) = \frac{s - q}{K},$$

and

$$
\begin{aligned}
\delta_i &= \frac{1}{\alpha + K_i\beta + K_i\gamma}\left(-\frac{\gamma}{\alpha} + \frac{K_i\beta\gamma}{\alpha(\alpha + K\beta)}\right)\\
&= \frac{1}{\alpha + K_i\beta + K_i\gamma}\frac{\gamma(\alpha + K\beta - K_i\beta)}{\alpha(\alpha + K\beta)}\\
&= \frac{\gamma}{K}\frac{1}{\alpha + K_i\beta + K_i\gamma}\left(\frac{K - K_i}{\alpha} + \frac{K_i}{\alpha + K\beta}\right)\\
&= \frac{d - e}{K^3\lambda}\frac{K^2 n\lambda}{K^2 n\lambda + 1 - c + n(c - d) + K_i nd}\\
&\quad \times \left(\frac{(K - K_i)K^2 n\lambda}{K^2 n\lambda + 1 - c + n(c - d)} + \frac{K_i(K^2 n\lambda)}{K^2 n\lambda + 1 - c + n(c - d) + Kne}\right)\\
&= \frac{(K - K_i)(d - e)}{KK_i d}(r_i - q) + \frac{K_i(d - e)}{K(K_i d - Ke)}(r_i - s).
\end{aligned}
$$

Substituting the expanded form of $\mathbf{V}^{-1}$, we get:

$$
\begin{aligned}
\mathbf{\Phi}^{(t)} &= \mathbf{p}^t\mathbf{I}_{Kn} + \frac{1}{n}\mathbf{P}^\top\left[(\mathbf{I} - \mathbf{V}^{-1})^t - p^t\mathbf{I}_K\right]\mathbf{P}\\
&= p^t\mathbf{I}_{Kn} + \frac{1}{n}\mathbf{P}^\top\left(\left(q\mathbf{I}_K + \frac{q - s}{K}\mathbf{1}_{K\times K} + \mathbf{Q}^\top\mathbf{D}([\delta_i]_{i=1}^R)\mathbf{Q}\right)^t - p^t\mathbf{I}_K\right)\mathbf{P}.
\end{aligned}
$$

Also, for $j \in \mathbb{Z}^+$, the following holds:

$$
\begin{aligned}
&\left(\frac{q - s}{K}\mathbf{1}_{K\times K} + \mathbf{Q}^\top\mathbf{D}([\delta_i\mathbf{Q}\mathbf{1}_{K_i\times K_i}]_{i=1}^R\mathbf{Q}\right)^j\\
&= \frac{(q - s)^j}{K}\mathbf{1}_{K\times K} + \sum_{k=1}^j\binom{j}{k}\left(\frac{q - s}{K}\mathbf{1}_{K\times K}\right)^{j-k}\left(\mathbf{Q}^\top\mathbf{D}([\delta_i]_{i=1}^R)\mathbf{Q}\right)^k\\
&= \frac{(q - s)^j}{K}\mathbf{1}_{K\times K} + \sum_{k=1}^j\binom{j}{k}\left(\frac{q - s}{K}\mathbf{Q}^\top\mathbf{Q}\right)^{j-k}\left(\mathbf{Q}^\top\mathbf{D}([\delta_i]_{i=1}^R)\mathbf{Q}\right)^k\\
&= \frac{(q - s)^j}{K}\mathbf{1}_{K\times K} + \mathbf{Q}^\top\mathbf{D}\left(\left[K_i^{j-1}\left\{\left(\frac{q - s}{K} + \delta_i\right)^j - \left(\frac{q - s}{K}\right)^j\right\}\right]_{i=1}^R\right)\mathbf{Q}.
\end{aligned}
$$

Hence, we get

$$
\begin{aligned}
\left(q\mathbf{I}_K + \frac{q - s}{K}\mathbf{1}_{K\times K} + \mathbf{Q}^\top\mathbf{D}([\delta_i]_{i=1}^R)\mathbf{Q}\right)^t &= q^t\mathbf{I}_K + \sum_{j=1}^t\binom{t}{j}q^{t-j}\left(\frac{q - s}{K}\mathbf{1}_{K\times K} + \mathbf{D}([\delta_i\mathbf{1}_{K_i\times K_i}]_{i=1}^R)\right)^j\\
&= q^t\mathbf{I}_K + \nu(t)\mathbf{1}_{K\times K} + \mathbf{Q}^\top\mathbf{D}([\mu_i^{(t)}/K_i]_{i=1}^R)\mathbf{Q},
\end{aligned}
$$

where $\nu(t)$ and $\mu_i^{(t)}$ are defined as

$$
\begin{aligned}
\nu(t) &= \left(q + \frac{q - s}{K}\right)^t - q^t;\\
\mu_i^{(t)} &= \left\{q + K_i\left(\frac{q - s}{K} + \delta_i\right)\right\}^t - \left\{q + K_i\left(\frac{q - s}{K}\right)\right\}^t, \quad i \in [R].
\end{aligned}
$$

Finally, we get the closed-form of the label averaging matrix $\mathbf{\Phi}^{(t)}$ as

$$
\begin{aligned}
\mathbf{\Phi}^{(t)} &= p^t \mathbf{I}_{Kn} + \frac{1}{n}\mathbf{P}^\top \left( \left( q\mathbf{I}_K + \frac{q-s}{K}\mathbf{1}_{K\times K} + \mathbf{Q}^\top \mathbf{D}([\delta_i]_{i=1}^R)\mathbf{Q} \right)^t - p^t \mathbf{I}_K \right) \mathbf{P} \\
&= p^t \mathbf{I}_{Kn} + \frac{q^t - p^t}{n}\mathbf{P}^\top \mathbf{P} + \mathbf{P}^\top \mathbf{Q}^\top \mathbf{D}([\mu_i^{(t)}/K_i]_{i=1}^R)\mathbf{Q}\mathbf{P} + \nu(t)\mathbf{1}_{Kn\times Kn} \\
&= \underbrace{\mathbf{P}^\top \mathbf{D}\left( \left[ \frac{\mu_i^{(t)}}{K_i n}\mathbf{1}_{K_i\times K_i} \right]_{i=1}^R \right)\mathbf{P}}_{\text{superclass-wise label averaging matrix}} + \underbrace{\frac{q^t - p^t}{n}\mathbf{P}^\top \mathbf{P}}_{\text{class-wise label averaging matrix}} + p^t \mathbf{I}_{Kn} + \frac{\nu(t)}{n}\mathbf{1}_{Kn\times Kn}.
\end{aligned}
$$

Then, the closed-form of $\mathbf{y}_i^{(t)}$ for the extended feature map is given as follows:

$$
\mathbf{y}_i^{(t)} = p^t \mathbf{e}(\hat{y}_i) + (q^t - p^t)\left( \frac{1}{n}\sum_{\{j:y_i=y_j\}} \mathbf{e}(\hat{y}_j) \right) + \frac{\mu_s^{(t)}}{K_s n}\left( \sum_{\{j:h(y_j)=s\}} \mathbf{e}(\hat{y}_j) \right) + (1-\mu_s^{(t)}-q^t)\frac{1}{K}\mathbf{1}_K
$$

(57)

where $h(y_i) = s$. Note that when $e = 0$, $\nu(t)$ and $\mu_s^{(t)}$ are given as

$$
\nu(t) = 0, \quad \mu_s^{(t)} = r_s^t - q^t,
$$

which recovers the result in (51).

## K    PROOF OF THEOREM 4.1, THEOREM 5.1 AND COROLLARY C.1

In this section, we present the complete proofs of Theorem 4.1 and 5.1, and Corollary C.1. As in Sec. 4, we define the corruption matrix $\mathbf{C} \in \mathbb{R}^{K\times K}$ by

$$
[\mathbf{C}]_{k,k'} := \frac{|\{i : y_i = k, \hat{y}_i = k'\}|}{n}.
$$

Then, the mean of the given labels for samples that share the same true label, denoted as $\frac{1}{n}\sum_{\{j:y_i=y_j\}} \mathbf{e}(\hat{y}_j)$, corresponds to the $y_i$-th row of the matrix $\mathbf{C}$, expressed as:

$$
\frac{1}{n}\sum_{\{j:y_i=y_j\}} \mathbf{e}(\hat{y}_j) = \mathbf{C}[y_i, :]^\top.
$$

Here, we aim to find the conditions for the prediction vector $\mathbf{y}_i^{(t)}$ to have the maximum value at the true label position $y_i$, indicating a correct prediction. To achieve this, we use the closed-form solution (13) of $\mathbf{y}_i^{(t)}$, presented in Lemma 4.1.

### K.1    PROOF OF THEOREM 4.1

According to Lemma 4.1, the closed-form outputs of the $t$-th distilled model can be formulated as

$$
\mathbf{y}_i^{(t)} = p^t \mathbf{e}(\hat{y}_i) + (q^t - p^t)\left( \frac{1}{n}\sum_{\{j:y_j=y_i\}} \mathbf{e}(\hat{y}_j) \right) + (r_s^t - q^t)\left( \frac{1}{K_s n}\sum_{\{j:h(y_j)=s\}} \mathbf{e}(\hat{y}_j) \right) + \frac{(1-r_s^t)}{K}\mathbf{1}_K,
$$

where $s = h(y_i)$ is the superclass of $y_i$, $K_s$ is the size of the superclass $s$, and $p$, $q$, and $r_s$ are defined as in (14). Also, we assume that the label error happens only within each superclass, which implies that $[\mathbf{C}]_{k,k'} = 0$ if $h(k) \neq h(k')$. Remind that we assume that the dataset is balanced with respect to both the ground-truth labels and given labels, i.e., $|\{i : y_i = k\}| = |\{i : \hat{y}_i = k\}| = n$ for $\forall k \in [K]$. Then, we can simplify the third term, the mean of superclass labels, using the indicator function, denoted as $\mathbb{1}$, as follows:

$$
(r_s^t - q^t)\left( \frac{1}{K_s n}\sum_{\{j:h(y_j)=s\}} \mathbf{e}(\hat{y}_j) \right) = \frac{r_s^t - q^t}{K_s}[\mathbb{1}(h(1)=s), \dots, \mathbb{1}(h(K)=s)],
$$

(58)

which means that third term is uniform inside the superclass, while zero outside the superclass. Now, we will analyze the prediction of clean samples and label-noise samples.

For the clean samples, i.e., $\hat{y}_i = y_i$, the prediction of the $t$-th distilled model for the $i$-th sample can be written as

$$[\mathbf{y}_i^{(t)}]_k = \begin{cases} p^t + (q^t - p^t)[\mathbf{C}]_{y_i,y_i} + (r_s^t - q^t)/K_s + (1 - r_s^t)/K, & k = y_i; \\ (q^t - p^t)[\mathbf{C}]_{y_i,k} + (r_s^t - q^t)/K_s + (1 - r_s^t)/K, & k \neq y_i, h(k) = h(y_i); \\ (1 - r_s^t)/K. & h(k) \neq h(y_i). \end{cases}$$

Then, the condition for the correct classification of the clean sample $i$ is given by

$$[\mathbf{C}]_{y_i,y_i} > [\mathbf{C}]_{y_i,k} - \frac{1}{(q/p)^t - 1}, \quad \forall k \neq y_i. \tag{59}$$

For the noisy samples such that $\hat{y}_i \neq y_i$, the prediction of the $t$-th distilled model can be written as

$$[\mathbf{y}_i^{(t)}]_k = \begin{cases} (q^t - p^t)[\mathbf{C}]_{y_i,y_i} + (r_s^t - q^t)/K_s + (1 - r_s^t)/K, & k = y_i; \\ p^t + (q^t - p^t)[\mathbf{C}]_{y_i,\hat{y}_i} + (r_s^t - q^t)/K_s + (1 - r_s^t)/K, & k = \hat{y}_i; \\ (q^t - p^t)[\mathbf{C}]_{y_i,k} + (r_s^t - q^t)/K_s + (1 - r_s^t)/K, & k \neq y_i, \hat{y}_i, h(k) = h(y_i); \\ (1 - r_s^t)/K, & h(k) \neq h(y_i). \end{cases}$$

Thus, the condition for the correct classification of the noisy sample $i$ is given as

$$[\mathbf{C}]_{y_i,y_i} > [\mathbf{C}]_{y_i,\hat{y}_i} + \frac{1}{(q/p)^t - 1} \quad \text{and} \tag{60}$$

$$[\mathbf{C}]_{y_i,y_i} > [\mathbf{C}]_{y_i,k}, \quad \forall k \neq y_i, \hat{y}_i \tag{61}$$

Note that (59) is the easiest condition, and (60) is the most stringent condition. Thus, if

$$[\mathbf{C}]_{k,k} > [\mathbf{C}]_{k,k'} + 1/\left((q/p)^t - 1\right), \quad \forall k, k' \in [K] \text{ with } k \neq k' \tag{62}$$

then the $t$-th student model can correctly classify both the noisy samples and clean samples. On the other hand, if

$$[\mathbf{C}]_{k,k} < [\mathbf{C}]_{k,k'} + 1/\left((q/p)^{t-1} - 1\right), \tag{63}$$

then the $(t-1)$-th distilled model fails to correctly classify label-noisy samples from class $k$ that have been mislabelled as $\hat{y}_i = k'$, and thus fails to achieve 100% population accuracy.

To find the regimes where the student model outperforms the teacher model, we need to find $t \in \mathbb{Z}^+$ that satisfies

$$[\mathbf{C}]_{k,k'} + \frac{1}{(q/p)^{t-1} - 1} > [\mathbf{C}]_{k,k} > [\mathbf{C}]_{k,k'} + \frac{1}{(q/p)^t - 1}, \tag{64}$$

In such a $t$, the student model correctly classifies the samples whose true labels are $k$ and the given labels are $k'$, but the teacher model makes mistakes.

## K.2 Proof of Theorem 5.1

The closed form solution for the output of the PLL student model for the $i$-th training instance is given by

$$\mathbf{y}_i^{(P)} = p\bar{\mathbf{y}}_i + (q - p)\left(\frac{1}{n} \sum_{\{j:y_j=y_i\}} \bar{\mathbf{y}}_j\right) + (r_s - q)\left(\frac{1}{K_s n} \sum_{\{j:h(y_j)=s\}} \bar{\mathbf{y}}_j\right) + (1 - r_s)\frac{\mathbf{1}_K}{K}, \tag{65}$$

for $s = h(i)$. This can be derived from Lemma 4.1, by plugging in the refined teacher's output $\bar{\mathbf{y}}_i$ at the position of $\mathbf{e}(\hat{y}_i)$ and considering the one more step of distillation $t = 1$. Here, $\bar{\mathbf{y}}_i$ is the two-hot vector with weights $1/2$, at the positions of top two labels with the highest values in the teacher's predictions.

Suppose that $[\mathbf{C}]_{y_i,y_i} > [\mathbf{C}]_{y_i,k}$ holds for all $k \neq y_i$. From the conditions (59) and (61), the true label is always included in the top-two candidate set. Let us define $\tilde{y}_i$ as

$$\tilde{y}_i = \arg\max_{k \neq y_i}[\mathbf{C}]_{y_i,k}. \tag{66}$$

Note that $\tilde{y}_i$ is always in the superclass of $h(y_i)$. Then, the equation for $\bar{\mathbf{y}}_i$ can be expressed as:

$$\bar{\mathbf{y}}_i = \frac{1}{2}\mathbf{e}(y_i) + \frac{1}{2}\mathbf{e}(\tilde{y}_i) \tag{67}$$

for the clean sample, and

$$\bar{\mathbf{y}}_i = \frac{1}{2}\mathbf{e}(y_i) + \frac{1}{2}\mathbf{e}(\hat{y}_i) \tag{68}$$

for the noisy sample. Then, $\frac{1}{n}\sum_{\{j:y_j=y_i\}}\bar{\mathbf{y}}_j$ and $\frac{1}{K_s n}\sum_{\{j:h(y_j)=s\}}\bar{\mathbf{y}}_j$ are given as

$$\frac{1}{n}\sum_{\{j:y_j=y_i\}}\bar{\mathbf{y}}_j = \frac{1}{2}\mathbf{e}(y_i) + \frac{1}{2}[\mathbf{C}]_{y_i,y_i}\mathbf{e}(\tilde{y}_i) + \frac{1}{2}\sum_{j \neq y_i}[\mathbf{C}]_{y_i,j}\mathbf{e}(j),$$

and

$$\frac{1}{K_s n}\sum_{\{j:h(y_j)=s\}}\bar{\mathbf{y}}_j = \frac{1}{K_s}[\mathbb{1}(h(1)=s),\ldots,\mathbb{1}(h(K)=s)],$$

since the true labels and the given labels are balanced. Hence, for the clean sample, the prediction of the PLL student model for the $i$-th sample can be written as

$$[\mathbf{y}_i^{(P)}]_k = \begin{cases} \frac{p}{2} + \frac{q-p}{2} + \frac{r_s-q}{K_s} + \frac{1-r_s}{K}, & k = y_i; \\ \frac{p}{2} + \frac{q-p}{2}([\mathbf{C}]_{y_i,y_i} + [\mathbf{C}]_{y_i,\tilde{y}_i}) + \frac{r_s-q}{K_s} + \frac{1-r_s}{K}, & k = \tilde{y}_i; \\ \frac{q-p}{2}[\mathbf{C}]_{y_i,k} + \frac{r_s-q}{K_s} + \frac{1-r_s}{K} & k \neq y_i, \tilde{y}_i, h(k) = h(y_i); \\ \frac{1-r_s}{K}, & h(k) \neq h(y_i). \end{cases}$$

Here, $[\mathbf{y}_i^{(P)}]_k$ has the maximum value at $k = y_i$, because $[\mathbf{C}]_{y_i,y_i} + [\mathbf{C}]_{y_i,\tilde{y}_i}$ is less than or equal to 1. For the noisy sample, when $\hat{y}_i = \tilde{y}_i$, we have

$$[\mathbf{y}_i^{(P)}]_k = \begin{cases} \frac{p}{2} + \frac{q-p}{2} + \frac{r_s-q}{K_s} + \frac{1-r_s}{K}, & k = y_i; \\ \frac{p}{2} + \frac{q-p}{2}([\mathbf{C}]_{y_i,y_i} + [\mathbf{C}]_{y_i,\tilde{y}_i}) + \frac{r_s-q}{K_s} + \frac{1-r_s}{K}, & k = \hat{y}_i; \\ \frac{q-p}{2}[\mathbf{C}]_{y_i,k} + \frac{r_s-q}{K_s} + \frac{1-r_s}{K} & k \neq y_i, \hat{y}_i, h(k) = h(y_i); \\ \frac{1-r_s}{K}, & h(k) \neq h(y_i). \end{cases}$$

On the other hand, when $\hat{y}_i \neq \tilde{y}_i$, we have

$$[\mathbf{y}_i^{(P)}]_k = \begin{cases} \frac{p}{2} + \frac{q-p}{2} + \frac{r_s-q}{K_s} + \frac{1-r_s}{K}, & k = y_i; \\ \frac{p}{2} + \frac{q-p}{2}[\mathbf{C}]_{y_i,\hat{y}_i} + \frac{r_s-q}{K_s} + \frac{1-r_s}{K}, & k = \hat{y}_i; \\ \frac{q-p}{2}([\mathbf{C}]_{y_i,y_i} + [\mathbf{C}]_{y_i,\tilde{y}_i}) + \frac{r_s-q}{K_s} + \frac{1-r_s}{K}, & k = \tilde{y}_i; \\ \frac{q-p}{2}[\mathbf{C}]_{y_i,k} + \frac{r_s-q}{K_s} + \frac{1-r_s}{K} & k \neq y_i, \hat{y}_i, \tilde{y}_i, h(k) = h(y_i); \\ \frac{1-r_s}{K}, & h(k) \neq h(y_i). \end{cases}$$

Therefore, the prediction of the PLL student model has the highest probability at the true label $y_i$ for all noisy samples as well.

Throughout all the cases, it can be concluded that the PLL model consistently achieves 100% population accuracy when the PLL candidate sets of size two always contain the true label. Note that the condition for this situation is given by (15).

To summarize, the $t$-th distilled model attains 100% population accuracy when

$$[\mathbf{C}]_{k,k} > [\mathbf{C}]_{k,k'} + 1/\left((q/p)^t - 1\right), \quad \forall k, k' \in [K] \text{ with } k \neq k'$$

holds. For the PLL student model, the condition for achieving 100% population accuracy is given by

$$[\mathbf{C}]_{k,k} > [\mathbf{C}]_{k,k'}, \quad \forall k, k' \in [K] \text{ with } k \neq k'$$

which implies that the PLL student model outperforms any $t$-th distilled model in the population accuracy.

### K.3 Proof of Corollary C.1

Suppose that the class-wise feature correlation at the $t$-distillation round is given by

$$[\mathbf{\Phi}'(t)]_{i,j} := \langle \phi(\mathbf{x}_i), \phi(\mathbf{x}_j) \rangle = \begin{cases} 1, & i = j; \\ c^{(t)}, & y_i = y_j, i \neq j; \\ d^{(t)}, & y_i \neq y_j, h(y_i) = h(y_j); \\ 0, & h(y_i) \neq h(y_j). \end{cases} \tag{69}$$

Then, following the similar analysis as in Sec. J.4, the label averaging matrix $\mathbf{\Phi}^{(t)} := \left[ \mathbf{I}_{Kn} - \left( \mathbf{I}_{Kn} + \mathbf{\Phi}'(t)/(K^2 n\lambda) \right)^{-1} \right]$ at the $t$-th distillation round can be expressed as follows:

$$\mathbf{\Phi}^{(t)} = \frac{1}{n} \mathbf{P}^\top \left[ (q^{(t)} - p^{(t)}) \mathbf{I}_K + \mathbf{D} \left( \left[ \frac{r_s^{(t)} - q^{(t)}}{K_s} \mathbf{1}_{K_s \times K_s} \right]_{s=1}^R \right) \right] \mathbf{P} + p^{(t)} \mathbf{I}_{Kn}, \tag{70}$$

where $p^{(t)}, q^{(t)}, r_s^{(t)}$ are defined as

$$p^{(t)} = \frac{1 - c^{(t)}}{K^2 n\lambda + 1 - c^{(t)}};$$

$$q^{(t)} = \frac{1 - c^{(t)} + n(c^{(t)} - d^{(t)})}{K^2 n\lambda + 1 - c^{(t)} + n(c^{(t)} - d^{(t)})};$$

$$r_s^{(t)} = \frac{1 - c^{(t)} + n(c^{(t)} - d^{(t)}) + K_s n d^{(t)}}{K^2 n\lambda + 1 - c^{(t)} + n(c^{(t)} - d^{(t)}) + K_s n d^{(t)}}.$$

For simplicity in notation, define $\mathcal{D} : \mathbb{R}^R \to \mathbb{R}^{K \times K}$ as

$$\mathcal{D}(\mathbf{u}) = \mathbf{D} \left( \left[ \frac{u_s}{K_s} \mathbf{1}_{K_s \times K_s} \right]_{s=1}^R \right). \tag{71}$$

*CLAIM.* For any $t \geq 1$,

$$\prod_{i=1}^t \mathbf{\Phi}^{(i)} = \frac{1}{n} \mathbf{P}^\top \left[ (\prod_{i=1}^t q^{(i)} - \prod_{i=1}^t p^{(i)}) \mathbf{I}_K + \mathcal{D} \left( \left[ \prod_{i=1}^t r_s^{(i)} - \prod_{i=1}^t q^{(i)} \right]_{s=1}^R \right) \right] \mathbf{P} + \prod_{i=1}^t p^{(i)} \mathbf{I}_{Kn},$$

*Proof.* When $t = 1$, the above claim holds trivially. Next, suppose our claim holds true for $t = k$. Then, we have

$$\prod_{i=1}^k \mathbf{\Phi}^{(i)} = \frac{1}{n} \mathbf{P}^\top \left[ (\prod_{i=1}^k q^{(i)} - \prod_{i=1}^k p^{(i)}) \mathbf{I}_K + \mathcal{D} \left( \left[ \prod_{i=1}^k r_s^{(i)} - \prod_{i=1}^k q^{(i)} \right]_{s=1}^R \right) \right] \mathbf{P} + \prod_{i=1}^k p^{(i)} \mathbf{I}_{Kn}$$

and

$$\mathbf{\Phi}^{(k+1)} = \frac{1}{n} \mathbf{P}^\top \left[ (q^{(k+1)} - p^{(k+1)}) \mathbf{I}_K + \mathcal{D} \left( [r_s^{(k+1)} - q^{(k+1)}]_{s=1}^R \right) \right] \mathbf{P} + p^{(k+1)} \mathbf{I}_{Kn}.$$

Since $\mathbf{P}\mathbf{P}^\top = n\mathbf{I}_K$ and $\mathcal{D}(\mathbf{u})\mathcal{D}(\mathbf{v}) = \mathcal{D}(\mathbf{u} \circ \mathbf{v})$, we have

$$\prod_{i=1}^{k+1} \mathbf{\Phi}^{(i)} = \frac{1}{n} \mathbf{P}^\top \left[ a\mathbf{I}_K + \mathcal{D}([b_i]_{i=1}^R) \right] \mathbf{P} + \prod_{i=1}^{k+1} p^{(i)} \mathbf{I}_{Kn},$$

for some $a$ and $b_i$. Then, $a$ and $b_i$ can be computed as following:

$$a = (\prod_{i=1}^k q^{(i)} - \prod_{i=1}^k p^{(i)})(q^{(k+1)} - p^{(k+1)}) + p^{(k+1)}(\prod_{i=1}^k q^{(i)} - \prod_{i=1}^k p^{(i)}) + \prod_{i=1}^k p^{(i)}(q^{(k+1)} - p^{(k+1)})$$

$$= \prod_{i=1}^{k+1} q^{(i)} - \prod_{i=1}^{k+1} p^{(i)},$$

and

$$
\begin{aligned}
b_i ={}& \left(r_s^{(k+1)} - q^{(k+1)}\right)\left(\prod_{i=1}^{k} r_s^{(i)} - \prod_{i=1}^{k} q^{(i)}\right) + \left(q^{(k+1)} - p^{(k+1)}\right)\left(\prod_{i=1}^{k} r_s^{(i)} - \prod_{i=1}^{k} q^{(i)}\right) \\
&+ \left(r_s^{(k+1)} - q^{(k+1)}\right)\left(\prod_{i=1}^{k} q^{(i)} - \prod_{i=1}^{k} p^{(i)}\right) + \quad + p^{(k+1)}\left(\prod_{i=1}^{k} r_s^{(i)} - \prod_{i=1}^{k} q^{(i)}\right) \\
&+ \left(r_s^{(k+1)} - q^{(k+1)}\right)\prod_{i=1}^{k} p^{(i)} \\
={}& \prod_{i=1}^{k+1} r_s^{(i)} - \prod_{i=1}^{k+1} q^{(i)}.
\end{aligned}
$$

Thus, by the mathematical induction, our claim holds true for all $t \geq 1$. $\qquad\square$

From our claim, we can derive a closed-form solution of $\mathbf{y}_i^{(t)}$ in the evolving feature map (69) as follows:

$$
\begin{aligned}
\mathbf{y}_i^{(t)} ={}& \prod_{i=1}^{t} p^{(i)} \mathbf{e}(\hat{y}_i) + \left(\prod_{i=1}^{t} q^{(i)} - \prod_{i=1}^{t} p^{(i)}\right)\left(\frac{1}{n} \sum_{\{j:y_i=y_j\}} \mathbf{e}(\hat{y}_j)\right) \\
&+ \left(\prod_{i=1}^{t} r_s^{(i)} - \prod_{i=1}^{t} q^{(i)}\right)\left(\frac{1}{K_s n} \sum_{\{j:h(y_j)=s\}} \mathbf{e}(\hat{y}_j)\right) + \left(1 - \prod_{i=1}^{t} r_s^{(i)}\right)\frac{1}{K}\mathbf{1}_K
\end{aligned}
\tag{72}
$$

The only difference from the original equation is that $p^t$, $q^t$, and $r_s^t$ have changed to $\prod_{i=1}^{t} p^{(i)}$, $\prod_{i=1}^{t} q^{(i)}$, and $\prod_{i=1}^{t} r_s^{(i)}$, respectively. Hence, Corollary C.1 holds.

# L ADDITIONAL EXPERIMENTAL RESULTS

## L.1 ABLATION STUDY: VARYING WEIGHT-DECAY VALUE $\lambda$

We perform an ablation study for $\lambda$ values that make $q/p$ near 2 as explained in Appendix E.2. For these $\lambda$ values, we observe that there is a consistent accuracy gain as the distillation round $t$ increases. Also, we observe that our PLL student model outperforms the multi-round SD models.

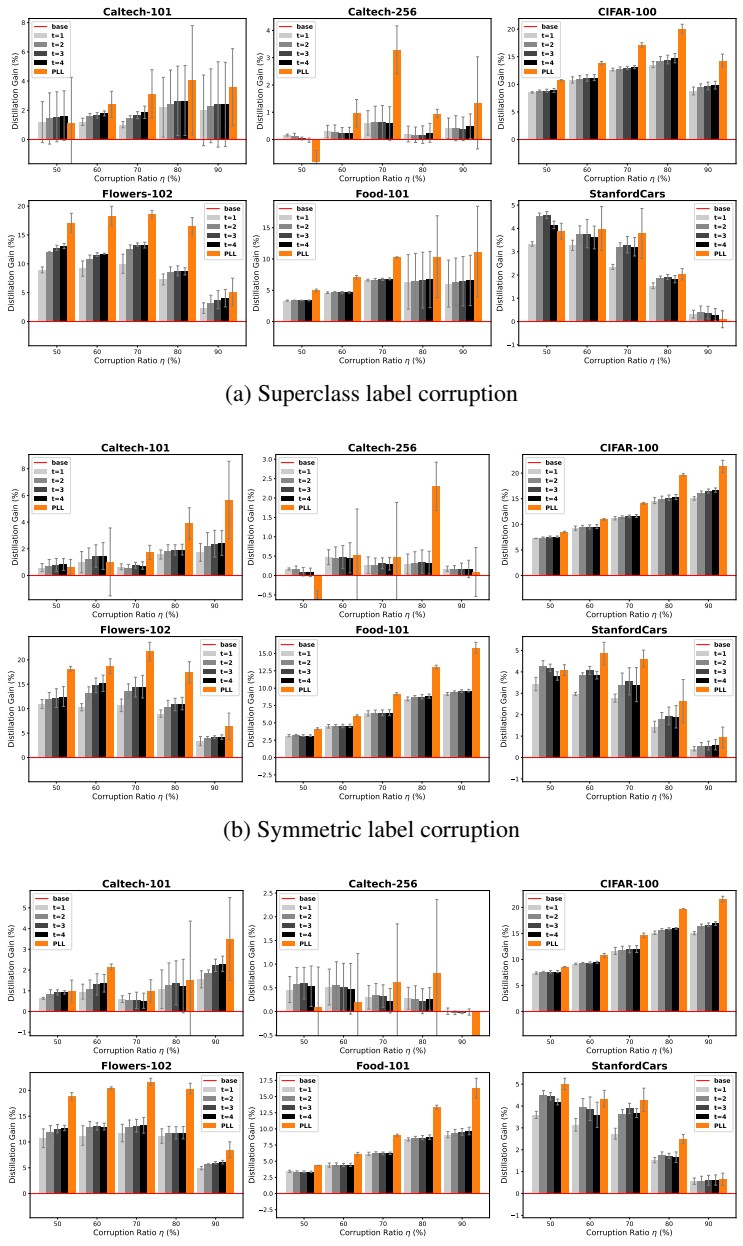

(a) Superclass label corruption

(b) Symmetric label corruption

(c) Asymmetric label corruption

Figure 14: Accuracy gap for each student model compared to the teacher model in three different label corruption scenarios. PLAIN represents the gap between multi-round SD students and the teacher model, while ORANGE represents the gap between PLL students and the teacher model. We use weight decay value $\lambda = 2 \times 10^{-4}$ for Caltech-101, Flowers-102, and StanfordCars datasets, and $\lambda = 2 \times 10^{-5}$ for CIFAR-100, Caltech-256, and Food-101 datasets.

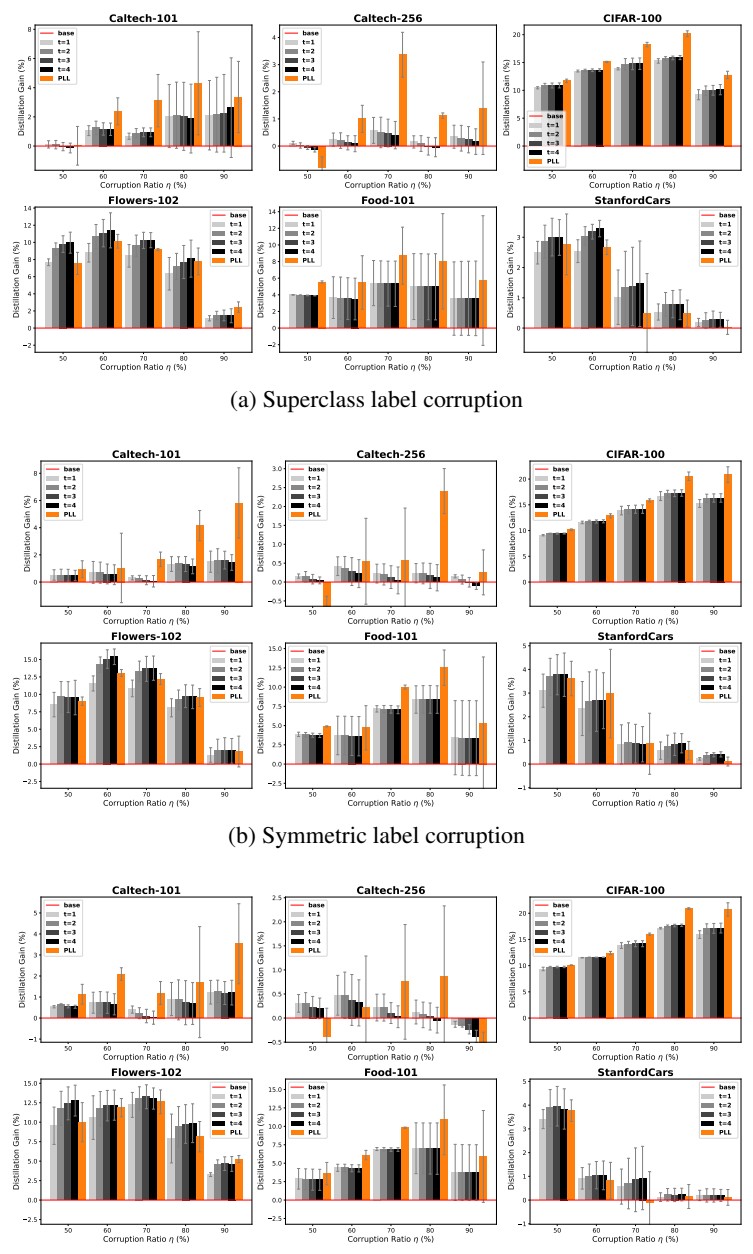

Figure 15: Accuracy gap for each student model compared to the teacher model in a three label corruption scenarios. PLAIN represents the gap between multi-round SD students and the teacher model, while ORANGE represents the gap between PLL students and the teacher model. We use weight decay value $\lambda = 1 \times 10^{-3}$ for Caltech-101, Flowers-102, and StanfordCars datasets, and $\lambda = 1 \times 10^{-4}$ for CIFAR-100, Caltech-256, and Food-101 datasets.

## L.2 Hierarchical Clustering Results

We conduct hierarchical clustering to estimate the super-class on six different real datasets. The figures below show the hierarchical clustering results of each dataset. Clustering is performed to place similar classes adjacent to each other.

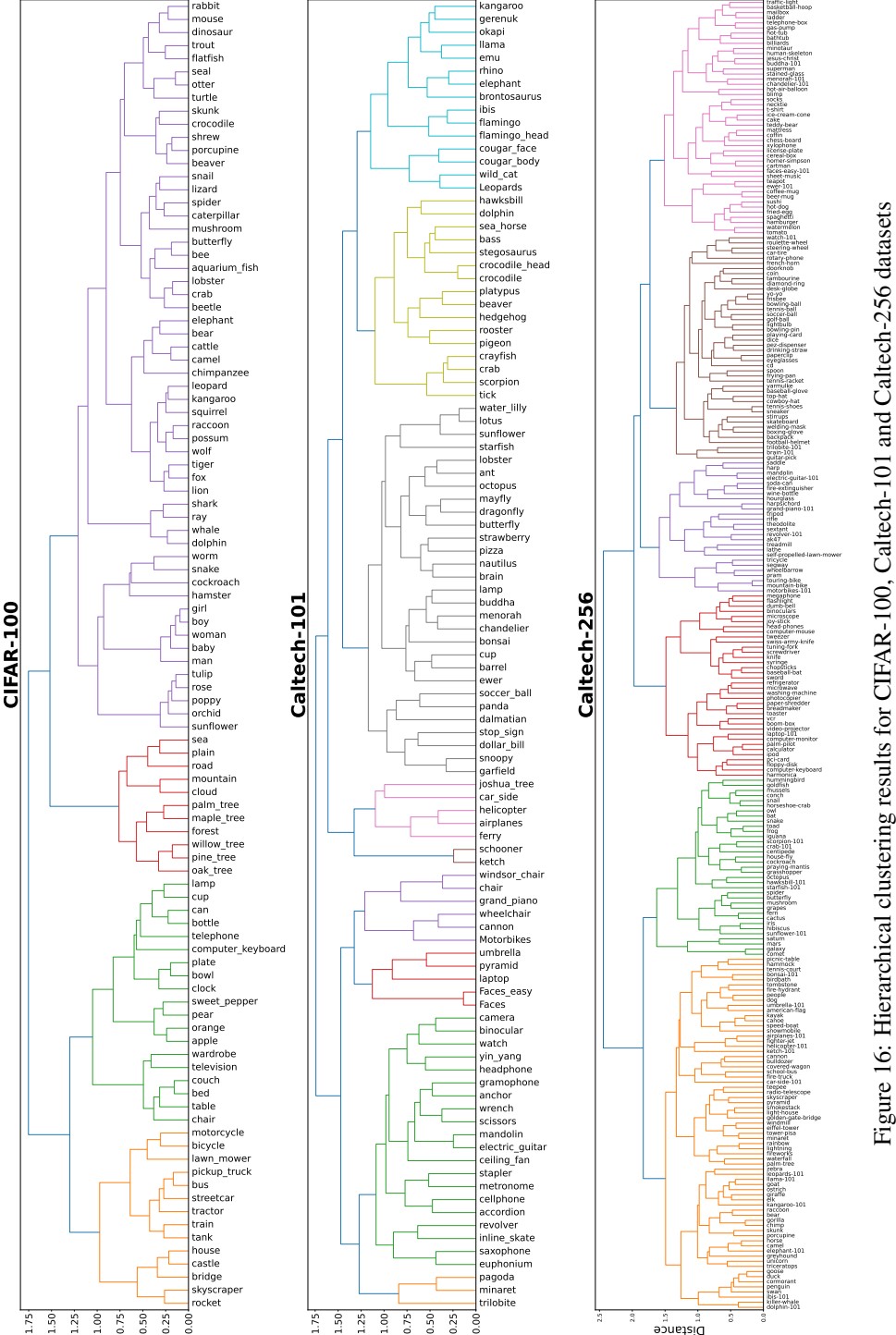

Figure 16: Hierarchical clustering results for CIFAR-100, Caltech-101 and Caltech-256 datasets

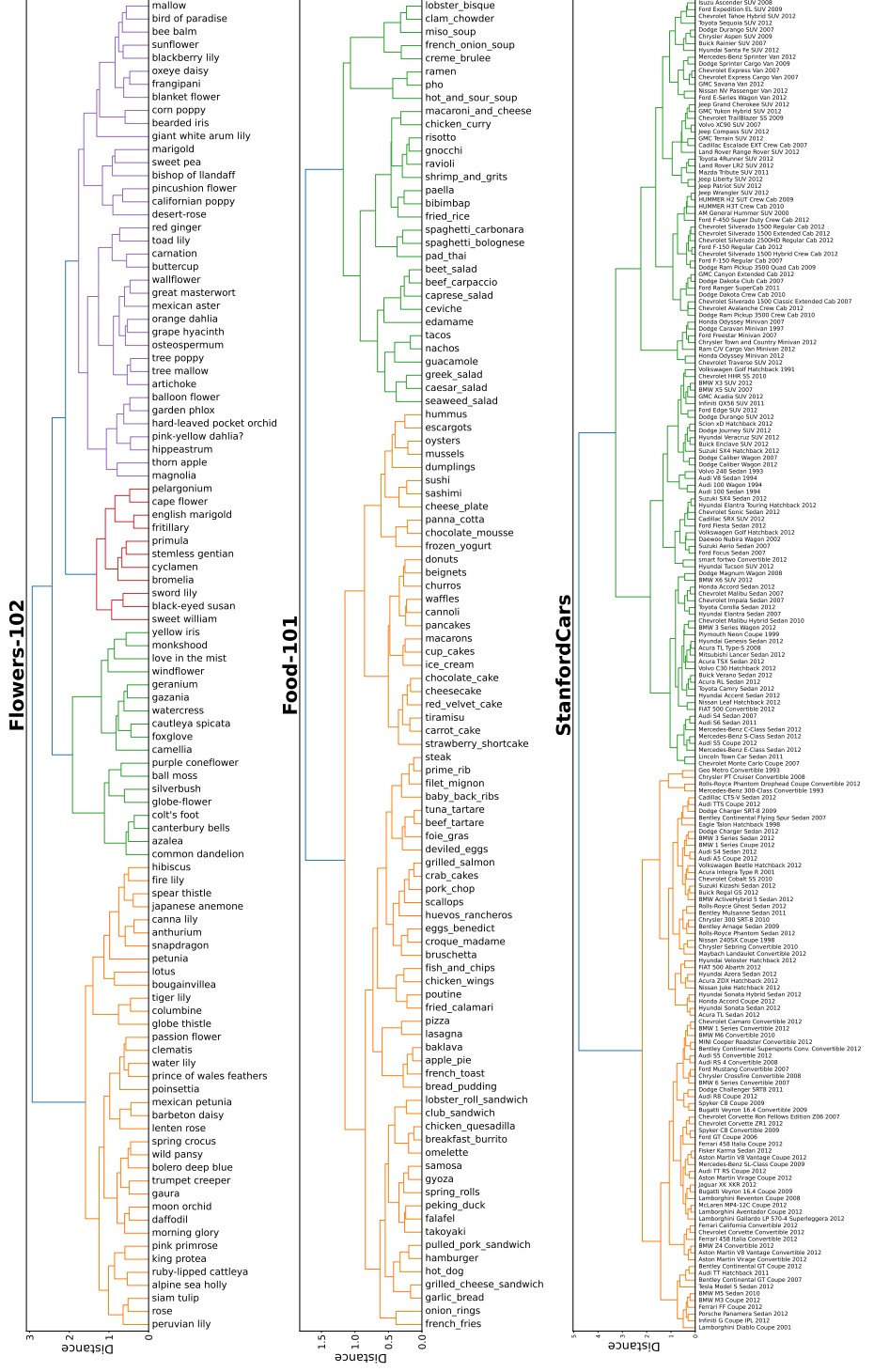

Figure 17: Hierarchical clustering results for Flowers-102, Food-101 and StanfordCars datasets

# M   DETAILED EMPIRICAL RESULTS

In this section, we report the full results of the experiments presented in the Sec. 6, Appendix F and L.1.

Table 3: (Details of Fig. 9a) Test accuracy of Caltech-101 dataset applying superclass label corruption, where weight decay value $\lambda = 5 \times 10^{-4}$.

| Distillation Step | 0.0 | 0.1 | 0.3 | 0.5 | 0.6 | 0.7 | 0.8 | 0.9 |
|---|---|---|---|---|---|---|---|---|
| 1 (Teacher) | 95.03±0.07% | 93.47±0.22% | 90.40±0.35% | 84.31±0.86% | 78.76±0.89% | 67.80±1.21% | 47.35±1.77% | 16.38±0.98% |
| 2 | **95.10±0.05%** | **93.74±0.20%** | **90.94±0.24%** | 85.46±0.85% | 79.88±0.65% | 68.59±1.28% | 47.83±2.22% | 18.45±1.42% |
| 3 | 95.05±0.09% | 93.74±0.24% | 90.84±0.37% | **85.60±1.01%** | 80.15±0.56% | 68.87±1.28% | 47.85±2.20% | 18.66±1.59% |
| 4 | 94.97±0.17% | 93.74±0.20% | 90.92±0.35% | 85.54±1.16% | 80.05±0.52% | 69.01±1.31% | 48.04±2.13% | 18.63±1.78% |
| 5 | 94.85±0.15% | 93.68±0.12% | 90.88±0.26% | 85.60±1.16% | 80.15±0.52% | 69.01±1.12% | 47.96±2.15% | 18.59±1.81% |
| PLL | 94.03±0.19% | 92.67±0.41% | 89.69±0.53% | 85.54±2.30% | **81.11±0.77%** | **70.79±1.11%** | **48.58±2.40%** | **19.78±1.45%** |

Table 4: (Details of Fig. 9a) Test accuracy of Caltech-256 dataset applying superclass label corruption, where weight decay value $\lambda = 5 \times 10^{-5}$.

| Distillation Step | 0.0 | 0.1 | 0.3 | 0.5 | 0.6 | 0.7 | 0.8 | 0.9 |
|---|---|---|---|---|---|---|---|---|
| 1 (Teacher) | 83.63±0.03% | 81.70±0.22% | 77.96±0.25% | 72.61±0.34% | 69.57±0.05% | 63.68±0.58% | 52.63±0.74% | 28.35±1.55% |
| 2 | **83.67±0.07%** | **81.98±0.15%** | **78.16±0.17%** | **72.66±0.30%** | 69.79±0.17% | 63.93±0.69% | 52.97±0.93% | 28.67±1.82% |
| 3 | 83.60±0.08% | 81.95±0.15% | 78.12±0.11% | 72.56±0.41% | **69.83±0.23%** | 63.89±0.68% | 52.91±0.98% | 28.60±1.83% |
| 4 | 83.53±0.08% | 81.88±0.14% | 78.02±0.09% | 72.53±0.39% | 69.80±0.24% | 63.86±0.65% | 52.92±0.95% | 28.56±1.88% |
| 5 | 83.53±0.05% | 81.83±0.21% | 77.93±0.10% | 72.42±0.39% | 69.80±0.25% | 63.86±0.65% | 52.88±0.90% | 28.50±1.86% |
| PLL | 82.64±0.24% | 80.23±0.29% | 76.08±1.13% | 71.00±0.46% | 69.63±1.67% | **65.37±1.45%** | **54.33±1.25%** | **28.68±2.74%** |

Table 5: (Details of Fig. 9a) Test accuracy of CIFAR-100 dataset applying superclass label corruption, where weight decay value $\lambda = 5 \times 10^{-5}$.

| Distillation Step | 0.0 | 0.1 | 0.3 | 0.5 | 0.6 | 0.7 | 0.8 | 0.9 |
|---|---|---|---|---|---|---|---|---|
| 1 (Teacher) | 70.60±0.05% | 65.60±0.11% | 59.58±0.22% | 52.38±0.19% | 46.76±0.11% | 39.59±0.25% | 28.33±0.32% | 13.21±0.22% |
| 2 | 71.73±0.22% | 69.55±0.11% | 66.03±0.20% | 62.13±0.21% | 58.46±0.60% | 53.24±0.43% | 43.10±0.37% | 22.62±1.22% |
| 3 | **72.05±0.22%** | 69.79±0.26% | 66.34±0.20% | 62.35±0.12% | 58.63±0.57% | 53.41±0.51% | 43.57±0.32% | 23.34±1.13% |
| 4 | 72.03±0.25% | 69.83±0.17% | 66.48±0.21% | 62.40±0.12% | 58.70±0.55% | 53.52±0.52% | 43.72±0.32% | 23.53±1.17% |
| 5 | 72.04±0.28% | **69.86±0.28%** | 66.49±0.17% | 62.41±0.14% | 58.74±0.54% | 53.63±0.52% | 43.86±0.33% | 23.65±1.22% |
| PLL | 69.46±0.16% | 67.85±0.05% | **66.91±0.23%** | **63.88±0.28%** | **61.33±0.43%** | **57.13±0.14%** | **48.60±0.87%** | **26.46±0.73%** |

Table 6: (Details of Fig. 9a) Test accuracy of Flowers-102 dataset applying superclass label corruption, where weight decay value $\lambda = 5 \times 10^{-4}$.

| Distillation Step | 0.0 | 0.1 | 0.3 | 0.5 | 0.6 | 0.7 | 0.8 | 0.9 |
|---|---|---|---|---|---|---|---|---|
| 1 (Teacher) | **88.10±0.17%** | 80.85±0.30% | 67.75±0.64% | 51.93±1.06% | 42.39±1.77% | 32.88±1.62% | 22.55±0.48% | 9.64±0.28% |
| 2 | 87.94±0.29% | 83.04±0.28% | 73.20±0.09% | 60.20±1.05% | 52.06±2.67% | 41.80±2.69% | 29.02±1.64% | 11.76±0.85% |
| 3 | 87.78±0.39% | 83.66±0.20% | 75.39±0.21% | 63.14±1.68% | 53.73±2.52% | 43.53±2.06% | 30.23±1.01% | 13.01±1.01% |
| 4 | 87.29±0.39% | 83.82±0.14% | 75.82±0.41% | 63.73±1.52% | 54.12±2.17% | 44.25±1.60% | 30.72±1.06% | 13.27±1.17% |
| 5 | 87.03±0.24% | **83.92±0.35%** | **75.92±0.60%** | **64.22±1.61%** | 54.44±1.53% | 44.61±1.39% | 31.18±1.06% | 13.43±1.10% |
| PLL | 85.62±0.20% | 79.71±0.37% | 74.84±0.32% | 63.86±1.07% | **56.90±3.11%** | **46.37±1.91%** | **34.44±1.25%** | **14.22±0.42%** |

Table 7: (Details of Fig. 9a) Test accuracy of Food-101 dataset applying superclass label corruption, where weight decay value $\lambda = 5 \times 10^{-5}$.

| Distillation Step | 0.0 | 0.1 | 0.3 | 0.5 | 0.6 | 0.7 | 0.8 | 0.9 |
|---|---|---|---|---|---|---|---|---|
| 1 (Teacher) | **63.52±0.02%** | 61.77±0.16% | 57.96±0.25% | 52.44±0.23% | 48.35±0.10% | 42.36±0.12% | 32.80±0.97% | 15.69±0.36% |
| 2 | 63.45±0.07% | **62.41±0.09%** | 59.74±0.30% | 56.06±0.08% | 53.35±0.20% | 49.57±0.28% | 39.21±3.73% | 19.00±3.81% |
| 3 | 63.29±0.08% | 62.22±0.07% | 59.68±0.24% | 56.01±0.10% | 53.30±0.18% | 49.61±0.29% | 39.30±3.79% | 19.11±3.88% |
| 4 | 63.17±0.08% | 62.13±0.09% | 59.63±0.25% | 55.97±0.11% | 53.24±0.17% | 49.61±0.28% | 39.33±3.79% | 19.13±3.93% |
| 5 | 63.00±0.12% | 62.04±0.13% | 59.55±0.23% | 55.96±0.11% | 53.24±0.17% | 49.63±0.26% | 39.35±3.80% | 19.17±3.93% |
| PLL | 63.31±0.09% | 61.93±0.09% | **59.99±0.37%** | **57.74±0.07%** | **55.73±0.25%** | **53.24±0.09%** | **43.02±5.77%** | **21.06±7.16%** |

Table 8: (Details of Fig. 9a) Test accuracy of StanfordCars dataset applying superclass label corruption, where weight decay value $\lambda = 5 \times 10^{-4}$.

| Distillation Step | 0.0 | 0.1 | 0.3 | 0.5 | 0.6 | 0.7 | 0.8 | 0.9 |
|---|---|---|---|---|---|---|---|---|
| 1 (Teacher) | 43.02±0.03% | 35.92±0.27% | 26.60±0.13% | 17.52±0.11% | 13.43±0.26% | 9.88±0.21% | 5.92±0.14% | 2.93±0.29% |
| 2 | **43.18±0.06%** | 38.04±0.15% | 29.97±0.11% | 20.77±0.09% | 16.81±0.16% | 12.22±0.13% | 7.60±0.25% | 3.20±0.35% |
| 3 | 42.97±0.06% | 38.80±0.15% | 31.58±0.06% | 21.76±0.61% | 17.65±0.07% | 12.63±0.16% | 7.85±0.18% | 3.27±0.42% |
| 4 | 42.38±0.14% | **38.96±0.15%** | 32.30±0.22% | 21.80±0.92% | 17.78±0.22% | 12.73±0.16% | 7.93±0.25% | 3.32±0.52% |
| 5 | 41.76±0.01% | 38.79±0.36% | **32.65±0.21%** | 21.81±1.08% | 17.96±0.24% | 12.68±0.22% | 7.82±0.36% | **3.33±0.53%** |
| PLL | 41.28±0.14% | 36.32±0.18% | 31.36±0.25% | **23.55±0.22%** | **19.67±0.47%** | **14.17±0.40%** | **8.88±0.40%** | 3.04±0.53% |

Table 9: (Details of Fig. 9b) Test accuracy of Caltech-101 dataset applying symmetric label corruption, where weight decay value $\lambda = 5 \times 10^{-4}$.

| Distillation Step | 0.0 | 0.1 | 0.3 | 0.5 | 0.6 | 0.7 | 0.8 | 0.9 |
|---|---|---|---|---|---|---|---|---|
| 1 (Teacher) | 95.03±0.07% | 93.51±0.18% | 91.11±0.48% | 87.92±0.17% | 84.56±0.45% | 80.76±0.54% | 69.59±0.36% | 45.97±0.66% |
| 2 | **95.10±0.05%** | **93.70±0.22%** | 91.74±0.52% | 88.40±0.47% | 85.39±0.78% | 81.14±0.45% | 70.97±0.74% | 47.56±1.39% |
| 3 | 95.05±0.09% | 93.63±0.23% | **91.84±0.47%** | **88.63±0.64%** | 85.45±0.85% | 81.11±0.46% | 71.22±0.71% | 47.85±1.43% |
| 4 | 94.97±0.17% | 93.47±0.14% | 91.78±0.35% | 88.61±0.63% | 85.41±0.88% | 81.09±0.48% | 71.29±0.64% | 47.75±1.51% |
| 5 | 94.85±0.15% | 93.41±0.07% | 91.80±0.26% | 88.59±0.57% | 85.50±0.81% | 81.01±0.46% | 71.41±0.87% | 47.62±1.43% |
| PLL | 94.03±0.19% | 92.78±0.52% | 91.53±0.50% | 88.63±0.72% | **85.56±2.25%** | **82.51±0.19%** | **73.69±1.44%** | **51.77±2.93%** |

Table 10: (Details of Fig. 9b) Test accuracy of Caltech-256 dataset applying symmetric label corruption, where weight decay value $\lambda = 5 \times 10^{-5}$.

| Distillation Step | 0.0 | 0.1 | 0.3 | 0.5 | 0.6 | 0.7 | 0.8 | 0.9 |
|---|---|---|---|---|---|---|---|---|
| 1 (Teacher) | 83.63±0.03% | 81.76±0.15% | 78.59±0.39% | 74.50±0.29% | 71.47±0.12% | 66.45±0.76% | 57.47±0.42% | **34.41±0.17%** |
| 2 | **83.67±0.07%** | **82.01±0.35%** | **78.78±0.39%** | **74.68±0.26%** | **71.64±0.11%** | 66.86±0.87% | 57.42±0.31% | 34.31±0.12% |
| 3 | 83.60±0.08% | 81.90±0.30% | 78.71±0.37% | 74.61±0.31% | 71.56±0.14% | 66.84±0.85% | 57.30±0.31% | 34.23±0.15% |
| 4 | 83.53±0.08% | 81.80±0.31% | 78.67±0.37% | 74.57±0.28% | 71.51±0.15% | 66.77±0.78% | 57.20±0.30% | 34.15±0.12% |
| 5 | 83.53±0.05% | 81.78±0.32% | 78.65±0.35% | 74.57±0.26% | 71.42±0.18% | 66.69±0.73% | 57.17±0.31% | 34.07±0.15% |
| PLL | 82.64±0.24% | 80.43±0.71% | 76.47±0.41% | 73.33±0.68% | 70.60±0.20% | **68.16±1.11%** | **58.38±0.33%** | 32.58±1.35% |

Table 11: (Details of Fig. 9b) Test accuracy of CIFAR-100 dataset applying symmetric label corruption, where weight decay value $\lambda = 5 \times 10^{-5}$.

| Distillation Step | 0.0 | 0.1 | 0.3 | 0.5 | 0.6 | 0.7 | 0.8 | 0.9 |
|---|---|---|---|---|---|---|---|---|
| 1 (Teacher) | 70.60±0.05% | 66.07±0.06% | 61.66±0.21% | 55.98±0.10% | 51.60±0.22% | 45.78±0.08% | 35.83±0.19% | 18.81±0.14% |
| 2 | 71.73±0.22% | 70.02±0.14% | 67.41±0.36% | 64.44±0.43% | 62.43±0.20% | 58.83±0.81% | 51.42±0.66% | 34.38±0.70% |
| 3 | **72.05±0.22%** | 70.25±0.18% | 67.72±0.35% | 64.59±0.36% | 62.49±0.18% | 59.03±0.79% | 51.77±0.55% | 35.48±0.79% |
| 4 | 72.03±0.25% | **70.28±0.19%** | **67.77±0.39%** | 64.68±0.39% | 62.54±0.18% | 59.16±0.77% | 51.91±0.57% | 35.62±0.87% |
| 5 | 72.04±0.28% | 70.27±0.14% | 67.76±0.43% | 64.72±0.37% | 62.63±0.17% | 59.23±0.80% | 52.05±0.58% | 35.76±0.95% |
| PLL | 69.46±0.16% | 68.40±0.21% | 67.48±0.34% | **65.52±0.14%** | **63.66±0.27%** | **61.09±0.35%** | **56.21±0.18%** | **40.55±0.25%** |

Table 12: (Details of Fig. 9b) Test accuracy of Flowers-102 dataset applying symmetric label corruption, where weight decay value $\lambda = 5 \times 10^{-4}$.

| Distillation Step | 0.0 | 0.1 | 0.3 | 0.5 | 0.6 | 0.7 | 0.8 | 0.9 |
|---|---|---|---|---|---|---|---|---|
| 1 (Teacher) | **88.10±0.17%** | 80.39±0.56% | 67.16±0.88% | 51.41±1.79% | 43.82±1.24% | 34.48±0.53% | 22.55±0.08% | 10.75±1.35% |
| 2 | 87.94±0.29% | 82.91±1.04% | 74.41±0.42% | 62.22±2.37% | 55.39±1.24% | 45.52±1.44% | 32.39±1.85% | 12.81±1.04% |
| 3 | 87.78±0.39% | 83.82±0.64% | 75.26±0.47% | 63.40±2.73% | 57.12±0.59% | 46.63±0.93% | 33.50±2.12% | 13.89±1.75% |
| 4 | 87.29±0.39% | 83.99±0.76% | 75.39±0.68% | 63.66±2.54% | 57.78±0.32% | 46.86±0.68% | 33.79±2.51% | 14.25±1.74% |
| 5 | 87.03±0.24% | **84.02±1.05%** | **75.56±0.60%** | 63.56±2.64% | 57.78±0.26% | 47.09±0.79% | 34.02±2.54% | **14.48±1.44%** |
| PLL | 85.62±0.20% | 80.07±0.49% | 74.48±0.85% | **66.83±2.73%** | **60.39±2.68%** | **50.26±1.38%** | **35.88±1.40%** | 14.35±1.59% |

Table 13: (Details of Fig. 9b) Test accuracy of Food-101 dataset applying symmetric label corruption, where weight decay value $\lambda = 5 \times 10^{-5}$.

| Distillation Step | 0.0 | 0.1 | 0.3 | 0.5 | 0.6 | 0.7 | 0.8 | 0.9 |
|---|---|---|---|---|---|---|---|---|
| 1 (Teacher) | **63.52±0.02%** | 61.92±0.10% | 58.24±0.24% | 53.39±0.23% | 49.73±0.07% | 44.31±0.23% | 35.49±0.24% | 18.44±0.32% |
| 2 | 63.45±0.07% | **62.56±0.10%** | 60.21±0.07% | 56.99±0.15% | 53.22±2.36% | 51.21±0.28% | 44.69±0.11% | 24.92±4.30% |
| 3 | 63.29±0.08% | 62.47±0.09% | 60.15±0.07% | 56.95±0.17% | 53.24±2.37% | 51.22±0.34% | 44.79±0.11% | 24.94±4.36% |
| 4 | 63.17±0.08% | 62.39±0.12% | 60.10±0.07% | 56.92±0.16% | 53.20±2.40% | 51.19±0.36% | 44.84±0.09% | 25.00±4.42% |
| 5 | 63.00±0.12% | 62.25±0.11% | 60.00±0.05% | 56.90±0.16% | 53.19±2.41% | 51.18±0.36% | 44.84±0.11% | 25.04±4.45% |
| PLL | 63.31±0.09% | 62.03±0.12% | **60.24±0.05%** | 57.97±0.29% | 54.26±2.81% | **54.01±0.09%** | **49.23±0.18%** | **29.36±8.03%** |

Table 14: (Details of Fig. 9b) Test accuracy of StanfordCars dataset applying symmetric label corruption, where weight decay value $\lambda = 5 \times 10^{-4}$.

| Distillation Step | 0.0 | 0.1 | 0.3 | 0.5 | 0.6 | 0.7 | 0.8 | 0.9 |
|---|---|---|---|---|---|---|---|---|
| 1 (Teacher) | 43.02±0.03% | 35.62±0.35% | 26.51±0.32% | 17.54±0.14% | 13.26±0.39% | 9.16±0.29% | 5.45±0.17% | 2.49±0.12% |
| 2 | **43.18±0.06%** | 37.89±0.17% | 29.77±0.30% | 21.27±0.23% | 16.76±0.10% | 12.20±0.11% | 6.89±0.34% | 2.77±0.22% |
| 3 | 42.97±0.06% | 38.86±0.06% | 30.91±0.43% | 21.95±0.17% | 16.89±0.46% | 12.40±0.14% | 7.20±0.44% | 2.84±0.25% |
| 4 | 42.38±0.14% | **38.95±0.11%** | 31.36±0.54% | 22.13±0.46% | 16.91±0.56% | 12.32±0.17% | 7.26±0.54% | 2.90±0.33% |
| 5 | 41.76±0.01% | 38.86±0.10% | 31.61±0.63% | 22.10±0.46% | 16.72±0.70% | 12.19±0.16% | 7.23±0.62% | 2.92±0.34% |
| PLL | 41.28±0.14% | 36.58±0.19% | **31.97±0.76%** | **24.76±0.37%** | **20.56±0.27%** | **14.61±0.99%** | **8.26±0.69%** | **2.93±0.34%** |

Table 15: (Details of Fig. 9c) Test accuracy of Caltech-101 dataset applying asymmetric label corruption, where weight decay value $\lambda = 5 \times 10^{-4}$.

| Distillation Step | 0.0 | 0.1 | 0.3 | 0.5 | 0.6 | 0.7 | 0.8 | 0.9 |
|---|---|---|---|---|---|---|---|---|
| 1 (Teacher) | 95.03±0.07% | 93.63±0.22% | 91.05±0.19% | 87.69±0.41% | 85.14±0.66% | 80.22±1.08% | 70.87±0.91% | 45.12±0.92% |
| 2 | **95.10±0.05%** | **93.66±0.29%** | 91.44±0.55% | 88.27±0.45% | 85.93±0.35% | 80.66±1.10% | 71.74±0.60% | 46.52±1.40% |
| 3 | 95.05±0.09% | 93.55±0.33% | 91.53±0.55% | 88.31±0.40% | 85.96±0.23% | 80.68±1.04% | 71.83±0.81% | 46.70±1.47% |
| 4 | 94.97±0.17% | 93.53±0.23% | **91.61±0.53%** | 88.40±0.37% | 85.89±0.21% | 80.63±1.04% | 71.79±0.75% | 46.83±1.37% |
| 5 | 94.85±0.15% | 93.53±0.24% | 91.57±0.65% | 88.46±0.35% | 85.98±0.22% | 80.57±0.95% | 71.72±0.82% | 46.87±1.25% |
| PLL | 94.03±0.19% | 92.74±0.09% | 90.84±0.68% | **88.71±0.64%** | **87.14±0.53%** | **81.43±0.78%** | **72.50±2.54%** | **48.73±2.36%** |

Table 16: (Details of Fig. 9c) Test accuracy of Caltech-256 dataset applying asymmetric label corruption, where weight decay value $\lambda = 5 \times 10^{-5}$.

| Distillation Step | 0.0 | 0.1 | 0.3 | 0.5 | 0.6 | 0.7 | 0.8 | 0.9 |
|---|---|---|---|---|---|---|---|---|
| 1 (Teacher) | 83.63±0.03% | 81.65±0.18% | 78.52±0.20% | 74.35±0.19% | 71.22±0.28% | 66.50±0.37% | 57.42±0.71% | 35.13±1.10% |
| 2 | **83.67±0.07%** | **81.86±0.10%** | **78.62±0.24%** | **74.82±0.49%** | 71.27±0.42% | 66.76±0.28% | 57.72±0.87% | **35.19±1.18%** |
| 3 | 83.60±0.08% | 81.72±0.14% | 78.58±0.30% | 74.80±0.55% | **71.32±0.43%** | 66.73±0.23% | 57.64±0.99% | 35.16±1.19% |
| 4 | 83.53±0.08% | 81.69±0.16% | 78.53±0.28% | 74.74±0.54% | 71.27±0.42% | 66.68±0.25% | 57.57±1.02% | 35.11±1.18% |
| 5 | 83.53±0.05% | 81.65±0.18% | 78.51±0.32% | 74.72±0.54% | 71.21±0.44% | 66.63±0.31% | 57.54±1.01% | 35.11±1.24% |
| PLL | 82.64±0.24% | 80.38±0.21% | 75.87±0.49% | 74.04±1.26% | 70.82±0.87% | **68.04±0.84%** | **57.91±2.38%** | 34.65±2.46% |

Table 17: (Details of Fig. 9c) Test accuracy of CIFAR-100 dataset applying asymmetric label corruption, where weight decay value $\lambda = 5 \times 10^{-5}$.

| Distillation Step | 0.0 | 0.1 | 0.3 | 0.5 | 0.6 | 0.7 | 0.8 | 0.9 |
|---|---|---|---|---|---|---|---|---|
| 1 (Teacher) | 70.60±0.05% | 66.13±0.18% | 61.97±0.11% | 55.76±0.19% | 51.61±0.19% | 45.46±0.32% | 35.83±0.11% | 18.88±0.32% |
| 2 | 71.73±0.22% | 69.87±0.36% | 67.29±0.19% | 64.21±0.11% | 62.15±0.15% | 58.55±0.21% | 52.36±0.32% | 34.72±0.40% |
| 3 | **72.05±0.22%** | 70.01±0.43% | 67.37±0.19% | 64.43±0.17% | 62.25±0.48% | 58.78±0.18% | 52.62±0.32% | 36.16±0.11% |
| 4 | 72.03±0.25% | **70.06±0.43%** | 67.43±0.15% | 64.50±0.18% | 62.29±0.42% | 58.86±0.19% | 52.74±0.30% | 36.34±0.15% |
| 5 | 72.04±0.28% | 70.06±0.46% | 67.47±0.14% | 64.57±0.18% | 62.32±0.41% | 58.89±0.25% | 52.84±0.26% | 36.46±0.16% |
| PLL | 69.46±0.16% | 68.41±0.15% | **67.59±0.20%** | **65.08±0.26%** | **63.55±0.28%** | **60.73±0.22%** | **56.51±0.10%** | **40.53±0.55%** |

Table 18: (Details of Fig. 9c) Test accuracy of Flowers-102 dataset applying asymmetric label corruption, where weight decay value $\lambda = 5 \times 10^{-4}$.

| Distillation Step | 0.0 | 0.1 | 0.3 | 0.5 | 0.6 | 0.7 | 0.8 | 0.9 |
|---|---|---|---|---|---|---|---|---|
| 1 (Teacher) | **88.10±0.17%** | 80.33±0.12% | 67.71±1.07% | 52.29±1.45% | 44.90±1.39% | 34.28±1.90% | 22.22±0.32% | 11.08±1.13% |
| 2 | 87.94±0.29% | 82.71±0.20% | 73.89±0.46% | 63.59±2.16% | 57.19±2.98% | 46.50±1.42% | 31.24±2.05% | 15.95±1.93% |
| 3 | 87.78±0.39% | 83.40±0.61% | 75.92±0.72% | 65.20±2.21% | 59.44±1.73% | 48.33±1.08% | 32.39±2.28% | 15.95±1.62% |
| 4 | 87.29±0.39% | 83.53±0.72% | 76.31±1.00% | 65.56±1.92% | 60.20±1.11% | 48.82±0.88% | 32.61±2.14% | 15.82±1.74% |
| 5 | 87.03±0.24% | **83.82±0.85%** | **76.57±1.15%** | 65.62±1.86% | 60.52±0.91% | 48.95±0.61% | 32.32±2.04% | 15.72±1.74% |
| PLL | 85.62±0.20% | 79.44±0.18% | 75.23±0.74% | **66.73±2.00%** | **62.22±1.67%** | **51.27±1.27%** | **36.44±1.60%** | **17.48±2.36%** |

Table 19: (Details of Fig. 9c) Test accuracy of Food-101 dataset applying asymmetric label corruption, where weight decay value $\lambda = 5 \times 10^{-5}$.

| Distillation Step | 0.0 | 0.1 | 0.3 | 0.5 | 0.6 | 0.7 | 0.8 | 0.9 |
|---|---|---|---|---|---|---|---|---|
| 1 (Teacher) | **63.52±0.02%** | 61.83±0.14% | 58.27±0.10% | 53.17±0.12% | 49.74±0.17% | 44.34±0.22% | 35.16±0.20% | 18.62±0.34% |
| 2 | 63.45±0.07% | **62.43±0.11%** | **60.17±0.12%** | 56.84±0.08% | 53.95±0.64% | 51.07±0.40% | 43.86±0.21% | 25.16±4.31% |
| 3 | 63.29±0.08% | 62.31±0.12% | 60.12±0.12% | 56.84±0.07% | 53.97±0.63% | 51.06±0.43% | 43.92±0.17% | 25.24±4.39% |
| 4 | 63.17±0.08% | 62.19±0.13% | 60.04±0.17% | 56.81±0.11% | 53.96±0.63% | 51.03±0.44% | 43.94±0.17% | 25.31±4.39% |
| 5 | 63.00±0.12% | 62.10±0.14% | 60.00±0.18% | 56.78±0.13% | 53.96±0.63% | 51.04±0.44% | 43.96±0.17% | 25.35±4.39% |
| PLL | 63.31±0.09% | 61.86±0.06% | 60.11±0.13% | **57.69±0.07%** | **55.71±0.73%** | **53.86±0.29%** | **48.52±0.33%** | **29.71±7.46%** |

Table 20: (Details of Fig. 9c) Test accuracy of StanfordCars dataset applying asymmetric label corruption, where weight decay value $\lambda = 5 \times 10^{-4}$.

| Distillation Step | 0.0 | 0.1 | 0.3 | 0.5 | 0.6 | 0.7 | 0.8 | 0.9 |
|---|---|---|---|---|---|---|---|---|
| 1 (Teacher) | 43.02±0.03% | 35.51±0.18% | 26.38±0.31% | 17.22±0.31% | 13.04±0.24% | 9.48±0.15% | 5.35±0.19% | 2.51±0.10% |
| 2 | **43.18±0.06%** | 37.84±0.05% | 29.76±0.26% | 21.00±0.31% | 16.76±0.35% | 11.79±0.39% | 6.78±0.46% | 3.03±0.20% |
| 3 | 42.97±0.06% | 38.70±0.04% | 31.13±0.14% | 21.71±0.53% | 16.97±0.62% | 12.05±0.32% | 7.08±0.51% | 3.12±0.20% |
| 4 | 42.38±0.14% | **38.73±0.03%** | 31.89±0.24% | 21.76±0.90% | 16.84±0.77% | 12.00±0.33% | 7.17±0.57% | 3.13±0.20% |
| 5 | 41.76±0.01% | 38.52±0.04% | **32.02±0.15%** | 21.77±1.15% | 16.63±0.68% | 11.87±0.36% | 7.19±0.58% | **3.18±0.16%** |
| PLL | 41.28±0.14% | 36.63±0.22% | 31.57±0.40% | **24.78±0.56%** | **20.69±0.34%** | **13.17±0.71%** | **8.20±1.14%** | 3.17±0.35% |

Table 21: (Details of Fig. 14a) Test accuracy of Caltech-101 dataset applying superclass label corruption, where weight decay value $\lambda = 2 \times 10^{-4}$.

| Distillation Step | 0.0 | 0.1 | 0.3 | 0.5 | 0.6 | 0.7 | 0.8 | 0.9 |
|---|---|---|---|---|---|---|---|---|
| 1 (Teacher) | 95.01±0.10% | 93.45±0.24% | 90.38±0.33% | 84.29±0.91% | 78.74±0.92% | 67.76±1.16% | 47.37±1.77% | 16.36±1.02% |
| 2 | **95.10±0.05%** | **93.70±0.19%** | **90.86±0.27%** | **84.41±0.85%** | 79.80±0.64% | 68.43±1.31% | 49.42±1.79% | 18.47±1.41% |
| 3 | 95.05±0.05% | 93.61±0.14% | 90.82±0.26% | 84.39±0.80% | 79.99±0.49% | 68.64±1.36% | 49.48±1.96% | 18.51±1.58% |
| 4 | 94.95±0.05% | 93.61±0.12% | 90.73±0.25% | 84.25±0.78% | 79.92±0.50% | 68.70±1.32% | 49.40±2.02% | 18.61±1.67% |
| 5 | 94.91±0.07% | 93.57±0.15% | 90.63±0.17% | 84.16±0.79% | 79.90±0.50% | 68.70±1.32% | 49.25±2.04% | 18.99±2.42% |
| PLL | 93.82±0.96% | 92.67±0.41% | 89.75±0.50% | 84.31±0.72% | **81.13±0.67%** | **70.87±1.13%** | **51.67±2.74%** | **19.72±1.43%** |

Table 22: (Details of Fig. 14a) Test accuracy of Caltech-256 dataset applying superclass label corruption, where weight decay value $\lambda = 2 \times 10^{-5}$.

| Distillation Step | 0.0 | 0.1 | 0.3 | 0.5 | 0.6 | 0.7 | 0.8 | 0.9 |
|---|---|---|---|---|---|---|---|---|
| 1 (Teacher) | 83.63±0.01% | 81.69±0.23% | 77.96±0.24% | 72.60±0.36% | 69.57±0.03% | 63.66±0.57% | 52.61±0.75% | 28.32±1.56% |
| 2 | **83.68±0.08%** | **81.97±0.16%** | **78.14±0.17%** | **72.62±0.28%** | **69.77±0.18%** | 63.87±0.68% | 52.88±0.95% | 28.62±1.82% |
| 3 | 83.64±0.08% | 81.91±0.19% | 78.09±0.09% | 72.52±0.37% | 69.76±0.21% | 63.84±0.69% | 52.80±0.91% | 28.54±1.91% |
| 4 | 83.55±0.08% | 81.83±0.23% | 78.03±0.11% | 72.46±0.34% | 69.72±0.21% | 63.78±0.63% | 52.76±0.94% | 28.48±1.85% |
| 5 | 83.50±0.07% | 81.78±0.25% | 77.92±0.10% | 72.45±0.33% | 69.68±0.26% | 63.72±0.70% | 52.73±0.98% | 28.40±1.89% |
| PLL | 82.49±0.12% | 80.23±0.31% | 76.09±1.11% | 71.00±0.46% | 69.63±1.68% | **65.42±1.44%** | **54.37±1.23%** | **28.70±2.70%** |

Table 23: (Details of Fig. 14a) Test accuracy of CIFAR-100 dataset applying superclass label corruption, where weight decay value $\lambda = 2 \times 10^{-5}$.

| Distillation Step | 0.0 | 0.1 | 0.3 | 0.5 | 0.6 | 0.7 | 0.8 | 0.9 |
|---|---|---|---|---|---|---|---|---|
| 1 (Teacher) | 70.35±0.02% | 64.82±0.07% | 58.54±0.26% | 51.24±0.08% | 45.54±0.12% | 38.39±0.20% | 27.34±0.27% | 12.82±0.17% |
| 2 | 71.45±0.18% | 68.02±1.09% | 65.87±0.23% | 61.70±0.19% | 59.01±0.16% | 52.27±0.27% | 42.63±0.40% | 22.04±0.97% |
| 3 | 71.59±0.24% | 68.12±1.06% | 66.26±0.37% | 62.07±0.37% | 59.11±0.20% | 53.07±1.15% | 43.05±0.26% | 22.75±0.95% |
| 4 | **71.63±0.23%** | **68.14±1.07%** | 66.31±0.39% | 62.11±0.40% | 59.15±0.21% | 53.15±1.16% | 43.17±0.22% | 22.83±0.97% |
| 5 | 71.62±0.18% | 68.14±1.05% | **66.34±0.38%** | 62.13±0.40% | 59.19±0.20% | 53.18±1.16% | 43.23±0.27% | 22.95±1.00% |
| PLL | 69.30±0.20% | 66.21±0.72% | 65.85±0.46% | **62.96±0.34%** | **60.65±0.21%** | **56.61±0.23%** | **47.50±0.81%** | **25.58±0.80%** |

Table 24: (Details of Fig. 14a) Test accuracy of Flowers-102 dataset applying superclass label corruption, where weight decay value $\lambda = 2 \times 10^{-4}$.

| Distillation Step | 0.0 | 0.1 | 0.3 | 0.5 | 0.6 | 0.7 | 0.8 | 0.9 |
|---|---|---|---|---|---|---|---|---|
| 1 (Teacher) | **88.01±0.17%** | 80.36±0.09% | 66.41±0.53% | 50.36±0.99% | 40.16±1.76% | 31.01±1.36% | 21.14±0.59% | 9.22±0.28% |
| 2 | 87.75±0.29% | 82.35±0.21% | 72.12±0.58% | 58.04±0.81% | 48.95±1.00% | 39.44±1.93% | 27.48±1.76% | 10.36±0.48% |
| 3 | 87.75±0.24% | 82.88±0.17% | 73.69±0.81% | 59.61±1.60% | 50.85±0.36% | 40.62±1.32% | 28.33±1.35% | 10.69±0.76% |
| 4 | 87.55±0.16% | 82.97±0.33% | **74.12±0.85%** | 60.16±1.72% | 51.24±0.32% | 41.24±0.74% | 28.86±1.59% | 10.69±0.90% |
| 5 | 87.09±0.30% | **83.04±0.49%** | 74.08±0.88% | **60.33±1.93%** | **51.57±0.52%** | **41.27±0.50%** | **29.25±1.78%** | 10.65±1.09% |
| PLL | 85.23±0.18% | 79.08±0.26% | 70.33±1.38% | 57.91±2.22% | 50.29±1.32% | 40.16±1.29% | 28.92±1.42% | **11.67±0.69%** |

Table 25: (Details of Fig. 14a) Test accuracy of Food-101 dataset applying superclass label corruption, where weight decay value $\lambda = 2 \times 10^{-5}$.

| Distillation Step | 0.0 | 0.1 | 0.3 | 0.5 | 0.6 | 0.7 | 0.8 | 0.9 |
|---|---|---|---|---|---|---|---|---|
| 1 (Teacher) | 63.46±0.03% | 61.66±0.17% | 57.79±0.24% | 52.15±0.24% | 48.09±0.06% | 42.04±0.14% | 32.56±1.05% | 15.58±0.46% |
| 2 | **63.52±0.05%** | **62.25±0.29%** | 59.72±0.26% | 56.14±0.27% | 51.75±2.53% | 47.45±2.69% | 37.55±3.69% | 19.13±4.01% |
| 3 | 63.28±0.08% | 62.10±0.30% | 59.66±0.28% | 56.12±0.25% | 51.68±2.56% | 47.42±2.68% | 37.54±3.70% | 19.15±4.05% |
| 4 | 63.13±0.04% | 61.99±0.29% | 59.56±0.24% | 56.08±0.28% | 51.63±2.53% | 47.39±2.68% | 37.52±3.69% | 19.16±4.06% |
| 5 | 62.98±0.08% | 61.90±0.29% | 59.47±0.23% | 56.04±0.26% | 51.59±2.52% | 47.37±2.70% | 37.52±3.71% | 19.15±4.09% |
| PLL | 63.36±0.14% | 61.85±0.20% | **59.93±0.36%** | **57.70±0.12%** | **53.57±3.23%** | **50.75±3.39%** | **40.59±5.43%** | **21.30±7.40%** |

Table 26: (Details of Fig. 14a) Test accuracy of StanfordCars dataset applying superclass label corruption, where weight decay value $\lambda = 2 \times 10^{-4}$.

| Distillation Step | 0.0 | 0.1 | 0.3 | 0.5 | 0.6 | 0.7 | 0.8 | 0.9 |
|---|---|---|---|---|---|---|---|---|
| 1 (Teacher) | 42.62±0.06% | 34.42±0.30% | 24.40±0.13% | 15.74±0.09% | 11.94±0.24% | 8.95±0.09% | 5.51±0.14% | 2.96±0.19% |
| 2 | 42.97±0.15% | 35.96±0.25% | 27.42±0.22% | 18.23±0.47% | 14.48±0.45% | 9.97±0.97% | 6.04±0.19% | 3.15±0.23% |
| 3 | **43.03±0.18%** | 36.90±0.21% | 28.61±0.45% | 18.60±0.63% | 14.99±0.44% | 10.28±1.27% | 6.29±0.31% | 3.21±0.33% |
| 4 | 42.92±0.09% | 37.36±0.28% | 28.77±0.69% | **18.75±0.72%** | 15.13±0.44% | 10.32±1.36% | **6.30±0.39%** | **3.25±0.37%** |
| 5 | 42.82±0.07% | **37.74±0.36%** | **28.88±0.76%** | 18.73±0.68% | **15.23±0.48%** | **10.41±1.48%** | 6.28±0.43% | 3.23±0.35% |
| PLL | 41.30±0.06% | 34.57±0.46% | 27.97±0.50% | 18.51±1.10% | 14.61±0.28% | 9.44±1.37% | 5.99±0.36% | 2.98±0.36% |

Table 27: (Details of Fig. 14b) Test accuracy of Caltech-101 dataset applying symmetric label corruption, where weight decay value $\lambda = 2 \times 10^{-4}$.

| Distillation Step | 0.0 | 0.1 | 0.3 | 0.5 | 0.6 | 0.7 | 0.8 | 0.9 |
|---|---|---|---|---|---|---|---|---|
| 1 (Teacher) | 95.01±0.10% | 93.51±0.18% | 91.09±0.51% | 87.86±0.12% | 84.52±0.43% | 80.72±0.57% | 69.57±0.31% | 45.87±0.73% |
| 2 | **95.10±0.05%** | **93.70±0.27%** | 91.72±0.54% | 88.38±0.50% | 85.23±0.83% | 81.03±0.55% | 70.89±0.84% | 47.37±1.39% |
| 3 | 95.05±0.05% | 93.59±0.30% | **91.90±0.42%** | 88.36±0.55% | 85.22±0.85% | 80.99±0.54% | 70.95±0.78% | 47.43±1.49% |
| 4 | 94.95±0.05% | 93.49±0.29% | 91.82±0.38% | 88.36±0.54% | 85.12±0.73% | 80.86±0.43% | 70.89±0.86% | 47.48±1.27% |
| 5 | 94.91±0.07% | 93.43±0.26% | 91.72±0.39% | 88.34±0.49% | 85.06±0.73% | 80.78±0.47% | 70.72±0.84% | 47.31±1.22% |
| PLL | 93.82±0.96% | 92.78±0.48% | 91.51±0.52% | **88.81±0.72%** | **85.56±2.39%** | **82.39±0.17%** | **73.71±1.42%** | **51.69±2.92%** |

Table 28: (Details of Fig. 14b) Test accuracy of Caltech-256 dataset applying symmetric label corruption, where weight decay value $\lambda = 2 \times 10^{-5}$.

| Distillation Step | 0.0 | 0.1 | 0.3 | 0.5 | 0.6 | 0.7 | 0.8 | 0.9 |
|---|---|---|---|---|---|---|---|---|
| 1 (Teacher) | 83.63±0.01% | 81.76±0.14% | 78.59±0.39% | 74.50±0.29% | 71.49±0.14% | 66.44±0.75% | 57.44±0.43% | 34.41±0.17% |
| 2 | **83.68±0.08%** | **81.98±0.34%** | **78.79±0.40%** | **74.65±0.25%** | **71.63±0.10%** | 66.84±0.87% | 57.38±0.32% | **34.62±0.40%** |
| 3 | 83.64±0.08% | 81.89±0.30% | 78.68±0.37% | 74.60±0.26% | 71.51±0.10% | 66.81±0.82% | 57.25±0.37% | 34.60±0.35% |
| 4 | 83.55±0.08% | 81.82±0.34% | 78.61±0.41% | 74.52±0.25% | 71.43±0.09% | 66.76±0.76% | 57.09±0.32% | 34.48±0.40% |
| 5 | 83.50±0.07% | 81.76±0.34% | 78.55±0.38% | 74.46±0.25% | 71.39±0.11% | 66.72±0.76% | 57.02±0.39% | 34.40±0.44% |
| PLL | 82.49±0.12% | 80.45±0.71% | 76.51±0.39% | 73.33±0.65% | 70.60±0.19% | **68.19±1.15%** | **58.41±0.31%** | 34.30±1.03% |

Table 29: (Details of Fig. 14b) Test accuracy of CIFAR-100 dataset applying symmetric label corruption, where weight decay value $\lambda = 2 \times 10^{-5}$.

| Distillation Step | 0.0 | 0.1 | 0.3 | 0.5 | 0.6 | 0.7 | 0.8 | 0.9 |
|---|---|---|---|---|---|---|---|---|
| 1 (Teacher) | 70.35±0.02% | 65.18±0.14% | 60.68±0.17% | 55.04±0.07% | 50.50±0.23% | 44.69±0.01% | 34.64±0.17% | 18.09±0.10% |
| 2 | 71.45±0.18% | 69.27±0.63% | 67.30±0.29% | 64.13±0.20% | 62.07±0.02% | 58.56±0.83% | 51.33±0.87% | 33.35±0.70% |
| 3 | 71.59±0.24% | 69.60±0.42% | 67.39±0.33% | 64.43±0.10% | 62.21±0.07% | 58.74±0.80% | 51.78±0.65% | 34.36±0.73% |
| 4 | **71.63±0.23%** | **69.64±0.39%** | **67.41±0.33%** | 64.45±0.10% | 62.26±0.08% | 58.83±0.77% | 51.87±0.61% | 34.41±0.74% |
| 5 | 71.62±0.18% | 69.62±0.38% | 67.39±0.32% | 64.47±0.08% | 62.30±0.11% | 58.85±0.81% | 51.93±0.60% | 34.42±0.76% |
| PLL | 69.30±0.20% | 67.34±0.43% | 67.14±0.18% | **65.20±0.11%** | **63.35±0.39%** | **60.54±0.30%** | **55.18±0.81%** | **38.94±1.43%** |

Table 30: (Details of Fig. 14b) Test accuracy of Flowers-102 dataset applying symmetric label corruption, where weight decay value $\lambda = 2 \times 10^{-4}$.

| Distillation Step | 0.0 | 0.1 | 0.3 | 0.5 | 0.6 | 0.7 | 0.8 | 0.9 |
|---|---|---|---|---|---|---|---|---|
| 1 (Teacher) | **88.01±0.17%** | 79.74±0.54% | 66.18±0.87% | 49.48±1.79% | 41.31±1.45% | 32.12±0.86% | 21.01±0.18% | 10.07±1.55% |
| 2 | 87.75±0.29% | 82.45±0.37% | 72.55±1.73% | 58.01±0.09% | 52.88±0.38% | 42.97±0.56% | 29.08±1.20% | 11.37±1.88% |
| 3 | 87.75±0.24% | 83.40±0.41% | 73.53±1.56% | **59.12±0.48%** | 55.59±1.06% | 45.42±0.82% | 30.23±1.42% | **11.99±2.72%** |
| 4 | 87.55±0.16% | **83.73±0.58%** | 73.89±1.54% | 59.08±0.48% | 56.37±1.13% | 45.78±1.04% | **30.75±1.71%** | 11.99±2.85% |
| 5 | 87.09±0.30% | 83.56±0.52% | **73.95±1.85%** | 59.02±0.73% | **56.70±1.18%** | **45.88±0.97%** | 30.65±1.78% | 11.96±2.74% |
| PLL | 85.23±0.18% | 78.63±1.05% | 71.24±2.18% | 58.50±1.76% | 54.38±1.73% | 44.31±0.50% | 30.56±1.14% | 11.86±3.07% |

Table 31: (Details of Fig. 14b) Test accuracy of Food-101 dataset applying symmetric label corruption, where weight decay value $\lambda = 2 \times 10^{-5}$.

| Distillation Step | 0.0 | 0.1 | 0.3 | 0.5 | 0.6 | 0.7 | 0.8 | 0.9 |
|---|---|---|---|---|---|---|---|---|
| 1 (Teacher) | 63.46±0.03% | 61.81±0.12% | 58.06±0.23% | 53.10±0.24% | 49.50±0.13% | 43.98±0.25% | 35.08±0.26% | 18.24±0.44% |
| 2 | **63.52±0.05%** | **62.27±0.32%** | 59.86±0.50% | 56.96±0.14% | 53.22±2.40% | 51.15±0.28% | 43.47±1.81% | 21.67±4.58% |
| 3 | 63.28±0.08% | 62.13±0.26% | 59.71±0.48% | 56.91±0.14% | 53.21±2.41% | 51.12±0.34% | 43.48±1.82% | 21.63±4.61% |
| 4 | 63.13±0.04% | 62.01±0.29% | 59.59±0.51% | 56.85±0.13% | 53.16±2.43% | 51.09±0.36% | 43.49±1.82% | 21.61±4.63% |
| 5 | 62.98±0.08% | 61.88±0.33% | 59.53±0.50% | 56.81±0.13% | 53.12±2.45% | 51.03±0.36% | 43.47±1.81% | 21.60±4.62% |
| PLL | 63.36±0.14% | 61.65±0.52% | **59.90±0.55%** | **58.00±0.19%** | **54.21±2.77%** | **53.98±0.10%** | **47.61±2.34%** | **23.58±8.33%** |

Table 32: (Details of Fig. 14b) Test accuracy of StanfordCars dataset applying symmetric label corruption, where weight decay value $\lambda = 2 \times 10^{-4}$.

| Distillation Step | 0.0 | 0.1 | 0.3 | 0.5 | 0.6 | 0.7 | 0.8 | 0.9 |
|---|---|---|---|---|---|---|---|---|
| 1 (Teacher) | 42.62±0.06% | 34.15±0.27% | 24.50±0.43% | 15.53±0.28% | 11.61±0.30% | 8.43±0.11% | 4.86±0.13% | 2.35±0.11% |
| 2 | 42.97±0.15% | 35.63±0.31% | 27.15±0.43% | 18.63±0.54% | 13.95±1.44% | 9.26±0.88% | 5.43±0.41% | 2.57±0.17% |
| 3 | **43.03±0.18%** | 36.80±0.37% | 28.37±0.75% | 19.21±0.74% | 14.26±1.55% | **9.35±0.88%** | 5.62±0.50% | 2.73±0.11% |
| 4 | 42.92±0.09% | 37.40±0.37% | 28.70±0.95% | **19.31±0.84%** | 14.29±1.59% | 9.29±0.90% | 5.70±0.49% | 2.74±0.12% |
| 5 | 42.82±0.07% | **37.62±0.43%** | **28.79±1.16%** | 19.31±0.94% | 14.29±1.48% | 9.27±0.83% | **5.74±0.42%** | **2.76±0.15%** |
| PLL | 41.30±0.06% | 34.69±0.02% | 27.86±0.69% | 19.14±0.70% | **14.58±2.18%** | 9.29±1.35% | 5.43±0.40% | 2.45±0.20% |

Table 33: (Details of Fig. 14c) Test accuracy of Caltech-101 dataset applying asymmetric label corruption, where weight decay value $\lambda = 2 \times 10^{-4}$.

| Distillation Step | 0.0 | 0.1 | 0.3 | 0.5 | 0.6 | 0.7 | 0.8 | 0.9 |
|---|---|---|---|---|---|---|---|---|
| 1 (Teacher) | 95.01±0.10% | 93.59±0.20% | 91.05±0.15% | 87.69±0.41% | 85.04±0.71% | 80.17±1.07% | 70.74±0.85% | 45.16±0.94% |
| 2 | **95.10±0.05%** | **93.66±0.34%** | 91.34±0.56% | 88.23±0.46% | 85.77±0.22% | 80.57±1.08% | 71.64±0.61% | 46.39±1.46% |
| 3 | 95.05±0.05% | 93.51±0.34% | **91.47±0.50%** | 88.33±0.44% | 85.79±0.23% | 80.40±1.07% | 71.60±0.82% | 46.43±1.44% |
| 4 | 94.95±0.05% | 93.51±0.30% | 91.42±0.50% | 88.25±0.42% | 85.77±0.29% | 80.26±1.13% | 71.47±0.88% | 46.35±1.44% |
| 5 | 94.91±0.07% | 93.53±0.24% | 91.36±0.54% | 88.23±0.45% | 85.70±0.22% | 80.18±1.10% | 71.43±0.86% | 46.37±1.51% |
| PLL | 93.82±0.96% | 92.72±0.10% | 90.80±0.73% | **88.82±0.61%** | **87.14±0.48%** | **81.36±0.74%** | **72.45±2.41%** | **48.69±2.46%** |

Table 34: (Details of Fig. 14c) Test accuracy of Caltech-256 dataset applying asymmetric label corruption, where weight decay value $\lambda = 2 \times 10^{-5}$.

| Distillation Step | 0.0 | 0.1 | 0.3 | 0.5 | 0.6 | 0.7 | 0.8 | 0.9 |
|---|---|---|---|---|---|---|---|---|
| 1 (Teacher) | 83.63±0.01% | 81.65±0.19% | 78.51±0.20% | 74.34±0.18% | 71.22±0.29% | 66.50±0.35% | 57.42±0.71% | 35.12±1.09% |
| 2 | **83.68±0.08%** | **81.86±0.10%** | **78.59±0.25%** | **74.80±0.49%** | 71.26±0.40% | 66.74±0.28% | 57.69±0.86% | **35.16±1.16%** |
| 3 | 83.64±0.08% | 81.72±0.14% | 78.53±0.27% | 74.77±0.52% | **71.28±0.44%** | 66.70±0.30% | 57.58±0.95% | 35.10±1.20% |
| 4 | 83.55±0.08% | 81.68±0.12% | 78.50±0.25% | 74.69±0.54% | 71.17±0.43% | 66.64±0.30% | 57.49±0.96% | 35.06±1.23% |
| 5 | 83.50±0.07% | 81.65±0.18% | 78.45±0.28% | 74.65±0.53% | 71.14±0.41% | 66.63±0.31% | 57.43±0.98% | 34.99±1.23% |
| PLL | 82.49±0.12% | 80.37±0.19% | 75.88±0.50% | 74.06±1.28% | 70.82±0.87% | **68.02±0.83%** | **57.95±2.39%** | 34.70±2.49% |

Table 35: (Details of Fig. 14c) Test accuracy of CIFAR-100 dataset applying asymmetric label corruption, where weight decay value $\lambda = 2 \times 10^{-5}$.

| Distillation Step | 0.0 | 0.1 | 0.3 | 0.5 | 0.6 | 0.7 | 0.8 | 0.9 |
|---|---|---|---|---|---|---|---|---|
| 1 (Teacher) | 70.35±0.02% | 65.40±0.17% | 61.05±0.17% | 54.78±0.22% | 50.68±0.29% | 44.38±0.36% | 34.81±0.03% | 18.21±0.25% |
| 2 | 71.45±0.18% | 69.13±0.60% | 67.27±0.31% | 64.15±0.28% | 62.20±0.28% | 58.23±0.19% | 51.96±0.13% | 34.15±0.85% |
| 3 | 71.59±0.24% | 69.30±0.49% | 67.36±0.27% | 64.48±0.13% | 62.26±0.27% | 58.49±0.18% | 52.37±0.18% | 35.27±1.02% |
| 4 | **71.63±0.23%** | **69.37±0.50%** | 67.36±0.25% | 64.53±0.10% | 62.27±0.26% | 58.56±0.21% | 52.47±0.22% | 35.33±1.02% |
| 5 | 71.62±0.18% | 69.36±0.50% | **67.38±0.21%** | 64.54±0.10% | 62.28±0.28% | 58.57±0.23% | 52.51±0.22% | 35.39±1.03% |
| PLL | 69.30±0.20% | 67.25±0.49% | 67.12±0.22% | **64.88±0.16%** | **63.10±0.02%** | **60.40±0.22%** | **55.71±0.16%** | **38.92±1.46%** |

Table 36: (Details of Fig. 14c) Test accuracy of Flowers-102 dataset applying asymmetric label corruption, where weight decay value $\lambda = 2 \times 10^{-4}$.

| Distillation Step | 0.0 | 0.1 | 0.3 | 0.5 | 0.6 | 0.7 | 0.8 | 0.9 |
|---|---|---|---|---|---|---|---|---|
| 1 (Teacher) | **88.01±0.17%** | 79.97±0.17% | 66.83±1.09% | 49.77±1.43% | 42.84±1.18% | 32.19±1.78% | 20.62±0.12% | 10.33±1.00% |
| 2 | 87.75±0.29% | 81.96±0.66% | 72.58±0.74% | 59.31±3.68% | 53.43±3.19% | 44.41±2.78% | 28.53±3.12% | 13.59±1.17% |
| 3 | 87.75±0.24% | 82.45±0.85% | 74.02±1.56% | 61.57±3.36% | 54.58±2.39% | 45.29±2.74% | 30.03±2.58% | 14.87±1.61% |
| 4 | 87.55±0.16% | 82.29±1.03% | **74.61±2.04%** | 62.19±3.36% | 54.97±2.05% | **45.46±2.86%** | 30.39±2.46% | 15.00±1.86% |
| 5 | 87.09±0.30% | **82.48±1.21%** | 74.58±2.17% | **62.55±3.20%** | **55.03±1.89%** | 45.23±2.62% | **30.49±2.47%** | 14.97±1.92% |
| PLL | 85.23±0.18% | 78.37±0.72% | 70.23±1.09% | 59.77±3.86% | 54.71±2.20% | 44.80±1.85% | 28.76±1.96% | **15.59±1.45%** |

Table 37: (Details of Fig. 14c) Test accuracy of Food-101 dataset applying asymmetric label corruption, where weight decay value $\lambda = 2 \times 10^{-5}$.

| Distillation Step | 0.0 | 0.1 | 0.3 | 0.5 | 0.6 | 0.7 | 0.8 | 0.9 |
|---|---|---|---|---|---|---|---|---|
| 1 (Teacher) | 63.46±0.03% | 61.74±0.16% | 58.06±0.10% | 52.97±0.10% | 49.49±0.12% | 43.97±0.21% | 34.79±0.11% | 18.46±0.38% |
| 2 | **63.52±0.05%** | **62.31±0.06%** | **60.00±0.22%** | 55.85±1.39% | 53.86±0.54% | 50.90±0.41% | 41.82±3.47% | 22.22±3.52% |
| 3 | 63.28±0.08% | 62.19±0.08% | 59.95±0.19% | 55.79±1.41% | 53.85±0.56% | 50.86±0.44% | 41.80±3.47% | 22.21±3.49% |
| 4 | 63.13±0.04% | 62.07±0.01% | 59.89±0.15% | 55.74±1.42% | 53.80±0.53% | 50.82±0.45% | 41.78±3.49% | 22.20±3.49% |
| 5 | 62.98±0.08% | 61.93±0.01% | 59.84±0.16% | 55.71±1.42% | 53.79±0.54% | 50.83±0.45% | 41.77±3.51% | 22.20±3.49% |
| PLL | 63.36±0.14% | 61.74±0.21% | 59.97±0.25% | **56.55±1.52%** | **55.59±0.68%** | **53.78±0.23%** | **45.67±4.76%** | **24.38±5.95%** |

Table 38: (Details of Fig. 14c) Test accuracy of StanfordCars dataset applying asymmetric label corruption, where weight decay value $\lambda = 2 \times 10^{-4}$.

| Distillation Step | 0.0 | 0.1 | 0.3 | 0.5 | 0.6 | 0.7 | 0.8 | 0.9 |
|---|---|---|---|---|---|---|---|---|
| 1 (Teacher) | 42.62±0.06% | 33.97±0.12% | 24.35±0.31% | 15.23±0.26% | 11.51±0.25% | 8.59±0.09% | 5.07±0.20% | 2.36±0.08% |
| 2 | 42.97±0.15% | 35.63±0.24% | 27.24±0.50% | 18.65±0.66% | 12.42±0.67% | 9.16±0.78% | 5.19±0.35% | **2.55±0.16%** |
| 3 | **43.03±0.18%** | 36.78±0.37% | 27.97±0.68% | 19.12±1.01% | 12.50±0.74% | 9.29±1.11% | 5.29±0.42% | 2.55±0.22% |
| 4 | 42.92±0.09% | 37.33±0.27% | 28.25±0.82% | **19.19±1.07%** | **12.56±0.79%** | 9.45±1.38% | 5.29±0.43% | 2.55±0.21% |
| 5 | 42.82±0.07% | **37.62±0.38%** | **28.34±1.02%** | 19.07±1.07% | 12.55±0.81% | **9.52±1.37%** | **5.30±0.42%** | 2.55±0.18% |
| PLL | 41.30±0.06% | 34.85±0.40% | 28.30±0.38% | 19.00±0.60% | 12.33±0.93% | 8.49±1.35% | 5.22±0.62% | 2.47±0.25% |

Table 39: (Details of Fig. 15a) Test accuracy of Caltech-101 dataset applying superclass label corruption, where weight decay value $\lambda = 1 \times 10^{-3}$.

| Distillation Step | 0.0 | 0.1 | 0.3 | 0.5 | 0.6 | 0.7 | 0.8 | 0.9 |
|---|---|---|---|---|---|---|---|---|
| 1 (Teacher) | 95.08±0.03% | 93.49±0.25% | 90.46±0.30% | 84.37±0.83% | 78.80±0.87% | 67.78±1.17% | 47.45±1.78% | 16.44±0.99% |
| 2 | **95.14±0.05%** | **93.91±0.32%** | **90.98±0.24%** | 85.56±0.78% | 80.01±0.71% | 68.80±1.17% | 49.65±1.76% | 18.43±1.48% |
| 3 | 95.14±0.03% | 93.89±0.25% | 90.84±0.22% | 85.81±1.07% | 80.40±0.72% | 69.26±1.21% | 49.85±1.89% | 18.74±1.60% |
| 4 | 95.05±0.05% | 93.91±0.31% | 90.78±0.22% | 85.93±1.02% | 80.45±0.83% | 69.45±0.96% | 50.10±1.87% | 18.84±1.96% |
| 5 | 94.93±0.08% | 93.80±0.24% | 90.75±0.19% | **86.00±1.04%** | 80.59±0.85% | 69.64±0.77% | 50.12±1.85% | 18.84±1.91% |
| PLL | 94.35±0.16% | 92.68±0.50% | 89.73±0.52% | 85.52±2.40% | **81.20±0.65%** | **70.91±1.00%** | **51.50±2.92%** | **20.01±1.66%** |

Table 40: (Details of Fig. 15a) Test accuracy of Caltech-256 dataset applying superclass label corruption, where weight decay value $\lambda = 1 \times 10^{-4}$.

| Distillation Step | 0.0 | 0.1 | 0.3 | 0.5 | 0.6 | 0.7 | 0.8 | 0.9 |
|---|---|---|---|---|---|---|---|---|
| 1 (Teacher) | 83.65±0.03% | 81.73±0.22% | 77.97±0.24% | 72.64±0.30% | 69.58±0.07% | 63.71±0.58% | 52.62±0.75% | 28.39±1.56% |
| 2 | **83.69±0.06%** | **82.01±0.17%** | **78.17±0.18%** | 72.92±0.11% | 69.85±0.14% | 64.04±0.64% | 53.04±0.92% | 28.73±1.81% |
| 3 | 83.59±0.09% | 81.93±0.23% | 78.13±0.19% | **72.96±0.14%** | 69.91±0.18% | 64.09±0.64% | 53.05±1.00% | 28.75±1.80% |
| 4 | 83.59±0.06% | 81.84±0.24% | 78.05±0.14% | 72.96±0.13% | 69.91±0.22% | 64.09±0.61% | 53.08±0.91% | **28.78±1.79%** |
| 5 | 83.55±0.04% | 81.81±0.22% | 78.02±0.12% | 72.93±0.17% | 69.94±0.26% | 64.03±0.62% | 53.07±0.86% | 28.65±1.81% |
| PLL | 82.64±0.25% | 80.20±0.29% | 76.08±1.16% | 71.65±0.97% | 69.61±1.70% | **65.36±1.49%** | **54.24±1.23%** | 28.67±2.73% |

Table 41: (Details of Fig. 15a) Test accuracy of CIFAR-100 dataset applying superclass label corruption, where weight decay value $\lambda = 1 \times 10^{-4}$.

| Distillation Step | 0.0 | 0.1 | 0.3 | 0.5 | 0.6 | 0.7 | 0.8 | 0.9 |
|---|---|---|---|---|---|---|---|---|
| 1 (Teacher) | 71.20±0.02% | 66.72±0.19% | 61.01±0.16% | 53.95±0.22% | 48.46±0.06% | 41.28±0.10% | 29.81±0.36% | 13.89±0.27% |
| 2 | 72.25±0.20% | 69.67±0.19% | 66.36±0.29% | 62.48±0.15% | 59.26±0.60% | 53.96±0.31% | 43.38±0.28% | 22.69±0.81% |
| 3 | 72.44±0.03% | 69.84±0.18% | 66.68±0.14% | 62.68±0.08% | 59.45±0.62% | 54.12±0.36% | 44.06±0.62% | 23.33±0.67% |
| 4 | **72.47±0.11%** | 69.98±0.26% | 66.77±0.14% | 62.78±0.09% | 59.60±0.62% | 54.27±0.32% | 44.30±0.63% | 23.60±0.72% |
| 5 | 72.45±0.14% | **70.04±0.22%** | 66.85±0.11% | 62.85±0.17% | 59.68±0.59% | 54.40±0.32% | 44.55±0.69% | 23.74±0.77% |
| PLL | 69.90±0.12% | 68.57±0.21% | **67.50±0.27%** | **64.65±0.30%** | **62.26±0.38%** | **58.38±0.35%** | **49.89±0.71%** | **28.20±1.41%** |

Table 42: (Details of Fig. 15a) Test accuracy of Flowers-102 dataset applying superclass label corruption, where weight decay value $\lambda = 1 \times 10^{-3}$.

| Distillation Step | 0.0 | 0.1 | 0.3 | 0.5 | 0.6 | 0.7 | 0.8 | 0.9 |
|---|---|---|---|---|---|---|---|---|
| 1 (Teacher) | **88.24±0.16%** | 81.83±0.41% | 70.20±0.42% | 55.36±1.00% | 46.60±1.77% | 35.72±1.45% | 24.97±0.28% | 10.07±0.18% |
| 2 | 87.97±0.09% | 83.82±0.35% | 75.75±0.37% | 64.31±0.63% | 55.78±2.44% | 45.72±2.78% | 32.25±1.00% | 12.39±1.09% |
| 3 | 87.61±0.23% | 84.54±0.51% | 77.81±0.33% | 67.39±1.04% | 57.42±2.13% | 48.24±2.20% | 33.59±0.93% | 13.33±1.47% |
| 4 | 87.25±0.14% | **84.61±0.35%** | 78.24±0.56% | 68.14±1.39% | 58.20±1.80% | 49.05±1.51% | 33.76±1.01% | 13.86±1.74% |
| 5 | 86.60±0.12% | 84.61±0.42% | 78.59±0.73% | 68.43±1.27% | 58.37±1.85% | 48.95±1.48% | 33.66±0.86% | 14.12±1.69% |
| PLL | 85.56±0.44% | 81.11±0.53% | **79.02±0.94%** | **72.48±2.64%** | **64.87±2.76%** | **54.31±1.04%** | **41.47±1.25%** | **15.23±2.53%** |

Table 43: (Details of Fig. 15a) Test accuracy of Food-101 dataset applying superclass label corruption, where weight decay value $\lambda = 1 \times 10^{-4}$.

| Distillation Step | 0.0 | 0.1 | 0.3 | 0.5 | 0.6 | 0.7 | 0.8 | 0.9 |
|---|---|---|---|---|---|---|---|---|
| 1 (Teacher) | 63.60±0.03% | 61.95±0.12% | 58.22±0.25% | 52.88±0.24% | 48.83±0.11% | 43.01±0.13% | 33.23±0.75% | 15.88±0.22% |
| 2 | **63.66±0.05%** | **62.41±0.19%** | 59.86±0.31% | 56.24±0.18% | 53.45±0.21% | 49.62±0.28% | 39.57±3.93% | 21.95±3.54% |
| 3 | 63.38±0.06% | 62.27±0.15% | 59.79±0.29% | 56.25±0.21% | 53.47±0.23% | 49.70±0.31% | 39.74±3.96% | 22.18±3.64% |
| 4 | 63.27±0.04% | 62.14±0.17% | 59.70±0.28% | 56.26±0.21% | 53.49±0.22% | 49.76±0.29% | 39.85±4.04% | 22.33±3.74% |
| 5 | 63.12±0.04% | 62.08±0.12% | 59.65±0.26% | 56.26±0.19% | 53.51±0.22% | 49.83±0.28% | 39.95±4.08% | 22.45±3.81% |
| PLL | 63.41±0.03% | 61.82±0.15% | **60.10±0.41%** | **57.85±0.11%** | **55.89±0.33%** | **53.26±0.04%** | **43.58±6.13%** | **27.06±7.03%** |

Table 44: (Details of Fig. 15a) Test accuracy of StanfordCars dataset applying superclass label corruption, where weight decay value $\lambda = 1 \times 10^{-3}$.

| Distillation Step | 0.0 | 0.1 | 0.3 | 0.5 | 0.6 | 0.7 | 0.8 | 0.9 |
|---|---|---|---|---|---|---|---|---|
| 1 (Teacher) | **43.30±0.00%** | 37.62±0.23% | 29.05±0.04% | 19.77±0.22% | 15.44±0.29% | 11.40±0.19% | 6.77±0.13% | 3.16±0.27% |
| 2 | 42.87±0.10% | **39.05±0.09%** | 32.19±0.25% | 23.10±0.25% | 18.71±0.36% | 13.75±0.09% | 8.31±0.18% | 3.47±0.43% |
| 3 | 41.74±0.12% | 38.54±0.36% | **32.83±0.54%** | 24.28±0.26% | 19.20±0.60% | 14.58±0.05% | 8.64±0.18% | **3.57±0.53%** |
| 4 | 40.37±0.13% | 37.44±0.34% | 32.49±0.55% | **24.34±0.27%** | 19.21±0.87% | 14.70±0.16% | 8.69±0.20% | 3.54±0.54% |
| 5 | 39.03±0.19% | 36.20±0.27% | 31.76±0.58% | 23.91±0.37% | 19.05±0.77% | 14.61±0.20% | 8.59±0.29% | 3.46±0.53% |
| PLL | 40.60±0.36% | 35.96±0.20% | 30.32±0.45% | 23.65±0.45% | **19.43±0.75%** | **15.18±0.95%** | **8.80±0.12%** | 3.26±0.62% |

Table 45: (Details of Fig. 15b) Test accuracy of Caltech-101 dataset applying symmetric label corruption, where weight decay value $\lambda = 1 \times 10^{-3}$.

| Distillation Step | 0.0 | 0.1 | 0.3 | 0.5 | 0.6 | 0.7 | 0.8 | 0.9 |
|---|---|---|---|---|---|---|---|---|
| 1 (Teacher) | 95.08±0.03% | 93.49±0.16% | 91.13±0.42% | 87.96±0.19% | 84.60±0.56% | 80.80±0.55% | 69.74±0.41% | 46.04±0.60% |
| 2 | **95.14±0.05%** | **93.74±0.23%** | 91.78±0.42% | 88.56±0.47% | 85.60±0.68% | 81.45±0.36% | 71.31±0.69% | 47.77±1.18% |
| 3 | 95.14±0.03% | 93.63±0.32% | 91.92±0.31% | 88.69±0.65% | 85.81±0.90% | 81.36±0.35% | 71.58±0.85% | 48.23±1.51% |
| 4 | 95.05±0.05% | 93.53±0.26% | **91.97±0.29%** | 88.75±0.67% | **86.06±0.88%** | 81.57±0.38% | 71.66±0.76% | 48.43±1.48% |
| 5 | 94.93±0.08% | 93.49±0.12% | 91.97±0.14% | **88.79±0.62%** | 86.06±1.04% | 81.55±0.27% | 71.68±0.67% | 48.48±1.41% |
| PLL | 94.35±0.16% | 92.72±0.57% | 91.51±0.57% | 88.61±0.71% | 85.62±2.28% | **82.53±0.28%** | **73.64±1.58%** | **51.69±3.19%** |

Table 46: (Details of Fig. 15b) Test accuracy of Caltech-256 dataset applying symmetric label corruption, where weight decay value $\lambda = 1 \times 10^{-4}$.

| Distillation Step | 0.0 | 0.1 | 0.3 | 0.5 | 0.6 | 0.7 | 0.8 | 0.9 |
|---|---|---|---|---|---|---|---|---|
| 1 (Teacher) | 83.65±0.03% | 81.77±0.13% | 78.60±0.39% | 74.50±0.29% | 71.48±0.13% | 66.48±0.74% | 57.47±0.44% | **34.42±0.16%** |
| 2 | **83.69±0.06%** | **82.03±0.34%** | **78.79±0.41%** | **74.75±0.26%** | 71.68±0.11% | 66.81±0.73% | 57.53±0.33% | 34.39±0.13% |
| 3 | 83.59±0.09% | 81.91±0.31% | 78.77±0.40% | 74.75±0.27% | **71.70±0.16%** | 66.78±0.70% | 57.45±0.28% | 34.36±0.14% |
| 4 | 83.59±0.06% | 81.90±0.30% | 78.76±0.37% | 74.70±0.31% | 71.63±0.19% | 66.80±0.65% | 57.43±0.32% | 34.32±0.16% |
| 5 | 83.55±0.04% | 81.82±0.32% | 78.71±0.39% | 74.64±0.27% | 71.57±0.20% | 66.87±0.69% | 57.43±0.33% | 34.25±0.18% |
| PLL | 82.64±0.25% | 80.41±0.71% | 76.47±0.44% | 73.29±0.65% | 70.59±0.18% | **67.80±1.06%** | **58.23±0.31%** | 32.47±1.37% |

Table 47: (Details of Fig. 15b) Test accuracy of CIFAR-100 dataset applying symmetric label corruption, where weight decay value $\lambda = 1 \times 10^{-4}$.

| Distillation Step | 0.0 | 0.1 | 0.3 | 0.5 | 0.6 | 0.7 | 0.8 | 0.9 |
|---|---|---|---|---|---|---|---|---|
| 1 (Teacher) | 71.20±0.02% | 67.18±0.06% | 62.92±0.37% | 57.28±0.14% | 53.12±0.17% | 47.33±0.03% | 37.33±0.28% | 19.89±0.13% |
| 2 | 72.25±0.06% | 70.21±0.12% | 67.52±0.09% | 64.52±0.11% | 62.35±0.25% | 58.50±0.30% | 51.96±0.68% | 34.94±0.47% |
| 3 | 72.44±0.03% | 70.45±0.10% | 67.72±0.06% | 64.69±0.12% | 62.53±0.23% | 58.73±0.24% | 52.29±0.60% | 35.99±0.53% |
| 4 | **72.47±0.11%** | 70.56±0.17% | 67.73±0.08% | 64.82±0.18% | 62.60±0.24% | 58.88±0.23% | 52.51±0.58% | 36.39±0.51% |
| 5 | 72.45±0.14% | **70.63±0.17%** | 67.72±0.13% | 64.87±0.19% | 62.67±0.24% | 58.95±0.27% | 52.66±0.55% | 36.58±0.54% |
| PLL | 69.90±0.12% | 69.13±0.18% | **67.93±0.10%** | **65.76±0.19%** | **64.08±0.11%** | **61.41±0.17%** | **57.02±0.19%** | **41.18±1.27%** |

Table 48: (Details of Fig. 15b) Test accuracy of Flowers-102 dataset applying symmetric label corruption, where weight decay value $\lambda = 1 \times 10^{-3}$.

| Distillation Step | 0.0 | 0.1 | 0.3 | 0.5 | 0.6 | 0.7 | 0.8 | 0.9 |
|---|---|---|---|---|---|---|---|---|
| 1 (Teacher) | **88.24±0.16%** | 81.57±0.58% | 69.77±1.04% | 55.42±2.00% | 48.04±1.45% | 37.97±0.23% | 25.29±0.21% | 11.60±1.53% |
| 2 | 87.97±0.09% | 83.95±1.07% | 76.80±0.26% | 66.37±2.47% | 58.33±2.16% | 48.66±1.08% | 34.28±0.68% | 14.97±2.28% |
| 3 | 87.61±0.23% | 84.77±0.76% | 78.20±0.40% | 67.35±2.00% | 61.27±2.60% | 51.60±1.35% | 35.62±1.23% | 15.52±1.85% |
| 4 | 87.25±0.14% | **84.80±0.49%** | 78.69±0.73% | 67.58±2.04% | 62.81±2.76% | 52.39±1.90% | 36.14±1.17% | 15.78±1.74% |
| 5 | 86.60±0.12% | 84.67±0.46% | 78.66±0.70% | 67.91±1.62% | 63.24±2.94% | 52.48±2.19% | 36.34±1.13% | 15.78±1.74% |
| PLL | 85.56±0.44% | 81.41±0.41% | **79.74±0.32%** | **73.56±1.51%** | **66.73±2.47%** | **59.67±1.68%** | **42.71±2.37%** | **17.97±4.24%** |

Table 49: (Details of Fig. 15b) Test accuracy of Food-101 dataset applying symmetric label corruption, where weight decay value $\lambda = 1 \times 10^{-4}$.

| Distillation Step | 0.0 | 0.1 | 0.3 | 0.5 | 0.6 | 0.7 | 0.8 | 0.9 |
|---|---|---|---|---|---|---|---|---|
| 1 (Teacher) | 63.60±0.03% | 62.08±0.09% | 58.54±0.22% | 53.82±0.21% | 50.21±0.09% | 44.87±0.19% | 36.15±0.25% | 18.91±0.23% |
| 2 | **63.66±0.05%** | **62.61±0.09%** | **60.20±0.03%** | 56.98±0.09% | 54.72±0.25% | 51.24±0.31% | 44.59±0.04% | 28.08±0.19% |
| 3 | 63.38±0.06% | 62.48±0.07% | 60.09±0.04% | 57.01±0.08% | 54.75±0.18% | 51.31±0.35% | 44.82±0.01% | 28.30±0.20% |
| 4 | 63.27±0.04% | 62.39±0.03% | 59.98±0.07% | 56.98±0.08% | 54.75±0.21% | 51.32±0.35% | 44.90±0.06% | 28.48±0.19% |
| 5 | 63.12±0.04% | 62.31±0.09% | 59.94±0.06% | 56.98±0.09% | 54.78±0.20% | 51.33±0.36% | 45.00±0.06% | 28.56±0.19% |
| PLL | 63.41±0.03% | 61.98±0.18% | 60.16±0.07% | **57.98±0.25%** | **56.22±0.07%** | **54.04±0.14%** | **49.17±0.10%** | **34.73±0.58%** |

Table 50: (Details of Fig. 15b) Test accuracy of StanfordCars dataset applying symmetric label corruption, where weight decay value $\lambda = 1 \times 10^{-3}$.

| Distillation Step | 0.0 | 0.1 | 0.3 | 0.5 | 0.6 | 0.7 | 0.8 | 0.9 |
|---|---|---|---|---|---|---|---|---|
| 1 (Teacher) | **43.30±0.00%** | 37.53±0.40% | 28.93±0.27% | 19.92±0.11% | 15.40±0.24% | 10.75±0.22% | 6.30±0.29% | 2.79±0.17% |
| 2 | 42.87±0.10% | **39.15±0.11%** | 31.79±0.21% | 23.35±0.21% | 18.36±0.18% | 13.52±0.23% | 7.74±0.40% | 3.20±0.25% |
| 3 | 41.74±0.12% | 38.87±0.02% | **32.24±0.23%** | **24.20±0.14%** | 19.25±0.25% | 14.16±0.50% | 8.09±0.53% | 3.34±0.32% |
| 4 | 40.37±0.13% | 38.07±0.02% | 31.96±0.46% | 24.11±0.07% | 19.47±0.34% | 14.31±0.52% | 8.24±0.67% | 3.35±0.36% |
| 5 | 39.03±0.19% | 36.91±0.15% | 31.12±0.21% | 23.72±0.12% | 19.24±0.14% | 14.15±0.68% | 8.20±0.80% | 3.39±0.41% |
| PLL | 40.60±0.36% | 36.20±0.09% | 30.18±0.31% | 24.00±0.18% | **20.25±0.31%** | **15.37±0.31%** | **8.93±1.12%** | **3.73±0.61%** |

Table 51: (Details of Fig. 15c) Test accuracy of Caltech-101 dataset applying asymmetric label corruption, where weight decay value $\lambda = 1 \times 10^{-3}$.

| Distillation Step | 0.0 | 0.1 | 0.3 | 0.5 | 0.6 | 0.7 | 0.8 | 0.9 |
|---|---|---|---|---|---|---|---|---|
| 1 (Teacher) | 95.08±0.03% | 93.64±0.23% | 91.13±0.24% | 87.69±0.41% | 85.08±0.71% | 80.32±1.11% | 70.97±0.95% | 45.18±0.94% |
| 2 | **95.14±0.05%** | **93.74±0.27%** | 91.65±0.61% | 88.33±0.42% | 86.04±0.36% | 80.91±0.96% | 72.04±0.60% | 46.74±1.32% |
| 3 | 95.14±0.03% | 93.57±0.34% | **91.67±0.61%** | 88.56±0.59% | 86.16±0.26% | 80.89±0.88% | 72.24±0.70% | 47.02±1.08% |
| 4 | 95.05±0.05% | 93.55±0.24% | 91.67±0.63% | 88.61±0.49% | 86.39±0.19% | 80.86±0.87% | 72.35±0.78% | 47.41±1.21% |
| 5 | 94.93±0.08% | 93.57±0.27% | 91.61±0.69% | 88.61±0.40% | 86.44±0.28% | 80.84±0.93% | 72.20±0.95% | 47.48±1.26% |
| PLL | 94.35±0.16% | 92.70±0.12% | 90.78±0.71% | **88.67±0.76%** | **87.23±0.62%** | **81.32±0.72%** | **72.47±2.65%** | **48.68±2.60%** |

Table 52: (Details of Fig. 15c) Test accuracy of Caltech-256 dataset applying asymmetric label corruption, where weight decay value $\lambda = 1 \times 10^{-4}$.

| Distillation Step | 0.0 | 0.1 | 0.3 | 0.5 | 0.6 | 0.7 | 0.8 | 0.9 |
|---|---|---|---|---|---|---|---|---|
| 1 (Teacher) | 83.65±0.03% | 81.65±0.17% | 78.50±0.20% | 74.38±0.18% | 71.22±0.28% | 66.54±0.33% | 57.43±0.73% | 35.15±1.09% |
| 2 | **83.69±0.06%** | **81.88±0.10%** | **78.69±0.21%** | 74.87±0.47% | 71.33±0.41% | 66.83±0.27% | 57.79±0.89% | **35.28±1.16%** |
| 3 | 83.59±0.09% | 81.74±0.14% | 78.63±0.27% | 74.86±0.55% | 71.41±0.44% | 66.80±0.23% | 57.70±1.05% | 35.27±1.30% |
| 4 | 83.59±0.06% | 81.72±0.14% | 78.57±0.27% | **74.88±0.51%** | 71.45±0.42% | 66.78±0.23% | 57.72±1.09% | 35.27±1.31% |
| 5 | 83.55±0.04% | 81.71±0.11% | 78.54±0.29% | 74.88±0.51% | **71.50±0.40%** | 66.79±0.23% | 57.71±1.09% | 35.27±1.25% |
| PLL | 82.64±0.25% | 80.41±0.21% | 75.86±0.49% | 73.98±1.23% | 70.80±0.94% | **68.01±0.90%** | **57.83±2.36%** | 34.60±2.41% |

Table 53: (Details of Fig. 15c) Test accuracy of CIFAR-100 dataset applying asymmetric label corruption, where weight decay value $\lambda = 1 \times 10^{-4}$.

| Distillation Step | 0.0 | 0.1 | 0.3 | 0.5 | 0.6 | 0.7 | 0.8 | 0.9 |
|---|---|---|---|---|---|---|---|---|
| 1 (Teacher) | 71.20±0.02% | 67.28±0.17% | 63.12±0.16% | 57.15±0.11% | 53.07±0.16% | 46.98±0.24% | 37.39±0.11% | 19.93±0.31% |
| 2 | 72.25±0.06% | 70.38±0.16% | 67.68±0.40% | 64.52±0.18% | 62.22±0.08% | 58.61±0.54% | 52.52±0.42% | 34.98±0.37% |
| 3 | 72.44±0.03% | 70.74±0.19% | 67.76±0.40% | 64.66±0.13% | 62.36±0.05% | 58.86±0.51% | 53.07±0.48% | 36.28±0.17% |
| 4 | **72.47±0.11%** | 70.85±0.16% | 67.78±0.34% | 64.75±0.17% | 62.44±0.10% | 58.92±0.48% | 53.28±0.22% | 36.58±0.11% |
| 5 | 72.45±0.14% | **70.89±0.15%** | 67.75±0.36% | 64.81±0.13% | 62.52±0.12% | 58.98±0.51% | 53.37±0.18% | 36.83±0.10% |
| PLL | 69.90±0.12% | 69.20±0.05% | **68.00±0.23%** | **65.71±0.19%** | **63.85±0.23%** | **61.62±0.45%** | **57.10±0.10%** | **41.51±0.53%** |

Table 54: (Details of Fig. 15c) Test accuracy of Flowers-102 dataset applying asymmetric label corruption, where weight decay value $\lambda = 1 \times 10^{-3}$.

| Distillation Step | 0.0 | 0.1 | 0.3 | 0.5 | 0.6 | 0.7 | 0.8 | 0.9 |
|---|---|---|---|---|---|---|---|---|
| 1 (Teacher) | **88.24±0.16%** | 81.14±0.12% | 70.62±1.32% | 56.08±2.00% | 49.08±1.38% | 38.24±1.94% | 24.93±0.44% | 12.09±1.12% |
| 2 | 87.97±0.09% | 83.66±0.17% | 76.60±0.57% | 66.80±3.26% | 60.33±2.80% | 49.97±3.47% | 36.08±1.81% | 17.03±1.42% |
| 3 | 87.61±0.23% | 84.15±0.26% | 78.04±0.49% | 68.07±2.88% | 62.06±1.72% | 51.14±3.14% | 36.67±1.74% | 17.75±1.20% |
| 4 | 87.25±0.14% | **84.28±0.32%** | 78.56±0.65% | 68.66±2.66% | 62.09±1.37% | 51.37±3.17% | 36.70±1.63% | 18.07±1.09% |
| 5 | 86.60±0.12% | 84.22±0.52% | 78.95±1.07% | 68.79±2.42% | 62.06±1.10% | 51.47±3.47% | 36.73±1.64% | 18.17±1.19% |
| PLL | 85.56±0.44% | 81.27±0.21% | **79.67±0.30%** | **74.93±2.52%** | **69.48±1.30%** | **59.93±1.54%** | **45.29±1.47%** | **20.59±2.17%** |

Table 55: (Details of Fig. 15c) Test accuracy of Food-101 dataset applying asymmetric label corruption, where weight decay value $\lambda = 1 \times 10^{-4}$.

| Distillation Step | 0.0 | 0.1 | 0.3 | 0.5 | 0.6 | 0.7 | 0.8 | 0.9 |
|---|---|---|---|---|---|---|---|---|
| 1 (Teacher) | 63.60±0.03% | 62.02±0.16% | 58.53±0.11% | 53.60±0.12% | 50.25±0.15% | 44.94±0.20% | 35.84±0.20% | 19.06±0.39% |
| 2 | **63.66±0.05%** | **62.58±0.01%** | **60.07±0.19%** | 57.03±0.14% | 54.64±0.24% | 51.08±0.44% | 44.21±0.20% | 28.16±0.77% |
| 3 | 63.38±0.06% | 62.47±0.05% | 59.98±0.24% | 56.98±0.11% | 54.68±0.22% | 51.13±0.47% | 44.36±0.27% | 28.46±0.80% |
| 4 | 63.27±0.04% | 62.34±0.04% | 59.92±0.30% | 56.96±0.12% | 54.67±0.17% | 51.15±0.42% | 44.49±0.28% | 28.59±0.81% |
| 5 | 63.12±0.04% | 62.27±0.06% | 59.89±0.29% | 56.95±0.12% | 54.68±0.18% | 51.16±0.39% | 44.56±0.32% | 28.73±0.87% |
| PLL | 63.41±0.03% | 61.95±0.12% | 59.93±0.21% | **57.94±0.12%** | **56.34±0.13%** | **53.97±0.29%** | **49.18±0.23%** | **35.36±1.76%** |

Table 56: (Details of Fig. 15c) Test accuracy of StanfordCars dataset applying asymmetric label corruption, where weight decay value $\lambda = 1 \times 10^{-3}$.

| Distillation Step | 0.0 | 0.1 | 0.3 | 0.5 | 0.6 | 0.7 | 0.8 | 0.9 |
|---|---|---|---|---|---|---|---|---|
| 1 (Teacher) | **43.30±0.00%** | 37.32±0.16% | 28.72±0.26% | 19.63±0.21% | 15.18±0.32% | 11.01±0.13% | 6.18±0.20% | 2.71±0.04% |
| 2 | 42.87±0.10% | **39.03±0.06%** | 31.85±0.22% | 23.23±0.20% | 18.32±0.47% | 13.75±0.28% | 7.71±0.25% | 3.27±0.16% |
| 3 | 41.74±0.12% | 38.66±0.08% | **32.29±0.28%** | 24.12±0.32% | 19.12±0.62% | 14.66±0.14% | 7.95±0.20% | 3.30±0.23% |
| 4 | 40.37±0.13% | 37.75±0.12% | 31.99±0.14% | 24.08±0.21% | 19.06±0.72% | 14.90±0.10% | 7.91±0.28% | 3.32±0.26% |
| 5 | 39.03±0.19% | 36.62±0.10% | 31.24±0.20% | 23.82±0.25% | 18.78±0.72% | 14.69±0.08% | 7.84±0.40% | 3.33±0.27% |
| PLL | 40.60±0.36% | 36.02±0.19% | 30.40±0.42% | **24.64±0.29%** | **19.52±0.66%** | **15.30±0.44%** | **8.68±0.33%** | **3.36±0.32%** |

