# OpenReview forum: "Rethinking Self-Distillation: Label Averaging and Enhanced Soft Label Refinement with Partial Labels"
_ICLR.cc/2025/Conference — ICLR 2025 Poster_

### Official Review · Reviewer_CZ26 · 2024-10-29

**Soundness:** 3
**Presentation:** 2
**Contribution:** 2
**Rating:** 6
**Confidence:** 2

**Summary:**

In this paper, the authors proposed introduce a single-round self-distillation method using refined partial labels from the teacher’s top two softmax outputs, referred to as the PLL student model. This approach replicates the benefits of multiround distillation in a single round. Experiments on several datasets demonstrate the proposed method is effective.

**Strengths:**

The analysis of self-distillation in linear probing reveals that self-distillation effectively performs label averaging among instances with high feature correlations when generating predictions on the training data.

**Weaknesses:**

Here are a few questions I would like to ask the authors:

1. What are the clear insights or significances of Sections 2.2, 3, and 4 in relation to the proposed method in Section 5? Could the authors summarize the roles and conclusions of Sections 2.2, 3, and 4, and their implications for Section 5? These sections are not sufficiently clear.

2. In Section 5, the authors state, "selecting the top two labels with the highest values and assigning a weight of 1/2 to each, setting all the other entries to zero." What is the rationale behind this approach?

3. The experiments use ResNet34 as the backbone for linear probe experiments; however, training ResNet34 with all parameters is not particularly time-consuming or resource-intensive. Should a backbone with a larger parameter count be chosen for the experiments?

4. When comparing existing PLL methods, the authors claim, "Our method differs by directly employing the refined partial labels derived from the teacher’s outputs, achieving the same benefits as multi-round distillation in just one round." Does this imply that other methods incorporating the teacher's output could achieve similar or even better results? Are there corresponding experiments to support this?

**Questions:**

Please see the weaknesses above.

---

> ### Author Response · Authors · 2024-11-18
> **Response to Reviewer CZ26 (1/3)**
>
> We sincerely thank the reviewer for the constructive feedback. Please see our responses to the reviewer’s questions.
>
> >**W1. What are the clear insights or significance of Sections 2.2, 3, and 4 in relation to the proposed method in Section 5? Could the authors summarize the roles and conclusions of Sections 2.2, 3, and 4, and their implications for Section 5? These sections are not sufficiently clear.**
>
> >**W2. In Section 5, the authors state, "selecting the top two labels with the highest values and assigning a weight of 1/2 to each, setting all the other entries to zero." What is the rationale behind this approach?**
>
> Sections 2.2, 3, and 4 are crucial for understanding the development and significance of the proposed method in Section 5. We introduced the PLL (Partial Label Learning) student model in Section 5 based on the intuition gained from our closed-form analysis of multi-round self-distillation effects in Section 3 and the resulting robustness to label noise observed in Section 4. Specifically, the PLL student model is designed to replicate the benefits of multi-round self-distillation—particularly its robustness to label noise—in a single round.
>
> In Section 3, we analyzed the effect of multi-round self-distillation and provided a quantified closed-form solution for the output of the $t$-th student model. This analysis revealed that multi-round self-distillation induces label averaging effects among instances with high feature correlations when generating predictions on the training data. This means that as the number of distillation rounds increases, the model's output predictions gradually form clusters corresponding to each ground-truth label based on input feature correlations. This clustering effect is illustrated in Figure 2, where the outputs become more similar for instances sharing the same true label as $t$ increases.
>
> In Section 4, we explored how this clustering effect enhances robustness to label noise. The label averaging makes the model's predictions less overfitted to the given (possibly noisy) labels and more influenced by the average labels among highly correlated instances. Recognizing that each distillation round gradually increases the weight on the average labels while decreasing the weight on the individual labels, we analyzed how many distillation rounds are sufficient for the student model to make correct predictions for all training instances, including those with noisy labels. This is summarized in Theorem 4.1, which captures the relationship between label corruption rates and the effectiveness of the label averaging process over multiple distillation rounds.
>
> Building on these insights, the PLL student model proposed in Section 5 aims to replicate the behavior of multi-round self-distillation through refinement of the teacher's output and a single step of self-distillation. The label averaging effect from multi-round self-distillation shifts the softmax output away from a one-hot vector of the given label toward the average labels among instances sharing the same ground-truth label. For clean samples, this averaging effect slightly reduces the confidence in the correct label but maintains the correct prediction as $t$ increases. For noisy samples, it decreases the confidence in the incorrect (noisy) label and moves the prediction toward the correct label.
>
> To facilitate this process efficiently, our PLL student model refines the teacher's output by selecting the top two labels with the highest values, assigning a weight of 1/2 to each, and setting all other entries to zero. Under mild conditions on the label corruption matrix—as stated in Theorem 5.1—we show that the top-two list obtained from the teacher's output always includes the ground-truth label: it ranks first for clean samples and second for noisy samples. Assigning equal weights to the top two labels balances the influence of the teacher's confidence and mitigates overconfidence in potentially incorrect labels due to noise.
>
> Using these refined targets (two-hot vectors), a single round of self-distillation—through label averaging among highly correlated inputs—ensures that the model's softmax outputs assign the highest probability to the ground-truth label for all training instances, achieving 100% accuracy. This is possible because the average of the two-hot vectors from instances of the same ground-truth class has its highest value at the ground-truth label, since the two-hot vectors always include the true label for both clean and noisy samples.
>
> Thus, the PLL student model effectively replicates the advantages of multi-round self-distillation in just a single round. This approach not only enhances computational efficiency but also maintains or improves performance, especially in high label-noise regimes.

---

> ### Author Response · Authors · 2024-11-18
> **Response to Reviewer CZ26 (2/3)**
>
> > **W3. The experiments use ResNet34 as the backbone for linear probe experiments; however, training ResNet34 with all parameters is not particularly time-consuming or resource-intensive. Should a backbone with a larger parameter count be chosen for the experiments?**
>
> Thank the reviewer for the insightful suggestion. We acknowledge that ResNet34 is not a particularly large model, and training it with all parameters is not highly resource-intensive. However, we chose ResNet34 for our experiments to effectively observe the progressive gains of multi-round self-distillation.
>
> To address the reviewer’s concern, we also conducted additional experiments using a larger backbone—a pretrained ViT-B (Vision Transformer) model—as a fixed feature extractor. The test accuracies (%) on the CIFAR-100 dataset across different label corruption rates are presented in the table below:
>
> |DistillationStep|0.0|0.1|0.3|0.5|0.6|0.7|0.8|0.9|
> |----------------|----|----|----|----|----|----|----|----|
> |1(Teacher)|79.21±0.02%|74.26±0.24%|67.14±0.06%|59.42±0.40%|54.32±0.34%|46.00±0.46%|34.55±0.28%|17.19±0.50%|
> |2|79.99±0.10%|74.77±0.28%|72.95±3.06%|71.29±0.64%|68.94±0.49%|64.40±0.56%|55.53±0.98%|33.43±0.56%|
> |3|80.24±0.12%|74.89±0.31%|73.28±3.18%|71.76±0.34%|69.22±0.42%|65.16±0.61%|56.88±1.17%|36.10±0.80%|
> |4|80.28±0.13%|**74.91±0.28%**|73.34±3.17%|**71.87±0.41%**|69.32±0.40%|65.41±0.58%|57.26±1.29%|36.84±0.98%|
> |5|**80.29±0.16%**|74.90±0.27%|**73.40±3.11%**|71.86±0.42%|69.37±0.41%|65.54±0.63%|57.44±1.27%|**37.22±1.11%**|
> |PLL|77.48±0.13%|73.78±0.50%|72.69±2.82%|71.56±0.15%|**69.76±0.45%**|**66.36±0.29%**|**58.42±0.58%**|36.04±1.46%|
>
> Using a larger backbone like ViT-B enhances feature extraction capabilities, resulting in a greater disparity between intra-class and inter-class feature correlations. For example, when calculating feature correlations on the CIFAR-100 dataset using the pretrained ViT-B model, we observed that the average intra-class feature correlation increased to 0.35, compared to 0.25 with ResNet34. This higher intra-class correlation amplifies the clustering effect of self-distillation on model predictions, allowing significant performance improvements to be achieved with fewer distillation steps, as implied by our Theorem 4.1.
>
> In experiments with the larger backbone (ViT-B), we found that most of the distillation gains occur within the first few rounds. As shown in the table above, nearly all performance improvements are observed in the first and second distillation steps when using ViT-B, although additional distillation steps still bring slight gains in high noise rate regimes. The PLL student model also effectively achieves the gains of multi-round self-distillation in a single round in high label-noise regimes for the ViT-B backbone. These additional experiments confirm that our approach is effective with larger models, further validating the versatility and robustness of our method.

---

> ### Author Response · Authors · 2024-11-18
> **Response to Reviewer CZ26 (3/3)**
>
> >**W4. When comparing existing PLL methods, the authors claim, "Our method differs by directly employing the refined partial labels derived from the teacher’s outputs, achieving the same benefits as multi-round distillation in just one round." Does this imply that other methods incorporating the teacher's output could achieve similar or even better results? Are there corresponding experiments to support this?**
>
> Partial Label Learning (PLL) is a type of weakly supervised learning where training instances are annotated with a set of candidate labels rather than a single ground-truth label. The goal of PLL is to train a model capable of predicting the ground-truth label for unseen data using this partially labeled dataset. Some existing PLL methods incorporate a teacher-student framework, leveraging the teacher model to refine or guide the candidate label distribution to improve the student model's ability to generalize  [1, 2, 3, 4, 5, 6, 7]. For instance, the teacher model may provide soft pseudo-labels or confidence scores to weight the candidate labels, enabling the student model to focus more on the most plausible label. This interaction helps mitigate ambiguity in candidate sets by progressively narrowing down the label distribution during training, ultimately enhancing the accuracy and robustness of the PLL framework.
>
> Our method differs from these existing PLL approaches in a key aspect. While traditional PLL studies focus on utilizing the given partially labeled dataset to train a model that predicts the ground-truth label precisely, our research emphasizes **constructing the candidate label set itself from the teacher’s output**, particularly in the context of noisy supervised training. We demonstrate that a noisy supervised training problem can be reformulated as a PLL problem by leveraging a pretrained feature extractor and linear probing.
>
> In our experiments (as shown in Figure 7 in Appendix D.2), we observe that features extracted by a pretrained ResNet34 exhibit high feature correlation among instances of the same class and low feature correlation between instances of different classes. In Section 3, we show that the principle behind self-distillation lies in prediction averaging based on feature correlation. Thus, even for noisy instances—where the ground-truth label differs from the given label—the outputs are influenced by the average predictions of other instances within the same ground-truth class. This effect causes the output probability at the ground-truth label position to become larger, significantly increasing the likelihood that the ground-truth label is included in the size-2 candidate sets derived from the teacher's outputs.
>
> By directly employing the refined partial labels from the teacher's outputs, our method effectively achieves the benefits of multi-round self-distillation in just one round. This approach is specifically designed to correct label noise by constructing candidate label sets that are more likely to include the true label, without relying on additional PLL techniques that adjust the candidate labels during training.
>
> Regarding the reviewer’s question, while other PLL methods incorporating the teacher's output aim to improve performance by refining the candidate label distribution, they typically focus on disambiguating given candidate labels rather than constructing the candidate set itself from the teacher's predictions. Our approach is unique in that it leverages the teacher's outputs to build the candidate label sets, directly addressing the issue of label noise in supervised learning.
>
> We did not conduct experiments comparing our method with other PLL approaches that utilize the teacher's output because the objectives and problem settings differ. Our research contributes a novel perspective by demonstrating how to construct candidate label sets from the teacher's outputs to enhance robustness to label noise in linear probing with self-distillation.
>
> [1] Xia, Shiyu, et al. "Towards effective visual representations for partial-label learning." *CVPR 2023*.
>
> [2] Li, Beibei, et al. "AsyCo: An Asymmetric Dual-task Co-training Model for Partial-label Learning." *arXiv preprint, 2024*.
>
> [3] Xu, Ning, et al. "Aligned Objective for Soft-Pseudo-Label Generation in Supervised Learning." *ICML 2024*.
>
> [4] Wang, Guangtai, et al. "Dealing with partial labels by knowledge distillation." *Pattern Recognition, 2025*.
>
> [5] Wang, Haobo, et al. "Pico: Contrastive label disambiguation for partial label learning." *ICLR 2022*.
>
> [6] Wu, Dong-Dong, Deng-Bao Wang, and Min-Ling Zhang. "Distilling Reliable Knowledge for Instance-Dependent Partial Label Learning." Proceedings of the AAAI Conference on Artificial Intelligence. Vol. 38. No. 14. 2024.
>
> [7] Li, Wei, et al. "Generalized Contrastive Partial Label Learning for Cross-Subject EEG-Based Emotion Recognition." *IEEE Transactions on Instrumentation and Measurement, 2024*.

---

> > ### Comment · Reviewer_CZ26 · 2024-11-25
> >
> > Thank you for your detailed response and I decide to raise my score.
> > Besides, we suggest that the authors will add a detailed discussion of existing PLL methods in the final version as in the responce.

---

### Official Review · Reviewer_g4HH · 2024-11-02

**Soundness:** 4
**Presentation:** 3
**Contribution:** 3
**Rating:** 8
**Confidence:** 4

**Summary:**

This paper examines the multi-round self-distillation for multi-class classification in the context of linear probing. By approximating the softmax function as a linear function and considering the feature correlation matrix as a low-rank structure, this paper derives a quantified closed-form solution for the output of the $t$-th student model, showing the effect of self-distillation can be interpreted as label averaging among highly correlated instances. The authors then derive the conditions of the label corruption matrix achieving 100% population accuracy for multi-round self-distillation, in the context of balance and superclass corruption. The authors further prove that distilling teacher's top-2 outputs enjoys better theoretical properties. Extensive experiments are conducted and show consistency with the proposed theories.

**Strengths:**

- The proposed theory is solid and valuable, especially providing insight into the effectiveness of distilling with partial labels.
- The authors provide numerical and visual analysis of the approximation of the softmax function and the feature correlation matrix, which is reasonable and convincing.
- This paper is well-organized and clearly written.

**Weaknesses:**

In the experiments, Generalized Cross Entropy loss is applied for PLL which differs from the setting of CE loss in the main theory. The authors need to explain whether applying GCE loss is fair for comparison or provide experimental results of CE loss for PLL.

**Questions:**

See Weaknesses.

---

> ### Author Response · Authors · 2024-11-18
> **Response to Reviewer g4HH**
>
> We sincerely thank the reviewer for the constructive feedback. Please see our responses to the reviewer’s questions.
>
> >**W1. In the experiments, Generalized Cross Entropy loss is applied for PLL which differs from the setting of CE loss in the main theory. The authors need to explain whether applying GCE loss is fair for comparison or provide experimental results of CE loss for PLL.**
>
> Thank the reviewer for the insightful comment. We clarify our rationale for using the Generalized Cross Entropy (GCE) loss in our experiments with the PLL student model, and provide additional experimental results using the Cross Entropy (CE) loss below.
>
> The GCE loss is designed to balance the trade-off between the robustness of the Mean Absolute Error (MAE) loss and the fast convergence of the CE loss. In a $K$-class classification problem, for a given data pair $(\boldsymbol{x}\_i, y\_i)$, the CE loss $\mathcal{L}\_\mathsf{CE}$ and the MAE loss $\mathcal{L}\_\mathsf{MAE}$ are defined as
> $$
> \mathcal{L}\_\mathsf{CE}(\mathbf{e}(y\_i), f(\boldsymbol{x}\_i;\theta)) = -\sum\_{k=1}^K [\mathbf{e}(y\_i)]\_k \log[f(\boldsymbol{x}\_i;\theta)]\_k, \quad \mathcal{L}\_\mathsf{MAE}(\mathbf{e}(y\_i), f(\boldsymbol{x}\_i;\theta)) = \sum\_{k=1}^K |[\mathbf{e}(y_i)]\_k - [f(\boldsymbol{x}\_i;\theta)]\_k|.
> $$ The gradients for each loss are given as
> $$
> \nabla\_\theta \mathcal{L}\_\mathsf{CE}(\mathbf{e}(y\_i), f(\boldsymbol{x}\_i;\theta)) = -\frac{1}{[f(\boldsymbol{x}\_i;\theta)]\_{y\_i}} \nabla\_\theta [f(\boldsymbol{x}\_i;\theta)]\_{y\_i}, \quad \nabla\_\theta \mathcal{L}\_\mathsf{MAE}(\mathbf{e}(y\_i), f(\boldsymbol{x}\_i;\theta)) = -2\nabla\_\theta [f(\boldsymbol{x}\_i;\theta)]\_{y\_i}.
> $$CE loss tends to converge quickly but can overfit to noisy labels, while MAE loss is robust to label noise but converges slowly.
>
> The GCE loss introduces a hyperparameter $q$ to interpolate between CE and MAE losses:
> $$\mathcal{L}\_\mathsf{GCE}(\mathbf{e}(y\_i), f(\boldsymbol{x}\_i;\theta)) = \frac{1 - ([f(\boldsymbol{x}\_i;\theta)]\_{y_i})^q}{q},
> $$with gradient:
> $$
> \nabla_\theta \mathcal{L}\_\mathsf{GCE}(\mathbf{e}(y\_i), f(\boldsymbol{x}\_i;\theta)) = -[f(\boldsymbol{x}\_i;\theta)]\_{y\_i}^{q-1} \nabla\_\theta [f(\boldsymbol{x}\_i;\theta)]\_{y\_i}.
> $$By adjusting $q$, GCE loss balances robustness to label noise and convergence speed.
>
> In our experiments, we observed that using CE loss with the PLL student model often led to instability during training. The PLL student model trains with a set of candidate labels for each sample with equal weights—in our case, the top two labels with weights of $1/2$ each. Using CE loss with equally weighted candidate labels can cause instability because the model may converge incorrectly when the candidate set includes incorrect labels.
>
> Various PLL approaches address this instability by refining the candidate label set during training or adjusting label weights. However, our focus is to demonstrate the effectiveness of one-step self-distillation using PLL without incorporating additional PLL techniques. Therefore, we utilized the GCE loss, which behaves similarly to the CE loss but offers greater stability during training.
>
> We also conducted additional experiments using CE loss with the PLL student model. The test accuracies on the CIFAR-100 dataset under varying label corruption rates ($\eta$) are presented below (corresponding to Figure 4 and Table 5 in the main text):
> |DistillationStep|0.0|0.1|0.3|0.5|0.6|0.7|0.8|0.9|
> |----------------|----|----|----|----|----|----|----|----|
> |1(Teacher)|70.6±0.1%|65.6±0.1%|59.6±0.2%|52.4±0.2%|46.8±0.1%|39.6±0.3%|28.3±0.3%|13.2±0.2%|
> |2|71.7±0.2%|69.6±0.1%|66.0±0.2%|62.1±0.2%|58.5±0.6%|53.2±0.4%|43.1±0.4%|22.6±1.2%|
> |3|**72.1±0.2%**|69.8±0.3%|66.3±0.2%|62.4±0.1%|58.6±0.6%|53.4±0.5%|43.6±0.3%|23.3±1.1%|
> |4|72.0±0.3%|69.8±0.3%|66.5±0.2%|62.4±0.1%|58.7±0.6%|53.5±0.5%|43.7±0.3%|23.5±1.2%|
> |5|72.0±0.3%|**69.9±0.3%**|66.5±0.2%|62.4±0.1%|58.7±0.5%|53.6±0.5%|43.9±0.3%|23.7±1.2%|
> |PLL(GCE)|69.5±0.2%|67.9±0.1%|**66.9±0.2%**|**63.9±0.3%**|**61.3±0.4%**|**57.1±0.1%**|**48.6±0.9%**|**26.5±0.7%**|
> |PLL(CE)|68.7/68.6/68.3%|66.3/66.4/**64.6**%|65.0/65.3/65.6%|62.1/62.8/62.3%|59.6/60.0/60.0%|55.8/55.6/55.0%|47.2/**42.4**/47.5%|**22.8**/25.2/25.8%|
>
> Across three repeated experiments, the PLL student model still generally outperforms the multi-round self-distillation model in high corruption regimes ($\eta\geq 0.6$), even with CE loss. However, we observed greater variability and occasional drops in accuracy when using CE loss. GCE loss provided more consistent and stable performance across different corruption rates. We will include these explanations in the revised manuscript to clarify our experimental setup.

---

> > ### Comment · Reviewer_g4HH · 2024-11-25
> >
> > I'm satisfied with the author's response and I still vote for acceptance.

---

### Official Review · Reviewer_XuSH · 2024-11-04

**Soundness:** 3
**Presentation:** 3
**Contribution:** 3
**Rating:** 6
**Confidence:** 3

**Summary:**

This paper presents a theoretical analysis of the mechanisms behind self-distillation in a linear probing setting. The analysis reveals that after $t$ rounds of self-distillation, the model's predictions converge to a weighted average of the provided labels, with the weights determined by the Gram matrix. Building on this finding, the authors investigate the effects of label noise and the efficiency of the self-distillation method. Experiments demonstrate the effectiveness of proposed single-round self-distillation method.

**Strengths:**

* This paper is well-written, particularly in Section 2, which formulates self-distillation and provides a clear overview of the results.
* This paper presents an interesting result in Theorem 2.1, establishing a connection between the predictions of the $t$-th distilled model and the given (possibly noisy) labels.

**Weaknesses:**

The main weakness about this paper is the significance of research problem.

* This paper focuses on self-distillation with linear probing and provides a theoretical analysis in this context. I believe that both self-distillation and linear probing are valuable techniques, but I am unclear about the purpose of combining self-distillation with linear probing. As far as I know, linear probing is widely used in self-supervised learning as a method to evaluate learned features. Why should we combine linear probing with self-distillation, especially in scenarios involving label noise?

* The proposed theory assumes a fixed feature extractor and therefore cannot be applied to trainable feature extractors, which are more commonly used in task adaptation and transfer learning.

**Questions:**

See my questions in weakness part.

---

> ### Author Response · Authors · 2024-11-18
> **Response to Reviewer XuSH (1/2)**
>
> We sincerely thank the reviewer for the constructive feedback. Please see our responses to the reviewer’s questions.
>
> >**W1. This paper focuses on self-distillation with linear probing and provides a theoretical analysis in this context. I believe that both self-distillation and linear probing are valuable techniques, but I am unclear about the purpose of combining self-distillation with linear probing. As far as I know, linear probing is widely used in self-supervised learning as a method to evaluate learned features. Why should we combine linear probing with self-distillation, especially in scenarios involving label noise?**
>
> Combining self-distillation with linear probing is particularly beneficial when adapting large pre-trained neural networks (foundation models) to downstream tasks. Fine-tuning an entire pre-trained model is computationally intensive and may risk overfitting, especially with limited labeled data or noisy labels. Linear probing, which trains only a linear classifier on top of a fixed feature extractor, offers a more efficient way to utilize the rich features of pre-trained models. While linear probing is efficient, the linear classifier can still overfit to noisy or corrupted labels. Self-distillation improves linear probing by reducing overfitting and enhancing generalization. By training the linear classifier over multiple rounds, self-distillation shifts the model's focus from fitting noisy labels to leveraging feature correlations among instances. This process helps the classifier make more robust predictions based on the underlying data structure rather than the potentially noisy labels.
>
> We also propose a new self-distillation scheme that achieves these benefits in a single round using partial labels from the teacher's top predictions. This approach further reduces computational complexity while maintaining or even improving performance. It is especially useful when computational resources are limited or when quick adaptation to new tasks is required without extensive retraining.
>
> In summary, combining self-distillation with linear probing enhances the effectiveness of linear probing by improving generalization and robustness, especially in domains with noisy labels (e.g., crowdsourced data). This combination is valuable in applications where computational efficiency and model robustness are critical.

---

> ### Author Response · Authors · 2024-11-18
> **Response to Reviewer XuSH (2/2)**
>
> >**W2. The proposed theory assumes a fixed feature extractor and therefore cannot be applied to trainable feature extractors, which are more commonly used in task adaptation and transfer learning.**
>
> While our main analysis assumes a fixed feature extractor and focuses on training a linear classifier on top of it, our theory can be generalized to scenarios where the feature extractor is updated during self-distillation, as we explain in Appendix C. Our framework decouples the gains from self-distillation into two components: feature learning (which modifies the feature map assumed in Equation (7)) and feature selection (which occurs during training the classifier, i.e., linear probing).
>
> To illustrate this point, consider the quantified gains of self-distillation in label-noise scenarios presented in Theorem 4.1. This theorem provides the sufficient number of distillation rounds required for the student's softmax output to assign the highest value to the ground-truth label for both clean and noisy samples. The condition depends not only on the class-wise label corruption rates but also on the relative gap in feature correlations between samples of the same ground-truth class, parameterized by $c$, and those of different ground-truth classes, parameterized by $d$.
>
> We can generalize our theory by allowing these correlations to evolve over distillation rounds, denoted by $c^{(i)}$ and $d^{(i)}$, due to feature updates during self-distillation. Recent work by Allen-Zhu and Li (2022) demonstrates that self-distillation's effectiveness arises from an implicit ensemble of teacher and student models, enabling the student to learn more diverse features when using the teacher's softmax outputs as targets. This suggests that the intra-class feature correlation $c^{(i)}$ may increase with each distillation round $i$, enhancing the separation between classes.
>
> Assuming the class-wise feature map defined in Equation (7), and allowing $c$ and $d$ to change over distillation steps, our parameters $p$ and $q$ in Equation (14), which govern the label averaging effect, also become functions of $i$: $$
> p^{(i)}:=(1-c^{(i)})/(K^2 n\lambda+1-c^{(i)}); \quad q^{(i)}:=(1-c^{(i)}+n(c^{(i)}-d^{(i)}))/(K^2 n\lambda+1-c^{(i)}+n(c^{(i)}-d^{(i)})).
> $$
> Under the extended class-wise feature correlation assumption, our Theorem 4.1 can be generalized to:
>
> **Theorem C.1 (extended version.)** *Under the evolving feature correlation model, the $t$-th distilled model achieves 100% population accuracy if*
> $$[\mathbf{C}]\_{k, k}>[\mathbf{C}]\_{k, k'}+\frac{1}{\prod\_{i=1}^t ({q^{(i)}}/{p^{(i)}})- 1},\quad\forall k,k'(\neq k)\in[K].$$
>
> If the student model learns more diverse features than the teacher, resulting in an increase of $c^{(i)}$ over distillation rounds, the ratio $q^{(i)}/p^{(i)}$
> also increases. This makes it easier for the student model to meet the condition for achieving 100% population accuracy. Consequently, the regime where the student model achieves perfect accuracy expands, leading to performance gains from self-distillation.
>
> In summary, while our primary focus was on understanding self-distillation's benefits with a fixed feature extractor, our analysis can be extended to incorporate feature learning dynamics. By integrating our findings with existing feature learning approaches, we demonstrate that self-distillation enhances performance through both label averaging and the evolution of feature representations. Our extended theory quantifies the gains from both aspects, as shown in Theorem C.1. This extended analysis is detailed in Appendix C of our manuscript.
>
> [1] Allen-Zhu, Z. and Li, Y. Towards understanding ensemble, knowledge distillation and self-distillation in deep learning. In The Eleventh International Conference on Learning Representations, 2022.

---

> > ### Comment · Reviewer_XuSH · 2024-11-23
> >
> > Thank you for your detailed response. My concerns are totally resolved. I believe the theoretical analysis in the main paper and appendix offers valuable insight into the area of self-distillation and transfer learning. Thus, I decide to raise my score.

---

### Official Review · Reviewer_UkvB · 2024-11-07

**Soundness:** 3
**Presentation:** 3
**Contribution:** 3
**Rating:** 8
**Confidence:** 3

**Summary:**

This work aims to analyze self-distillation in linear probing with neural network feature extractors. The contributions of this work are threefold: (1) the analysis reveals that self-distillation effectively performs label averaging among instances with highly correlated features when generating predictions on the training data; (2) the analysis quantifies the number of distillation rounds needed to achieve 100% population accuracy in the presence of label noise; and (3) based on the theoretical analysis, the authors introduce a novel self-distillation approach that achievers similar benefits of multi-round distillation in a single round.

**Strengths:**

1. Solid Presentation
Overall, this manuscript is well-written and polished. The introduction provides a satisfying overview of the rich theoretical analysis, which is a strong aspect of the work. Some parts of the analysis are somewhat difficult to follow, but experts in this specific field likely won’t struggle to understand the details.

2. Interesting theoretical analysis
One of the main theoretical contributions of this paper is demonstrating that the effect of self-distillation can be interpreted as label averaging among highly correlated instances. This interpretation offers a new perspective on the mechanism of self-distillation, even in the absence of feature evolution. Initially, I thought this was merely a special case of Allen-Zhu and Li’s study. However, after reviewing previous work, I discovered a notable difference in how the effects of self-distillation are analyzed, which I believe is worth sharing with the community.

3. A New Approach to Converting Multi-Round Self-Distillation into a Single Round
The proposed method takes a different approach by directly using the refined partial labels derived from the teacher's outputs, achieving the same benefits as multi-round distillation in just one round. This is especially appealing, as one of the main drawbacks of self-distillation is the need for multiple rounds, which makes training very inefficient. A single-shot method is highly desirable, and we hope this approach will be widely adopted for training classifiers in noisy label settings.

**Weaknesses:**

1. Most sections are well-written, but some parts would benefit from clearer descriptions or revision. Please refer to the section below.

2. The experiments are somewhat weak, but this is understandable since this work is more theoretical.

**Questions:**

1. The second contribution of this work is to determine the number of distillation rounds required to achieve 100% population accuracy in the presence of label noise. On first reading, the term "100% population accuracy" may not be immediately clear. Introducing a brief explanation of this concept in the first place would be beneficial for readers who may be less familiar with the subject.

2. Could you explain the intuition behind the condition (Eq. 10) in Theorem 4.1? At first glance, it seems almost implausible to reduce training losses to zero in the presence of label noise, even with multiple rounds of self-distillation.

3. It seems that the terms “true label” and “given label” are not properly defined. As I understand it, the “given label” refers to the target label, which may include noise, while the “true label” represents the oracle.

---

> ### Author Response · Authors · 2024-11-18
> **Response to Reviewer UkvB (1/2)**
>
> We sincerely thank the reviewer for the constructive feedback. Please see our responses to the reviewer’s questions.
>
> >**Q1. The second contribution of this work is to determine the number of distillation rounds required to achieve 100% population accuracy in the presence of label noise. On first reading, the term "100% population accuracy" may not be immediately clear. Introducing a brief explanation of this concept in the first place would be beneficial for readers who may be less familiar with the subject.**
>
> Thank you for your feedback. We will clarify the term “100% population accuracy” where it first appears in the manuscript. In our problem setup, we consider a ground-truth distribution of input-label pairs $(\mathbf{x},y(\mathbf{x}))\sim\mathcal{P}$. The population accuracy of a classifier $\boldsymbol{\theta}\in\mathbb{R}^{d\times K}$ is defined as $\mathbb{E}\_{(\mathbf{x},y(\mathbf{x})) \sim\mathcal{P}}\left[\mathbb{1}\left(\arg\max\_{k\in[K]}[\sigma(\boldsymbol{\theta}^{\top} \phi(\mathbf{x}))]\_k =y(\mathbf{x})\right)\right],$ i.e., the probability that the classifier (the softmax output) correctly predicts the ground-truth label $y$ for an input $\mathbf{x}$, drawn from $\mathcal{P}$.
>
> Achieving 100% population accuracy means that the classifier correctly identifies the ground-truth class for all possible inputs from the distribution $\mathcal{P}$. In other words, despite the presence of label noise in the training data, the classifier perfectly generalizes to the true underlying data distribution. Assuming a sufficiently large number $n$ of training instances per class, this indicates that the trained model can correctly classify the (potentially noisy) training instances into their true ground-truth classes rather than overfitting to the given noisy labels. We will incorporate this explanation into the manuscript to ensure clarity for all readers.

---

> ### Author Response · Authors · 2024-11-18
> **Response to Reviewer UkvB (2/2)**
>
> >**Q2. Could you explain the intuition behind the condition (Eq. 10) in Theorem 4.1? At first glance, it seems almost implausible to reduce training losses to zero in the presence of label noise, even with multiple rounds of self-distillation.**
>
> (Eq. 10) in Theorem 4.1 provides a condition under which the $t$-th self-distilled model can achieve 100% population accuracy, even in the presence of label noise in the training dataset. The key intuition behind this condition lies in the label averaging effect of multi-round self-distillation.
>
> Each round of self-distillation causes the model's output predictions to gradually shift away from the one-hot encoded given labels—which may be noisy—toward the average label vectors of instances with high feature correlations (i.e., instances from the same ground-truth class). This process is described in (Eq. 9) of Theorem 3.1. Essentially, at each distillation round, the student model is trained to fit new targets that increasingly reflect the collective information from highly correlated instances, rather than fitting the noisy labels. As the number of distillation rounds $t$ increases, the influence of individual noisy labels diminishes, and the student model relies more on the averaged labels derived from other instances of the same class. This shift reduces the impact of label noise because the averaging process amplifies the true signal (correct labels) while diluting the noise (incorrect labels).
>
> (Eq. 10) specifies the sufficient number of distillation rounds $t$ needed for this label averaging effect to ensure that the model's softmax outputs assign the highest probability to the ground-truth label for all training instances—including those with noisy labels.
> To understand the condition in (Eq. 10) more intuitively, consider (Eq. 13), which shows that the output prediction for a particular training instance can be expressed as a weighted combination of: 1) The given (possibly noisy) label, 2) The average label vector among instances from the same ground-truth class, 3) The average label vector among instances from the same superclass, 4) The uniform distribution vector.
>
> The weights assigned to these components depend on the number of distillation rounds $t$ and shift away from the individual noisy label toward the average label as $t$ increases. As long as the proportion of correctly labeled instances in each class is higher than the proportion of mislabeled ones (i.e., $[C]\_{k,k}>[C]\_{k,k’}$ for all $k\neq k’$), the average label vector will have its highest value at the ground-truth label position. Therefore, after sufficient distillation rounds, the model's predictions will correctly identify the ground-truth label, even for samples that were initially mislabeled.
>
> In summary, the condition in (Eq. 10) captures the relationship between the label corruption rates and the effectiveness of the label averaging process over multiple distillation rounds. It provides the minimum number of rounds needed for the model to overcome label noise by leveraging the collective information from correlated instances, ultimately achieving perfect accuracy despite the presence of noisy labels. We will add this high-level intuition in the main text.
>
> >**Q3. It seems that the terms “true label” and “given label” are not properly defined. As I understand it, the “given label” refers to the target label, which may include noise, while the “true label” represents the oracle.**
>
> In our paper, the true label $y(\mathbf{x})$ refers to the ground-truth class of each sample $\mathbf{x}$. This represents the actual class to which the sample inherently belongs, as determined by the underlying data distribution. The given label $\hat{y}$ is the label provided during training. This is the target label used by the teacher model, and it may include label noise. We will revise our paper to clarify these terms.

---

> > ### Comment · Reviewer_UkvB · 2024-11-26
> >
> > Thank you for your detailed response. My original concern has been resolved. I believe this paper meets the acceptance standards of ICLR.

---

### Author Response · Authors · 2024-11-21
**The revised paper is uploaded**

We sincerely thank the reviewers for their constructive feedback. Based on their comments, we have revised our paper accordingly, marking the changes in blue. The modifications are as follows:

* **(Sec. 1 L77-78, Sec. 2 L246-250)**: Clarified the explanation regarding 100% population accuracy. (Reviewer UkvB, Q1)
* **(Sec. 2 L130-131, 134-135 / App.J.1 L1578, L1581, 1682 / App.K L1866, 1885, 1922)**: Clearly defined “ground-truth label” and “given label”, and replaced "provided label" with "given label" throughout the manuscript. (Reviewer UkvB, Q3)
* **(App.C)**: Elaborated on extending our work to more general settings of self-distillation, involving trainable feature extractors during multi-round self-distillations. (Reviewer XuSH, W2)
* **(App.E.3)**: Provided a more detailed explanation for the use of the GCE loss in our experiments with the PLL student model. (Reviewer g4HH, W1)
* **(App.H.2)**: Added experiments using a larger feature extractor backbone (ViT-B) to the ablation section. (Reviewer CZ26, W3)

We have also provided detailed responses to all the reviewers' questions. We hope that these responses address the reviewer’s concerns and questions.

---

### Meta-Review · Area_Chair_D3Yn · 2024-12-21

**Metareview:**

This paper provides a theoretical analysis of the mechanisms underlying self-distillation in a linear probing setting. The analysis demonstrates that after several rounds of self-distillation, the model's predictions converge to a weighted average of the provided labels, with the weights determined by the Gram matrix. Leveraging this insight, the authors examine the impact of label noise and evaluate the efficiency of the self-distillation method.

Experiments validate the effectiveness of the proposed single-round self-distillation method. A key theoretical contribution of the paper is the interpretation of self-distillation as label averaging among highly correlated instances, offering a fresh perspective on its mechanism even in the absence of feature evolution. Additionally, the authors present numerical and visual analyses of the approximation of the softmax function and the feature correlation matrix, providing reasonable and convincing evidence to support their findings.

**Additional Comments On Reviewer Discussion:**

There are some concerns on experiments and theoretical results and insights. After rebuttal, the authors have addressed the concerns raised by the reviewers. The reviewers have increased the rating accordingly.

---

### Decision · Program_Chairs · 2025-01-22

Accept (Poster)